# Universally Invariant Learning in Equivariant GNNs

**Jiacheng Cen**[1 2 3], **Anyi Li**[1 2 3], **Ning Lin**[1 2 3], **Tingyang Xu**[4 5], **Yu Rong**[4 5]
**Deli Zhao**[4 5], **Zihe Wang**[1 2 3], **Wenbing Huang**[1 2 3*]

[1] Gaoling School of Artificial Intelligence, Renmin University of China
[2] Beijing Key Laboratory of Research on Large Models and Intelligent Governance
[3] Engineering Research Center of Next-Generation Intelligent Search and Recommendation, MOE
[4] DAMO Academy, Alibaba Group, Hangzhou, China    [5] Hupan Lab, Hangzhou, China
{jiacc.cn, li_anyi, ninglin00}@outlook.com; {xuty_007, yu.rong}@hotmail.com;
zhaodeli@gmail.com; wang.zihe@ruc.edu.cn; hwenbing@126.com

## Abstract

Equivariant Graph Neural Networks (GNNs) have demonstrated significant success across various applications. To achieve completeness—that is, the universal approximation property over the space of equivariant functions—the network must effectively capture the intricate multi-body interactions among different nodes. Prior methods attain this via deeper architectures, augmented body orders, or increased degrees of steerable features, often at high computational cost and without polynomial-time solutions. In this work, we present a theoretically grounded framework called Uni-EGNN for constructing complete equivariant GNNs that is both efficient and practical. We prove that a complete equivariant GNN can be achieved through two key components: 1) a complete scalar function, referred to as the canonical form of the geometric graph; and 2) a full-rank steerable basis set. Leveraging this finding, we propose an efficient algorithm for constructing complete equivariant GNNs based on two common models: EGNN and TFN. Empirical results demonstrate that our model demonstrates superior completeness and excellent performance with only a few layers, thereby significantly reducing computational overhead while maintaining strong practical efficacy.

## 1 Introduction

Various types of scientific data, including chemical molecules, proteins, and other particle-based physical systems, are often represented as *geometric graphs* [1; 2]. This data structure not only captures node characteristics and edge information but also includes a 3D vector (such as position, velocity, etc.) for each node. To effectively process geometric graphs, equivariant Graph Neural Networks (GNNs) have been developed that enable equivariant message passing over nodes while adhering to the $E(3)$ or $SE(3)$ symmetries inherent in physical laws. These models have achieved significant success in various scientific tasks, including physical dynamics simulation, molecular property prediction, and protein design [3–30].

In current literature, a key objective in the design of equivariant GNNs is to achieve completeness, which refers to the universal approximation property [31]. This concept was initially explored by [32], who proved the universality of high-degree steerable models, specifically TFN [33]. Subsequent models, such as SEGNN [34] and MACE [35], can be similarly confirmed to be complete by establishing their relations with TFN. Regrettably, the theory proposed by [32] applies only to point clouds (i.e., fully connected geometric graphs) and relies on sufficiently large values of the layer number, body order, and feature degree.

---

*Wenbing Huang is the corresponding author.

39th Conference on Neural Information Processing Systems (NeurIPS 2025).

More recently, the introduction of the GWL-test [36] and further researches [37–40] have provided a new perspective by connecting the completeness of invariant models to the geometric isomorphism problem through an extension of the Weisfeiler-Lehman (WL) test [41]. Unfortunately, the GWL-test has only quasi-polynomial solutions [42], inheriting the same weakness as the original WL-test.

In this paper, we explore complete equivariant GNNs in a more effective and operable way. To begin with, we reformulate existing equivariant GNNs as an expansion of multi-body high-degree bases coupled with the Clebsch–Gordan (CG) tensor product (Eqs. (2) and (3)). This reformulation allows us to clearly identify the key limitations of current methods: they fail to achieve complete expansion when the degree and body order of CG tensor products are constrained, or incur prohibitively high computational costs otherwise. To overcome these weaknesses, we propose a novel expansion of equivariant GNNs (Eq. (4)), which does not necessarily require CG tensor product: the sum over a finite basis set with dynamic weights dependent on the input. The completeness is achieved if the weights (called the scalar function) are complete and the basis set is full-rank.

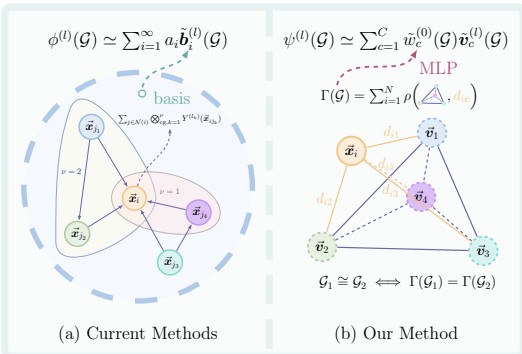

Figure 1: Difference between current methods and our Method. (a) Current methods can be understood as an expansion of multi-body high-degree bases $\tilde{\boldsymbol{b}}_i^{(l)}$ coupled with the Clebsch–Gordan (CG) tensor product (Eq. (3)). The completeness is achieved with sufficiently large values of the body order and feature degree. (b) In contrast, our method proposes a novel expansion which does not necessarily require CG tensor product (Eq. (4)): the sum over a finite basis set with dynamic weights dependent on the input. The completeness is ensured if the weights (the scalar function $\Gamma(\mathcal{G})$) are complete and the basis set $\tilde{\boldsymbol{V}}^{(l)}$ is full-rank.

Interestingly, constructing a complete scalar function is equivalent to solving the geometric isomorphism problem, which, unlike the traditional isomorphism task on topological graphs, can be resolved in polynomial time [42]. Actually, we have presented an algorithm operating with a complexity of $\mathcal{O}(N^6)$ where $N$ is the number of nodes, based on the four-point positioning principle, and further turned it into *canonical form* for comprehensive embeddings of geometric graphs. Additionally, we theoretically prove that a full-rank basis set of any degree can always be constructed, if the input geometric graph is asymmetric. Building on this foundation, we propose a more efficient algorithm with a complexity of $\mathcal{O}(N^2)$ to construct the canonical form specifically for asymmetric graphs.

Thanks to our theoretical findings, we modify existing equivariant GNNs (such as EGNN [43] and TFN [33]) into complete models. Specifically, we introduce two complete implementations of EGNN, termed EGNN/TFN$_{\mathrm{cpl}}$-global and EGNN/TFN$_{\mathrm{cpl}}$-local. We conduct seven sets of experiments, validating our theoretical results and demonstrating the superior performance of our method compared to previous models. Our model achieves great performance with few layers, thereby significantly reducing computational overhead while maintaining strong practical efficacy[2].

## 2  Related Work

**Equivariant GNNs.**  According to the operators used to build the model, equivariant GNNs can be categorized into two main classes: scalarization-based models and tensor-product-based models. Scalarization-based models (*e.g.*, EGNN [43] and PaiNN [44]) utilize scalars (*e.g.* distance and angles) as coefficients to linearly combine 3D vectors during node updates. In contrast, tensor-product-based models (*e.g.*, TFN [33] and MACE [35]) utilize spherical harmonics to maintain the equivariance of message passing and realize interactions between steerable features of different degrees through CG tensor products. Historically, researchers maintained that tensor-product-based models could provide richer information, despite their significantly higher computational resource demands [45–47]. However, the recent success of spherical-scalarization models such as SO3KRATES [48], HEGNN [49], and GotenNet [50] has prompted a reevaluation of this assumption, suggesting a

---

[2]Code is available at https://github.com/GLAD-RUC/Uni-EGNN.

potential shift towards scalar learning. Our research provides an important conclusion in this direction: a complete $l$th-degree steerable model can be constructed using two components—a complete scalar model and a full-rank basis set of $l$th-degree steerable features.

**Completeness of Equivariant GNNs.** Completeness in equivariant GNNs refers to their ability to approximate any continuous function with arbitrary precision on geometric graphs, thus sparking diverse research avenues. Initially, [32] demonstrated the universality of tensor-product-based models, notably TFN [33], in fully connected geometric graphs. Subsequently, building on the work of [51], later studies began examining the completeness of scalar models. For instance, [52] established the completeness of models such as DimeNet [53], GemNet [54], and SphereNet [55] under $\mathcal{A}$-unsymmetry conditions. Meanwhile, frame-based models [56–59] emerged, innovatively separating geometric structures into group operations and orbits for separate processing. However, this approach can disrupt permutation invariance in symmetric graphs. Furthermore, the GWL test framework, inspired by the WL-test on topological graphs, offers an upper bound on the expressive power of equivariant GNNs over sparse graphs, thereby influencing further researches [37–40]. Nevertheless, it is worth noting that many of these methods rely on message passing and require multiple iterations, akin to multi-layer networks. Our method utilizes the canonical form to derive a complete scalar function, and when integrated with a basis set, facilitates the construction of a fully equivariant neural network that can be effectively achieved with just a single-layer architecture.

## 3 Method

In this section, we first introduce necessary preliminaries in § 3.1. Then in § 3.2, we review existing equivariant GNNs from the perspective of basis construction. Subsequently, we present our framework in § 3.3, which proposes to construct complete equivariant GNNs from the perspective based on output space. The two components to achieve completeness, including a canonical form of geometric graphs and a full-rank steerable basis set, are presented in § 3.4. Finally in § 3.5, we demonstrate how to modify typical equivariant GNNs to be complete based on our theoretical findings. Comprehensive theoretical analysis and proofs are provided in Appendix A.

### 3.1 Preliminaries

**Geometric graph.** A geometric graph of $N$ nodes is defined as $\mathcal{G} := (\boldsymbol{H}, \vec{\boldsymbol{X}}; \boldsymbol{A})$, where $\boldsymbol{H} := \{\boldsymbol{h}_i \in \mathbb{R}^{C_H}\}_{i=1}^N$ and $\vec{\boldsymbol{X}} := \{\vec{\boldsymbol{x}}_i \in \mathbb{R}^3\}_{i=1}^N$ are node features and 3D coordinates, respectively; $\boldsymbol{A} \in \mathbb{R}^{N \times N}$ is the adjacency matrix representing topological connections between different nodes and each edge could be assigned with an edge feature $\boldsymbol{e}_{ij}$ if necessary.

**Equivariance.** Let $\mathbb{X}$ and $\mathbb{Y}$ be the input and output vector spaces, respectively. A function $\phi : \mathbb{X} \to \mathbb{Y}$ is called *equivariant* with respect to group $\mathfrak{G}$ if

$$\forall \mathfrak{g} \in \mathfrak{G}, \phi(\rho_{\mathbb{X}}(\mathfrak{g})\boldsymbol{x}) = \rho_{\mathbb{Y}}(\mathfrak{g})\phi(\boldsymbol{x}), \tag{1}$$

where $\rho_{\mathbb{X}}$ and $\rho_{\mathbb{Y}}$ are the group representations in the input and output spaces, respectively. Since translation invariance can always be achieved by translating the center of all coordinates to the origin, we omit translation and focus solely on equivariance concerning $\mathrm{O}(3)$, the group consisting of rotation and inversion. This implies $\mathrm{O}(3) = \mathrm{SO}(3) \times C_i$, where $\mathrm{SO}(3)$ represents the rotation group, and $C_i = \{\mathfrak{e}, \mathfrak{i}\}$ denotes the inversion group, with $\mathfrak{e}$ as the identity and $\mathfrak{i}$ as the inversion. We define two additional groups: the Euclidean group $\mathrm{E}(3)$ that further involves translation into $\mathrm{O}(3)$, and permutation group $\mathrm{S}_N$ acting on a set of $N$ elements.

**Irreducible representations and steerable features.** For a group element $\mathfrak{r} \in \mathrm{SO}(3)$, its representation is typically achieved through irreducible representations such as the Wigner-D matrix $\boldsymbol{D}^{(l)}(\mathfrak{r}) \in \mathbb{R}^{(2l+1) \times (2l+1)}$, where $l$ denotes the degree. We call features $\tilde{\boldsymbol{v}}^{(l)}$ as $l$th-degree steerable features if they are transformable using $\boldsymbol{D}^{(l)}$. The most common steerable features can be obtained via spherical harmonics, expressed as $Y^{(l)}(\vec{\boldsymbol{x}}/\|\vec{\boldsymbol{x}}\|) = [Y_m^{(l)}(\vec{\boldsymbol{x}}/\|\vec{\boldsymbol{x}}\|)]_{m=-l}^l$, which satisfy the property $Y^{(l)}(\boldsymbol{R}_{\mathfrak{r}}\vec{\boldsymbol{x}}/\|\boldsymbol{R}_{\mathfrak{r}}\vec{\boldsymbol{x}}\|) = \boldsymbol{D}^{(l)}(\mathfrak{r})Y^{(l)}(\vec{\boldsymbol{x}}/\|\vec{\boldsymbol{x}}\|)$. According to [60], spherical harmonics provide a *complete* basis set for $\mathrm{SO}(3)$-equivariant functions on the sphere. Clebsch–Gordan (CG) tensor products are usually leveraged to facilitate interactions between steerable features of differing degrees. Specifically, given features $\tilde{\boldsymbol{v}}^{(l_1)}$ and $\tilde{\boldsymbol{v}}^{(l_2)}$, a new feature $\tilde{\boldsymbol{v}}^{(l)}$ can be generated, for $|l_1 - l_2| \le l \le l_1 + l_2$. This calculation is abbreviated as $\tilde{\boldsymbol{v}}^{(l)} = \tilde{\boldsymbol{v}}^{(l_1)} \otimes_{\mathrm{cg}} \tilde{\boldsymbol{v}}^{(l_2)}$. By including

parity [61], the above concepts can be naturally extended to the orthogonal group O(3). A more detailed discussion is available in Appendix A.1.

## 3.2 Rethinking equivariant GNNs: a perspective based on basis construction

By definition, a complete equivariant model is able to express any equivariant function. For instance, spherical harmonics provide complete bases for SO(3)-equivariant functions on the sphere: $\phi(\vec{x}/\|\vec{x}\|)$, enabling the representation of all such functions. In 3D space beyond the sphere, a complete function is achieved by combining spherical harmonics with a learnable radial basis function $\varphi(\|\vec{x}\|)$. However, deriving complete equivariant functions on geometric graphs, namely $\tilde{\phi}^{(l)}(\mathcal{G})$,[3] is more complex due to the need to consider multi-body interactions and permutation symmetry.

To begin with, let us define a function set $\mathbb{F}^{(l)}$, which encompasses all square-integrable $l$th-degree steerable functions on geometric graphs. Let $\mathbb{B}^{(l)} \subset \mathbb{F}^{(l)}$ denote the basis set that expands the function space of equivariant GNNs. The learning process of equivariant GNNs becomes an approximation:

$$\tilde{\phi}^{(l)}(\mathcal{G}) \approx \sum_{\tilde{\boldsymbol{b}}_i^{(l)} \in \mathbb{B}^{(l)}} a_i \tilde{\boldsymbol{b}}_i^{(l)}(\mathcal{G}), \tag{2}$$

where $a_i \in \mathbb{R}$ are weight coefficients independent of the input. Enhancing the expressiveness of equivariant GNNs involves enlarging $\mathbb{B}^{(l)}$ until it coincides with $\mathbb{F}^{(l)}$. When $\mathrm{span}(\mathbb{B}^{(l)}) = \mathbb{F}^{(l)}$, the model is termed a *complete $l$th-degree steerable model*, or simply a *complete* model, and the basis set is denoted as $\mathbb{B}_{\mathrm{cpl}}^{(l)}$. Specifically, complete 0th-degree steerable models are referred to as *complete scalar models*.

Below, we demonstrate that standard equivariant GNNs share a common basis set construction, differing primarily in their body-order formulation. For simplicity and without loss of generality, we assume uniform node and edge features that can either be treated as identical or safely ignored. This abstraction allows us to focus purely on the geometric and topological aspects of the problem while maintaining the core theoretical insights.

**Example 3.1** (Basis Set of Common Equivariant GNNs). The basis set for common equivariant GNNs, such as EGNN [43], HEGNN [49], TFN [33], and MACE [35], can be unified into the following form:

$$\mathbb{B}_\nu^{(l)} = \{\sum_i \sum_{\boldsymbol{j} \in \mathcal{N}(i)} \bigotimes_{\mathrm{cg}, k=1}^\nu Y^{(l_k)}(\vec{\boldsymbol{x}}_{ij_k}/\|\vec{\boldsymbol{x}}_{ij_k}\|)\}, \tag{3}$$

where $\nu \geq 1$ denotes the body order, $\boldsymbol{j} = (j_1, \ldots, j_\nu)$ represents all chosen ordered neighbors, and $\vec{\boldsymbol{x}}_{ij_k} := \vec{\boldsymbol{x}}_i - \vec{\boldsymbol{x}}_{j_k}$. When only single-body neighbor is considered (*i.e.*, $\nu = 1$), Eq. (3) forms the basis set for EGNN [43] and HEGNN [49], applicable to degrees $l = 1$ and $l \geq 1$, respectively. In contrast, for multi-body interactions, Eq. (3) corresponds to TFN [33] when the body order $\nu = 1$, or MACE [35] with higher body orders $\nu \geq 1$.

Although these models adhere to the form of Eq. (3), their expressive power differs markedly. Multi-body models can create entirely new bases through tensor products and achieve completeness by either utilizing a sufficiently high body order [62] or stacking multiple layers [32], whereas single-body steerable models could not. However, multi-body models suffer from notable limitations: a) They may not encompass steerable features of all degrees, if input degrees are improperly chosen. b) The computational complexity increases sharply with higher degrees $l$ and body orders $\nu$, resulting in significant computational costs.

## 3.3 Reformulating equivariant GNNs: a perspective based on output space

Expanding the basis set of a complete equivariant GNN is theoretically appealing but practically challenging. In this section, we adopt an alternative approach by deriving the necessary model design from the characteristics of the desired output results. We identify a key insight: a complete $l$th-degree steerable model can be constructed using two components—a complete scalar model and a full-rank basis set of $l$th-degree steerable features.

---

[3]For clarity, our main text focuses exclusively on graph-level functions, where $\tilde{\phi}^{(l)}(\mathcal{G})$ remains invariant to any node permutation. Notably, our discussion is generalizable to both node- and edge-level functions, which will be discussed in Appendix A.2

We denote the target $l$th-degree steerable function as $\tilde{\phi}^{(l)} \in \mathbb{F}^{(l)}$. Although it might be represented with infinite bases in $\mathbb{B}_{\text{cpl}}^{(l)}$, the output of $\tilde{\phi}^{(l)}(\mathcal{G})$ is still a vector in $\mathbb{R}^{2l+1}$. Let $\tilde{V}^{(l)}(\mathcal{G}) = [\tilde{v}_1^{(l)}(\mathcal{G}), \ldots, \tilde{v}_C^{(l)}(\mathcal{G})] \in \mathbb{R}^{(2l+1) \times C}$ define an $l$th-degree steerable model with $C$-channel outputs of $l$th-degree steerable feature ($C \geq 2l + 1$). We have the following theorem:

**Theorem 3.2** (Dynamic Method). *Given a geometric graph $\mathcal{G}$, suppose there is a matrix $\tilde{V}^{(l)}(\mathcal{G})$ with $C$ channels of $l$th-degree steerable features denoted as $\tilde{v}_c^{(l)}(\mathcal{G})$ satisfying $\mathrm{span}(\tilde{V}^{(l)}(\mathcal{G})) = \mathbb{F}^{(l)}(\mathcal{G}) := \{\tilde{f}^{(l)}(\mathcal{G}) \mid \tilde{f}^{(l)} \in \mathbb{F}^{(l)}\} \subset \mathbb{R}^{2l+1}$. Then for any $l$th-degree steerable function $\tilde{\phi}^{(l)} \in \mathbb{F}^{(l)}$, there always exists $\tilde{w}^{(0)}(\mathcal{G}) := [\tilde{w}_c^{(0)}(\mathcal{G})]_{c=1}^C$ with $C$-channel output scalars, such that*

$$\tilde{\phi}^{(l)}(\mathcal{G}) = \sum_{c=1}^C \tilde{w}_c^{(0)}(\mathcal{G})\tilde{v}_c^{(l)}(\mathcal{G}). \tag{4}$$

A notable distinction from Eq. (2) is that the coefficient in front of the steerable features changes from $a_i$, which is independent of the input, to a scalar (function) that depends on the entire graph, denoted as $\tilde{w}_c^{(0)}(\mathcal{G})$. A natural idea is that if we can get a complete scalar model (capturing all possible scalars), then we can always satisfy the requirements of Theorem 3.2. Another crucial consideration is that the validity of Theorem 3.2 requires the basis matrix $\tilde{V}^{(l)}(\mathcal{G})$ to satisfy $\mathrm{span}(\tilde{V}^{(l)}(\mathcal{G})) = \mathbb{F}^{(l)}(\mathcal{G})$. We call such basis matrix that meets this criterion as $\mathbb{F}^{(l)}(\mathcal{G})$-*full-rank*. When $\mathbb{F}^{(l)}(\mathcal{G}) = \mathbb{R}^{2l+1}$, the concept of $\mathbb{F}^{(l)}(\mathcal{G})$-*full-rank* is equivalent to the traditional definition of *full-rank*. For simplicity, in contexts with no ambiguity, we refer to it as *full-rank*.

By comparing Eq. (4) with Eq. (2), our proposed dynamic method offers greater flexibility and simplicity, embodying the concept of complexity transfer. Instead of constructing the difficult complete basis set $\mathbb{B}_{\text{cpl}}^{(l)}$, we transform the process into obtaining a complete scalar function and then design the complete steerable model through Theorem 3.2. In the next section, we will detail how to acquire the complete scalar model and the full-rank basis set of $l$th-degree steerable features needed for Theorem 3.2.

## 3.4 Reach completeness: canonical form and full-rank basis set

In this section, we introduce how to obtain the complete scalar function $\tilde{w}_c^{(0)}(\mathcal{G})$ (referred to as the canonical form) and the full-rank basis set $\tilde{V}^{(l)}(\mathcal{G})$ required by Theorem 3.2.

**Geometric isomorphism and canonical form.** An equivalent problem to constructing a complete scalar function is determining the isomorphism of geometric graphs. To differentiate this from traditional graph isomorphism problem for topological graphs, we specifically refer to our task involving geometric graphs as *geometric isomorphism*. This concept can be defined similarly to the GWL-test [36], which assesses the equivalence of geometric graphs based on their geometric structures and embeddings.

**Definition 3.3** (Geometric Isomorphism). *Two geometric graphs $\mathcal{G}(\vec{X}^{(\mathcal{G})}, A^{(\mathcal{G})})$ and $\mathcal{H}(\vec{X}^{(\mathcal{H})}, A^{(\mathcal{H})})$ are called geometrically isomorphic if they fulfill both of the following isomorphisms*

1. ***Point Cloud Isomorphism:*** *The two point clouds $\vec{X}^{(\mathcal{G})}$ and $\vec{X}^{(\mathcal{H})}$ are isomorphic, i.e., $\exists \sigma \in S_N, \mathfrak{g} \in E(3), \forall i, \vec{x}_i^{(\mathcal{G})} = \mathfrak{g} \cdot \vec{x}_{\sigma(i)}^{(\mathcal{H})}$. Here, all $\langle \sigma, \mathfrak{g} \rangle$ make a nonempty set $\mathbb{M}(\mathcal{G}, \mathcal{H})$.*

2. ***Topological Isomorphism:*** *The topological graphs associated with the point clouds are isomorphic, i.e., $\exists \langle \sigma, \mathfrak{g} \rangle \in \mathbb{M}(\mathcal{G}, \mathcal{H}), \forall i, \forall j, [A_{ij}^{(\mathcal{G})}] = [A_{\sigma(i)\sigma(j)}^{(\mathcal{H})}]$.*

*Moreover, we denote the geometric isomorphism between $\mathcal{G}$ and $\mathcal{H}$ as $\mathcal{G} \cong \mathcal{H}$.*

The most significant difference from the isomorphism problem in topological graphs is that, to date, it remains unclear whether there exists a polynomial-time algorithm capable of determining whether two topological graphs are isomorphic [42; 63]. In contrast, when it comes to distinguishing whether two geometric graphs are geometrically isomorphic in accordance with Definition 3.3, the situation appears to be more favorable. There are indeed algorithms that can resolve this issue in polynomial time, as highlighted by [64]. For example, we present Algos. 1 and 2, which is based on the four-point positioning principle and operates with complexities of $\mathcal{O}(N^6)$ and $\mathcal{O}(N^8)$, respectively.

In most applications, the objective of an equivariant GNN is not to distinguish two geometric graphs, but rather to obtain a comprehensive embedding of a geometric graph. Consequently, we define the *canonical form* similarly to the definition provided in [64] as follows:

**Definition 3.4** (Canonical Form of Geometric Graph). *A canonical form of geometric graph is a graph-level scalar function* $\Gamma : (\mathbb{R}^{N \times 3}, \mathbb{R}^{N \times N}) \to \mathbb{R}^H$, *satisfy* $\mathcal{G} \cong \mathcal{H} \iff \Gamma(\mathcal{G}) = \Gamma(\mathcal{H})$.

Built upon Algos. 1 and 2, we have derived a canonical form $\Gamma(\cdot)$ as presented in Algo. 3. In summary, the process involves three steps: a) traversing all ordered four-point sets to serve as reference points for four-point positioning, with a time complexity of $\mathcal{O}(N^4)$; b) transforming the original coordinates into distance vectors measured from the reference points, and converting both point sets and edge sets into scalar sets, with a time complexity of $\mathcal{O}(N)$; c) mapping these scalar sets to the desired canonical form using DeepSet [65].

**Theorem 3.5** (General Canonical Form). *Given any geometric graph* $\mathcal{G}$, *Algo. 3 provides a method to create canonical form with time complexity* $\mathcal{O}(N^6)$.

However, Algo. 3 remains impractical because of its high complexity. We now explore the feasibility of attaining a more efficient canonical form by sacrificing some versatility. The crux of Algo. 3 lies in the choice of four non-coplanar points, which contributes to a quartic complexity in traversal. If we reformulate this as a generative problem, by treating the non-coplanar points as learnable graph-level features, we may reduce this quartic complexity factor. Specifically, we consider applying a E(3)-equivariant GNN $\zeta$ to generate four non-coplanar nodes (called virtual nodes below). We regard them as reference points in the four-point positioning principle, avoiding the fourth-order complexity cost by the original traversal. Thus, we derive Algo. 4, yielding more efficiency.

**Theorem 3.6** (Faster Canonical Form). *Given an* E(3)-*equivariant function* $\zeta$ *that generates four non-coplanar points on* $\mathcal{G}$, *Algo. 4 is able to create canonical form with time complexity* $\mathcal{O}(N^2)$.

It should be noted that Algo. 3 can encode any geometric graph, while Algo. 4 requires that four non-coplanar reference nodes can be learned through an equivariant GNN. In past studies, such as FastEGNN [66], it is assumed that non-coplanar virtual nodes can always be found, but in the discussion later, we will see that this is not the case.

**Full-rank basis set.** As previously addressed, the validity of Eq. (4) critically depends on the $\mathbb{F}^{(l)}(\mathcal{G})$-full-rank basis set, a condition not easy to satisfy. The study in [49] has demonstrated that, for symmetric geometric graphs (defined in Definition A.5), $\mathbb{F}^{(l)}(\mathcal{G})$ will degenerate to zero functions for certain values of degree $l$, implying that it is hard to design a full-rank basis set in this case. In practice, most scenarios study the case of asymmetric graphs, so we give priority to discussing them here. Our key inquiry is whether full-rank basis sets can always be constructed for asymmetric geometric graphs. We first present a pivotal theorem that significantly aids in our analysis.

**Theorem 3.7** (Coloring on Asymmetric Graph). *In an asymmetric geometric graph, each point can be assigned a unique color. This implies the existence of an* E(3) *invariant function that maps the features of the* $i$th *node to distinct values* $\boldsymbol{h}_i$. *Similarly, for directed edges* $ij$, *their features are mapped to distinct values* $\boldsymbol{e}_{ij}$.

We propose two node coloring methods: a) Distance to the center, denoted as $\oplus$; b) Tensor product, denoted as $\otimes$. We denote the uncolored model by $\varnothing$ and illustrate the coloring procedure in Fig. 2, taking the $\oplus$ method as an example. A detailed description is provided in Table 9. This theorem is fundamental and forms the core of our subsequent analyses. It decouples the roles of different nodes and edges, simplifying our discussion. This framework enables us to enhance or suppress specific basis functions by selecting appropriate weight functions, or even to focus on a particular basis function by setting the weights of all other basis functions to zero. Given that the node coordinates of the entire geometric graph are non-coplanar, the coordinate differences $\{\vec{\boldsymbol{x}}_{ij}\}$ form a full-rank matrix, which allows us to achieve the desired $\tilde{\boldsymbol{V}}^{(l)}(\mathcal{G})$. Additionally, spherical harmonic functions facilitate the mapping of first-degree steerable features to higher-degree representations. We obtain the following theorem.

**Theorem 3.8** (Existence of Full-Rank Basis Set). *For any given asymmetric graph* $\mathcal{G}$, *an* $\mathbb{F}^{(l)}(\mathcal{G})$-*full-rank* $\tilde{\boldsymbol{V}}^{(l)}(\mathcal{G})$ *can always be constructed for any degree* $l$.

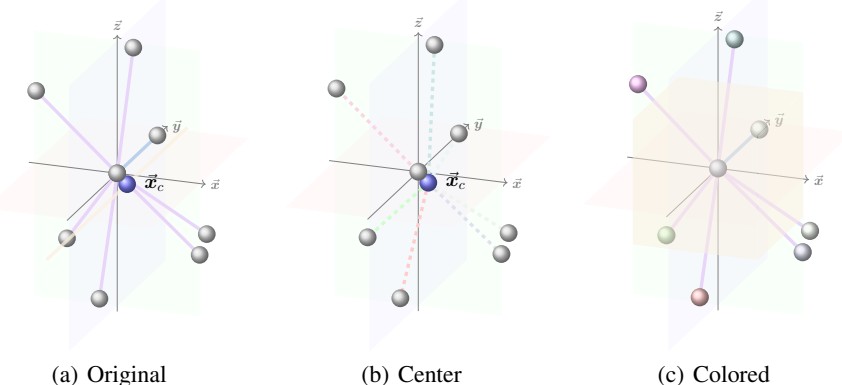

| (a) Original | (b) Center | (c) Colored |

Figure 2: Different vectors are decoupled by coloring, thereby expanding the space of virtual nodes. The yellow area in the figure represents the out space of 1st-degree functions. Fig. 2(a): All nodes have the same features, and the output can only appear in a straight line in one-dimensional space. Fig. 2(b): Calculate the distance to the center $\|\vec{x}_{ic}\|$ to update each node feature so that each node has a different color (feature). Fig. 2(c): After coloring, all vectors are decoupled, so the output can be generated in the entire 3D space.

For symmetric graphs, the situation is more complex, making the construction of a full-rank basis set considerably difficult. An potentially feasible approach could be using a tensor-product-based model, such as TFN [33] for basis set construction, which will be detailed in the appendix Appendix A.1.3.

### 3.5 Implementation of Complete Models

The last subsection demonstrated how to achieve theoretical completeness. Here, we present a practical implementation of complete models, using EGNN [43] as an example due to its simplicity and universality. In Example 3.1, we point out that EGNN could be incomplete since its basis set $\mathbb{B}^{(1)}$ is possibly reduced to the sum of all edges $\{\vec{x}_{ij}\}$. However, the edge set itself $\{\vec{x}_{ij}\}$ is full-rank and could be consider as the full-rank basis set $\tilde{V}^{(1)}$ in Theorem 3.2. To make EGNN complete, the remaining issue is how to implement the complete scalar function $\tilde{w}_c^{(0)}$. Inspired by Algo. 4, we derive the E(3)-equivariant function $\zeta$ in Theorem 3.6 using FastEGNN [66; 67], along with the pre-coloring process and several other modifications.

Essentially, we first color the geometric graph and then construct the reference virtual nodes through FastEGNN in the following form:

$$\vec{Z} = \mathbf{1}_4 \vec{x}_c + \frac{1}{N} \sum_{i=1}^{N} \varphi_{\vec{Z}}(h_i) \cdot (\vec{x}_i - \vec{x}_c), \qquad \vec{M}_i = \mathbf{1}_4 \vec{x}_i - \vec{Z}, \tag{5}$$

where $\vec{Z} \in \mathbb{R}^{4 \times 3}$ denotes the coordinates of four reference virtual nodes, $\vec{M}_i \in \mathbb{R}^{4 \times 3}$ computes coordinate difference between node $i$ and all virtual nodes, and $\vec{x}_c = \frac{1}{N} \sum_{i=1}^{N} \vec{x}_i$ signifies the coordinate center. In Algo. 4, the key point is to convert the coordinate information into a distance vector to the reference virtual node. To further refine the information, we employ the inner product form from GMN [68], as an alternative to distance vectors in Algo. 4. The node features are updated as follows:

$$h_i = h_i + \varphi_h(m_i), \qquad m_i = \vec{M}_i^\top \vec{M}_i / \|\vec{M}_i^\top \vec{M}_i\|_F. \tag{6}$$

Afterward, global operations (e.g., pooling) aggregate the information from all points and edges, mimicking DeepSet [65] calculations to obtain the desired canonical form. Strictly speaking, the virtual nodes forming the tetrahedron should also be included in Algo. 4, but we omit this term as it degrades performance in our experiments.

If we consider a single-layer EGNN, it becomes the following form:

$$\text{EGNN}_{\text{cpl}}(\mathcal{G}) = \frac{1}{N} \sum_{i=1}^{N} \vec{x}_i' = \vec{x}_c + \sum_{\langle i,j \rangle \in \mathcal{E}} \varphi_{\vec{x}}(m_{ij,\text{cpl}}) \cdot \vec{x}_{ij}, \tag{7}$$

where the message $m_{ij,\text{cpl}} = \varphi_m(h_i, h_j, \|\vec{x}_{ij}\|^2, e_{ij})$ has been a complete scalar function since $h_i$ already contains the information in the canonical form. Through Theorem 3.2, we verify that such single-layer EGNN is a complete model.

The above approach can also be extended to tensor-product-based models, and we have implemented corresponding improvements in TFN [33]. Since virtual nodes (i.e. $\vec{Z}$) are generated globally across

Table 1: *The Completeness Test.*

| | **The Completeness Test** | |
| **GNN Layer** | 2-body (Table. 3 in GWL) | 3-body (Fig. 2(b) in IASR) |
|---|---|---|
| SchNet$_{2\text{-body}}$ | 50.0 ± 0.0 | 50.0 ± 0.0 |
| EGNN$_{2\text{-body}}$ | 50.0 ± 0.0 | 50.0 ± 0.0 |
| GVP-GNN$_{3\text{-body}}$ | **100.0** ± 0.0 | 50.0 ± 0.0 |
| TFN$_{2\text{-body}}$ | 50.0 ± 0.0 | 50.0 ± 0.0 |
| MACE$_{3\text{-body}}$ | **100.0** ± 0.0 | 50.0 ± 0.0 |
| MACE$_{4\text{-body}}$ | **100.0** ± 0.0 | **100.0** ± 0.0 |
| Basic$_{\text{cpl}}$ | **100.0** ± 0.0 | **100.0** ± 0.0 |
| SchNet$_{\text{cpl}}$ | **100.0** ± 0.0 | **100.0** ± 0.0 |
| EGNN$_{\text{cpl}}$ | **100.0** ± 0.0 | **100.0** ± 0.0 |
| GVP-GNN$_{\text{cpl}}$ | **100.0** ± 0.0 | **100.0** ± 0.0 |
| TFN$_{\text{cpl}}$ | **100.0** ± 0.0 | **100.0** ± 0.0 |

Table 2: *The Chirality Test.*

| | # Color | # TP | **The Chirality Test** Fig. 4(a) | Fig. 4(b) | Fig. 4(c) |
|---|---|---|---|---|---|
| Basic | ∅ | | 50.0 ± 0.0 | 50.0 ± 0.0 | 50.0 ± 0.0 |
| | ⊕ | | **100.0** ± 0.0 | **100.0** ± 0.0 | 50.0 ± 0.0 |
| | ⊗ | | **100.0** ± 0.0 | **100.0** ± 0.0 | **100.0** ± 0.0 |
| | ∅ | ✓ | 75.0 ±15.0 | 95.0 ±15.0 | **100.0** ± 0.0 |
| | ⊕ | ✓ | **100.0** ± 0.0 | **100.0** ± 0.0 | **100.0** ± 0.0 |
| | ⊗ | ✓ | **100.0** ± 0.0 | **100.0** ± 0.0 | **100.0** ± 0.0 |
| EGNN | ∅ | | 50.0 ± 0.0 | 50.0 ± 0.0 | 50.0 ± 0.0 |
| | ⊕ | | **100.0** ± 0.0 | **100.0** ± 0.0 | 50.0 ± 0.0 |
| | ⊗ | | **100.0** ± 0.0 | **100.0** ± 0.0 | **100.0** ± 0.0 |
| | ∅ | ✓ | **100.0** ± 0.0 | **100.0** ± 0.0 | **100.0** ± 0.0 |
| | ⊕ | ✓ | **100.0** ± 0.0 | **100.0** ± 0.0 | **100.0** ± 0.0 |
| | ⊗ | ✓ | **100.0** ± 0.0 | **100.0** ± 0.0 | **100.0** ± 0.0 |

Table 3: MSE loss for predicting the coordinates of the Monge point and twelve-point center and incenter of a tetrahedron. Models with a superscript ∗ indicate that they utilize only half of the hidden dimension of the standard node embeddings. The optimal value is denoted in **bold**, while the suboptimal value is indicated with an underline.

| | **Monge Point** | | | | **Twelve-point Center** ($\times 10^{-1}$) | | | | **Incenter** ($\times 10^{-2}$) | | | |
| | 1-layer | 2-layer | 3-layer | 4-layer | 1-layer | 2-layer | 3-layer | 4-layer | 1-layer | 2-layer | 3-layer | 4-layer |
|---|---|---|---|---|---|---|---|---|---|---|---|---|
| EGNN | 1.188 | 0.374 | 0.185 | 0.131 | 1.320 | 0.413 | 0.265 | 0.161 | 32.54 | 0.527 | 0.141 | 0.036 |
| FastEGNN | 1.187 | 0.374 | 0.149 | 0.979 | 1.320 | 0.417 | 0.194 | 0.107 | 32.40 | 0.421 | 0.188 | 0.055 |
| HEGNN | 1.188 | 0.356 | 0.157 | 0.110 | 1.320 | 0.317 | 0.192 | 0.159 | 32.54 | 0.386 | 0.078 | 0.058 |
| TFN | 1.188 | 0.467 | 0.289 | 0.221 | 1.320 | 0.519 | 0.315 | 0.246 | 32.54 | 2.715 | 0.151 | 0.060 |
| MACE | 1.188 | 0.468 | 0.288 | 0.223 | 1.320 | 0.521 | 0.316 | 0.246 | 32.54 | 2.707 | 0.153 | 0.090 |
| Equiformer | 0.416 | 0.296 | 0.099 | 0.116 | 0.456 | 0.187 | 0.122 | 0.147 | 0.736 | 0.077 | 0.055 | 0.054 |
| EGNN$_{\text{cpl}}$-global | **0.108** | **0.086** | 0.097 | 0.104 | **0.112** | **0.097** | 0.099 | 0.088 | **0.025** | **0.018** | **0.015** | **0.012** |
| EGNN$^*_{\text{cpl}}$-global | 0.175 | 0.116 | 0.148 | 0.111 | 0.155 | 0.144 | 0.174 | 0.152 | 0.046 | 0.025 | 0.059 | 0.026 |
| EGNN$_{\text{cpl}}$-local | 0.460 | 0.210 | 0.083 | **0.052** | 0.510 | 0.232 | **0.095** | **0.067** | 0.388 | 0.258 | 0.031 | 0.029 |
| EGNN$^*_{\text{cpl}}$-local | 0.460 | 0.252 | 0.138 | 0.091 | 0.508 | 0.250 | 0.182 | 0.102 | 0.395 | 0.320 | 0.084 | 0.090 |
| TFN$_{\text{cpl}}$-global | 0.468 | 0.162 | 0.233 | 0.084 | 0.516 | 0.319 | 0.187 | 0.099 | 8.983 | 0.398 | 0.077 | 0.051 |
| TFN$^*_{\text{cpl}}$-global | 0.473 | 0.291 | 0.158 | 0.096 | 0.525 | 0.317 | 0.181 | 0.105 | 7.858 | 0.354 | 0.085 | 0.054 |
| TFN$_{\text{cpl}}$-local | 0.460 | 0.210 | 0.089 | 0.088 | 0.518 | 0.237 | 0.097 | 0.078 | 0.676 | 0.048 | 0.029 | 0.021 |
| TFN$^*_{\text{cpl}}$-local | 0.473 | 0.244 | **0.082** | 0.073 | 0.519 | 0.271 | 0.122 | 0.085 | 0.693 | 0.056 | 0.039 | 0.024 |

the entire graph, we refer to these models as EGNN/TFN$_{\text{cpl}}$-global. To better capture local information, virtual nodes can also be generated for individual nodes, resulting in the EGNN/TFN$_{\text{cpl}}$-local variants. Detailed model architectures are provided in Tables 12 and 13.

# 4 Experiment

In this section, we conduct two categories of experiments, totaling seven in number, to validate our theory and methods: a) three toy datasets in § 4.1 to assess the expressivity of our models; b) the remaining four in § 4.2 evaluate the actual performance of our models. Here, we provide a brief overview of the experiments. For further details, please refer to Appendix B.

## 4.1 Expressivity

**Dataset setup.** The **completeness test** is derived from the GWL-test [32] and the IASR-test [69], with the objective of distinguishing between two geometrically non-isomorphic graphs. We here focus on the *k-chain* test and 2-body/3-body tasks for testing. The **chirality test** is an experiment designed by ourselves, which includes three tasks as illustrated in Fig. 4. This study aims to address two objectives: a) assess the model's ability to distinguish geometric graphs under reflection transformations; (2) investigate whether distinct coloring strategies generate different virtual node coordinates, as validated by determinant calculations serving as mathematical indicators for reflection determination.

**Models.** For the completeness test, we selected several baseline models, including SchNet [70], EGNN [43], GVP-GNN [71], TFN [33], and MACE [35]. Furthermore, models utilizing our canonical form are denoted as *Model*$_{\text{cpl}}$, with *Basic*$_{\text{cpl}}$ specifically using the canonical form as the graph embedding directly. For the chirality test, we provide two options to identify reflections: a) Determinant $\det\big([\vec{z}_2, \vec{z}_3, \vec{z}_4] - \vec{z}_1 \mathbf{1}_{1\times 3}\big)$; b) Tensor product, as described in Example A.15.

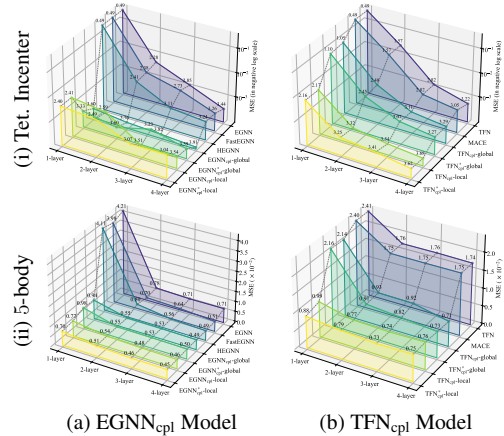

(i) Tet. Incenter

(ii) 5-body

(a) EGNN$_{cpl}$ Model  (b) TFN$_{cpl}$ Model

Figure 3: Visualization of MSE loss.

Table 4: MSE loss on 5-body system.

| | MSE Loss on 5-body system ($\times 10^{-2}$) | | | |
| --- | --- | --- | --- | --- |
| | 1-layer | 2-layer | 3-layer | 4-layer |
| EGNN | 4.214 | 0.780 | 0.710 | 0.712 |
| FastEGNN | 3.983 | 0.705 | 0.640 | 0.509 |
| HEGNN | 4.114 | 0.801 | 0.561 | 0.489 |
| TFN | 2.411 | 1.758 | 1.758 | 1.739 |
| MACE | 2.403 | 1.754 | 1.746 | 1.746 |
| Equiformer | 0.805 | 0.682 | 0.465 | 0.657 |
| EGNN$_{cpl}$-global | 0.943 | 0.546 | 0.530 | 0.492 |
| EGNN$_{cpl}^{*}$-global | 0.985 | 0.554 | 0.533 | 0.498 |
| EGNN$_{cpl}$-local | 0.768 | 0.537 | 0.481 | 0.458 |
| EGNN$_{cpl}^{*}$-local | **0.703** | **0.513** | **0.455** | **0.450** |
| TFN$_{cpl}$-global | 2.144 | 0.934 | 0.916 | 0.708 |
| TFN$_{cpl}^{*}$-global | 2.163 | 0.910 | 0.825 | 0.734 |
| TFN$_{cpl}$-local | 0.988 | 0.766 | 0.738 | 0.761 |
| TFN$_{cpl}^{*}$-local | 0.883 | 0.792 | 0.734 | 0.746 |

Table 5: 100-body dataset.

| | MSE Loss ($\times 10^{-2}$) |
| --- | --- |
| EGNN | 1.36 |
| FastEGNN | 1.10 |
| TFN$_{l\leq 2}$ | 3.77 |
| MACE$_{l\leq 2}$ | 3.83 |
| Equiformer$_{l\leq 2}$ | 0.90 |
| HEGNN$_{l\leq 1}$ | 1.13 |
| HEGNN$_{l\leq 2}$ | 0.97 |
| HEGNN$_{l\leq 3}$ | 0.94 |
| HEGNN$_{l\leq 6}$ | 0.86 |
| EGNN$_{cpl}$-global | 0.98 |
| EGNN$_{cpl}$-local | **0.73** |
| TFN$_{cpl}$-global$_{l\leq 2}$ | 1.78 |
| TFN$_{cpl}$-local$_{l\leq 2}$ | 1.73 |

Table 6: Prediction error ($\times 10^{-2}$) on MD17 dataset (3 runs).

| | Aspirin | Benzene | Ethanol | Malonal. | Naph. | Salicylic | Toluene | Uracil |
| --- | --- | --- | --- | --- | --- | --- | --- | --- |
| EGNN | 14.41$_{\pm 0.15}$ | 62.40$_{\pm 0.53}$ | 4.64$_{\pm 0.01}$ | 13.64$_{\pm 0.01}$ | 0.47$_{\pm 0.02}$ | 1.02$_{\pm 0.02}$ | 11.78$_{\pm 0.07}$ | 0.64$_{\pm 0.01}$ |
| FastEGNN | 9.81$_{\pm 0.11}$ | 60.84$_{\pm 0.14}$ | 4.65$_{\pm 0.00}$ | 12.82$_{\pm 0.02}$ | 0.38$_{\pm 0.00}$ | 1.05$_{\pm 0.08}$ | 10.88$_{\pm 0.08}$ | 0.56$_{\pm 0.01}$ |
| TFN$_{l\leq 2}$ | 12.37$_{\pm 0.18}$ | 58.48$_{\pm 1.98}$ | 4.81$_{\pm 0.04}$ | 13.62$_{\pm 0.08}$ | 0.49$_{\pm 0.01}$ | 1.03$_{\pm 0.02}$ | 10.89$_{\pm 0.01}$ | 0.84$_{\pm 0.02}$ |
| MACE$_{l\leq 2}$ | 10.43$_{\pm 0.44}$ | 59.71$_{\pm 2.21}$ | 4.83$_{\pm 0.03}$ | 13.78$_{\pm 0.04}$ | 0.44$_{\pm 0.02}$ | 0.94$_{\pm 0.01}$ | 10.20$_{\pm 0.11}$ | 0.74$_{\pm 0.01}$ |
| Equiformer$_{l\leq 2}$ | 9.84$_{\pm 0.10}$ | **33.28**$_{\pm 0.15}$ | 4.69$_{\pm 0.03}$ | 13.06$_{\pm 0.04}$ | 0.34$_{\pm 0.01}$ | 0.86$_{\pm 0.01}$ | **9.50**$_{\pm 0.09}$ | 0.57$_{\pm 0.01}$ |
| HEGNN$_{l\leq 1}$ | 10.32$_{\pm 0.58}$ | 62.53$_{\pm 7.62}$ | 4.63$_{\pm 0.01}$ | 12.85$_{\pm 0.01}$ | 0.38$_{\pm 0.01}$ | 0.90$_{\pm 0.05}$ | 10.56$_{\pm 0.10}$ | 0.56$_{\pm 0.02}$ |
| HEGNN$_{l\leq 2}$ | 10.04$_{\pm 0.45}$ | 61.80$_{\pm 5.92}$ | 4.63$_{\pm 0.01}$ | 12.85$_{\pm 0.01}$ | 0.39$_{\pm 0.01}$ | 0.91$_{\pm 0.06}$ | 10.56$_{\pm 0.05}$ | 0.55$_{\pm 0.01}$ |
| HEGNN$_{l\leq 3}$ | 10.20$_{\pm 0.23}$ | 62.82$_{\pm 4.25}$ | 4.63$_{\pm 0.01}$ | 12.85$_{\pm 0.02}$ | 0.37$_{\pm 0.01}$ | 0.94$_{\pm 0.10}$ | 10.55$_{\pm 0.16}$ | **0.52**$_{\pm 0.01}$ |
| HEGNN$_{l\leq 6}$ | 9.94$_{\pm 0.07}$ | 59.93$_{\pm 5.21}$ | 4.62$_{\pm 0.01}$ | 12.85$_{\pm 0.01}$ | 0.37$_{\pm 0.02}$ | 0.88$_{\pm 0.02}$ | 10.56$_{\pm 0.33}$ | 0.54$_{\pm 0.01}$ |
| EGNN$_{cpl}$-global | 9.60$_{\pm 0.09}$ | 58.24$_{\pm 1.40}$ | 4.64$_{\pm 0.01}$ | 12.85$_{\pm 0.01}$ | 0.39$_{\pm 0.01}$ | 0.95$_{\pm 0.05}$ | 10.37$_{\pm 0.16}$ | 0.56$_{\pm 0.02}$ |
| EGNN$_{cpl}$-local | 9.52$_{\pm 0.42}$ | 44.90$_{\pm 1.53}$ | 4.62$_{\pm 0.00}$ | **12.80**$_{\pm 0.02}$ | 0.36$_{\pm 0.02}$ | 0.94$_{\pm 0.05}$ | 10.21$_{\pm 0.06}$ | 0.57$_{\pm 0.00}$ |
| TFN$_{cpl}$-global$_{l\leq 2}$ | **9.49**$_{\pm 0.04}$ | 58.24$_{\pm 0.42}$ | 4.63$_{\pm 0.00}$ | 12.82$_{\pm 0.00}$ | **0.33**$_{\pm 0.00}$ | **0.80**$_{\pm 0.00}$ | 10.24$_{\pm 0.02}$ | 0.53$_{\pm 0.00}$ |
| TFN$_{cpl}$-local$_{l\leq 2}$ | 9.52$_{\pm 0.07}$ | 48.77$_{\pm 6.51}$ | 4.64$_{\pm 0.00}$ | 12.83$_{\pm 0.02}$ | 0.34$_{\pm 0.00}$ | 0.81$_{\pm 0.01}$ | 10.95$_{\pm 0.01}$ | 0.53$_{\pm 0.00}$ |

**Results.** Results are presented in Table 1 (the 2-body/3-body of completeness test), Table 2 (the chirality test) and Table 10 (the *k-chain* test of completeness test). As shown in Table 1, all models with canonical form, *i.e.* Model$_{cpl}$, reach an 100% accuracy, no matter whether the original model could distinguish the geometric graphs. As shown in Table 2, the model utilizing the determinant is highly dependent on the coloring scheme. In the special case illustrated in Fig. 4(c), only the model employing the tensor product can effectively distinguish between the geometric graphs, which demonstrates the importance of choosing an appropriate staining strategy.

## 4.2 Performance on Physical Systems

**Dataset setup.** We conducted four experiments to evaluate our model's performance across different scenarios: a) **tetrahedron center prediction**, which involves predicting the Monge point, twelve-point center, and incenter of a tetrahedron based on its four points (with identical node features); b) **5-body system** [72], c) **100-body system** [66], d) **MD17 dataset** [68], and e) **Water-3D mini** [66; 67], where predictions are made based on initial coordinates and velocities. Specifically, for the 5-body, 100-body, and Water-3D mini systems, we predict the future positions of charged particles or fluid particles, while for the MD17 dataset, we predict future atomic trajectories within eight different molecules.

These experiments provide a comprehensive evaluation of our model. We assess its performance on graph-level targets (a) and node-level targets (b-e), explore the influence of model architecture layers (a-b), and test its scalability in large systems (c,e) as well as its applicability to real-world datasets (d-e). Detailed information on each dataset can be found in Table 11.

**Models.** In both experiments, we selected the following baseline models: EGNN [43], FastEGNN [66; 67], HEGNN [49], TFN [33], MACE [35], and Equiformer [73]. We compared these baseline models with our proposed models, specifically EGNN/TFN$_{cpl}$-local and EGNN/TFN$_{cpl}$-global. More detailed information is provided in Tables 14 and 15.

**Results.** We evaluated the performance of each model across varying numbers of layers in the first two experiment. For specific values, please refer to Table 3 and Table 4. The experimental results are visualized in Fig. 3 to facilitate comparison. Our model demonstrates superior performance across all

Table 7: Results on Water-3D-mini.

| | MSE Loss on Water-3D-mini ($\times 10^{-4}$) | | | |
| | 1-layer | 2-layer | 3-layer | 4-layer |
|---|---|---|---|---|
| EGNN | 4.904 | 4.323 | 3.649 | 3.338 |
| FastEGNN | 4.885 | 4.332 | 3.782 | 3.259 |
| HEGNN | 4.885 | 4.138 | 3.519 | 3.287 |
| EGNN$_{cpl}$-global | 4.368 | 3.711 | 3.294 | 3.248 |
| EGNN$_{cpl}$-local | **3.611** | **3.320** | **2.803** | **2.495** |

tasks, often achieving results comparable to those of other models with multiple layers—sometimes even with just a single layer. In addition, our improved model has also achieved significant improvements on large-scale datasets and real-world datasets, see Tables 5 to 7, where our model is in the leading position in the former and in the latter, where we achieve performance leadership in more than half of the tasks.

**Additional findings.** Furthermore, the baselines in Fig. 3 demonstrate that losses are equal when a single layer is used, confirming the degradation predicted in Example A.16 and highlighting the necessity of node coloring. While TFN and MACE outperform EGNN and HEGNN with a single layer, their advantage decreases with two or more layers. This is likely because EGNN/HEGNN may initially lack necessary basis functions, whereas tensor products used in TFN/MACE in deeper layers produce redundant terms. The introduction of the dynamic method significantly enhances the performance of both EGNN and TFN, showing its ability to construct the missing basis initially and suppress redundant terms in tensor products, thus achieving *adaptive* basis construction. From this perspective, exploring how to encode invariant features with more powerful architectures (*e.g.*, advancing from GNNs to Graph Transformers [74]) represents a promising direction for future research. This perspective also helps explain the performance gains observed in models such as GotenNet [50] compared with HEGNN [49].

# 5   Conclusion

We propose a novel framework called Uni-EGNN, which is a dynamic method for constructing complete equivariant GNNs. In contrast to previous models, which required stacking multiple layers, increasing body order, and enhancing the degree of steerable features to achieve completeness, our dynamic method simplifies this process by requiring only two essential components: a) A canonical form (a complete scalar function) of a geometric graph; b) A full-rank steerable basis set. Additionally, we have developed an efficient implementation of dynamic method leveraging a polynomial algorithm for geometric isomorphism problems. Experimental results demonstrate that our dynamic method excels in both expressiveness and practical performance.

# 6   Limitations

For symmetric graphs, constructing a full-rank basis set remains an open problem. To date, it has only been established that tensor-product-based models (*e.g.*, TFN [33]) can construct such a basis, while for scalar methods (*e.g.*, HEGNN [49]) this remains uncertain—and we even conjecture that it may not be feasible. Another limitation of this work lies in the experimental evaluation: due to the lack of tasks with objective functions of degree $l \geq 2$, we are unable to empirically validate the effectiveness of the proposed method in predicting high-degree steerable objective functions.

# Acknowledgement

This work was jointly supported by the following projects and institutions: the National Natural Science Foundation of China (No. 62376276, No. 62172422); Beijing Nova Program (No. 20230484278); the Fundamental Research Funds for the Central Universities; the Research Funds of Renmin University of China (23XNKJ19); Public Computing Cloud, Renmin University of China; Damo Academy (Hupan Laboratory) through Damo Academy (Hupan Laboratory) Innovative Research Program and Damo Academy Research Intern Program.

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

# Contents of Appendix

# A Theoretical Analysis and Proofs

## A.1 Detailed Preliminaries

This section introduces the equivariant representation and the CG tensor product with inversion, along with the symmetric graph structure and its related properties.

### A.1.1 Equivariance

Let $\mathbb{X}$ and $\mathbb{Y}$ be the input and output vector spaces, respectively. A function $\phi : \mathbb{X} \to \mathbb{Y}$ is called *equivariant* with respect to group $\mathfrak{G}$ if

$$\forall \mathfrak{g} \in \mathfrak{G}, \phi(\rho_{\mathbb{X}}(\mathfrak{g})\boldsymbol{x}) = \rho_{\mathbb{Y}}(\mathfrak{g})\phi(\boldsymbol{x}), \tag{8}$$

where $\rho_{\mathbb{X}}$ and $\rho_{\mathbb{Y}}$ are the group representations in the input and output spaces, respectively.

Since we can permit translation invariance by simply translating the center of all coordinates to the origin, we only discuss equivariance with respect to $\mathrm{O}(3)$ in our paper. Specifically, the group $\mathrm{O}(3)$ consists of rotation and inversion, implying $\mathrm{O}(3) = \mathrm{SO}(3) \times C_i$, where $\mathrm{SO}(3)$ is the rotation group and $C_i = \{\mathfrak{e}, \mathfrak{i}\}$ denotes the inversion group with $\mathfrak{e}$ representing the identity and $\mathfrak{i}$ representing the inversion. In the current literature, researchers typically utilize irreducible representations (*i.e.* Wigner-D matrix $\boldsymbol{D}^{(l)}(\mathfrak{r}) \in \mathbb{R}^{(2l+1)\times(2l+1)}$ with $l$ denoting the degree) to represent $\mathrm{SO}(3)$ and parities $p \in \{\pm 1\}$ to represent the inversion group $C_i$ as follows:

$$\rho^{(l,p)}(\mathfrak{r}\mathfrak{m}) := \sigma^{(p)}(\mathfrak{m})\boldsymbol{D}^{(l)}(\mathfrak{r}), \tag{9}$$

where $\sigma^{(p)}(\mathfrak{m}) = p$ if $\mathfrak{m} = \mathfrak{i}$, while $\sigma^{(p)}(\mathfrak{m}) = 1$ if $\mathfrak{m} = \mathfrak{e}$. When considering only rotations, a $(2l+1)$-dimensional variable that can be rotated by the $l$th-degree Wigner-D matrices is referred to as an $l$th-degree steerable feature. When further involving the concept of inversion, we adopt the notation presented in e3nn [61]. Specifically, an inversion-invariant $l$th-degree steerable feature (*i.e.*, $p = 1$) is referred to as an $l$e-type, while an inversion-equivariant feature (*i.e.*, $p = -1$) is designated as an $l$o-type. These designations correspond to *even parity* and *odd parity*, respectively.

**Example A.1** (Spherical Harmonics)**.** Spherical harmonics $Y^{(l)}(\vec{\boldsymbol{x}}) = [Y_m^{(l)}(\vec{\boldsymbol{x}})]_{m=-l}^{l}$ are $l$th-degree steerable features, *i.e.* $Y^{(l)}(\boldsymbol{R}_{\mathfrak{r}}\vec{\boldsymbol{x}}) = \boldsymbol{D}^{(l)}(\mathfrak{r})Y^{(l)}(\vec{\boldsymbol{x}})$, and the parity is given as $p = (-1)^l$. It means spherical harmonics are $l$o-type when $l$ is odd and $l$e-type when $l$ is even. According to [60], spherical harmonics offer a *complete* set of function bases of $\mathrm{SO}(3)$-equivariant functions.

Another example about common physical quantities is presented in Example A.2.

**Example A.2** (Common Physical Quantities)**.** Scalars (*e.g.* mass, charge and energy) are $0$e-type (invariant to both rotation and inversion). Pseudo-scalar (*e.g.* magnetic charge and magnetic flux) are $0$o-type (invariant to rotation but change signs under inversion). Vectors (*e.g.* coordinate and velocity) are $1$o-type, while pseudo-vectors (*e.g.* torque and angular momentum) are $1$e-type.

**Example A.3** (Common Invariant features)**.** Due to the various advantages of scalars, there are a large number of construction methods, which we briefly list here:

1. Purely mathematical representation: tensor product (spherical-scalarization) [48–50], topological characteristics [75–77], cluster expansion basis [78; 79], subgraph blocks [80–82], frames [56; 83; 84], normalization operators [85; 86], LLM-based information [87];

2. Features based on physical and biochemical prior knowledge: distance matrix [40; 88–92], chemical bond length, angle and dihedral angle [93–97], force [98; 99], fractional coordinates [21; 100; 101], canonical ordering [102; 103].

### A.1.2 Clebsch–Gordan (CG) tensor product

Tensor product $\otimes(\cdot, \cdot)$ is a common operator to describe interaction between steerable features. Given an $l_1$th-degree steerable feature $\tilde{\boldsymbol{v}}^{(l_1)}$ and an $l_2$th-degree steerable feature $\tilde{\boldsymbol{v}}^{(l_2)}$, calculating their tensor product yields a new feature $\tilde{\boldsymbol{v}}^{(l_1)} \otimes \tilde{\boldsymbol{v}}^{(l_2)}$ with its group representation as $\boldsymbol{D}^{(l_1)}(\mathfrak{r}) \otimes \boldsymbol{D}^{(l_2)}(\mathfrak{r})$. The Clebsch-Gordan rule reveals that any tensor product of irreducible representations can be represented as the direct sum of the irreducible representations. Let $\mathbb{V}^{(l_1)}$ and $\mathbb{V}^{(l_2)}$ denote the irreducible

representation space of $l_1$th-degree and $l_2$th-degree, respectively. Then, we have the following relation:

$$\mathbb{V}^{(l_1)} \otimes \mathbb{V}^{(l_2)} = \bigoplus_{l=|l_1-l_2|}^{l_1+l_2} \mathbb{V}^{(l)}. \tag{10}$$

This further demonstrates that the group representation matrix $\boldsymbol{D}^{(l_1)}(\mathfrak{r}) \otimes \boldsymbol{D}^{(l_2)}(\mathfrak{r})$ is similar to the matrix $\bigoplus_{l=|l_1-l_2|}^{l_1+l_2} \boldsymbol{D}^{(l)}(\mathfrak{r})$. In general, the $l$th-degree component from the tensor product $\tilde{\boldsymbol{v}}^{(l_1)} \otimes \tilde{\boldsymbol{v}}^{(l_2)}$ can be expressed as

$$v_m^{(l)} = \sum_{m_1=-l_1}^{l_1} \sum_{m_2=-l_2}^{l_2} Q_{(l_1,m_1),(l_2,m_2)}^{(l,m)} v_{m_1}^{(l_1)} v_{m_2}^{(l_2)}, \tag{11}$$

where $Q_{(l_1,m_1),(l_2,m_2)}^{(l,m)}$ is called the *Clebsch–Gordan coefficients*. To simplify writing, when specifying the degree of the input and output, we directly abbreviate the above formula to $\tilde{\boldsymbol{v}}^{(l)} = \tilde{\boldsymbol{v}}^{(l_1)} \otimes_{\mathrm{cg}} \tilde{\boldsymbol{v}}^{(l_2)}$. We introduce the decomposition of the moment of inertia in Example A.4 to enhance understanding.

It is straightforward to extend the tensor product to incorporate inversion. When performing the tensor product of two steerable features—one exhibiting odd parity and the other exhibiting even parity—the resulting feature will necessarily exhibit odd parity. Conversely, if both features share the same parity, the result will be of even parity. For simplicity, we will omit discussions related to inversion in the subsequent analyses.

**Example A.4** (Decomposition of Reducible Representations)**.** For the moment of inertia $\mathrm{vec}(\overleftrightarrow{\boldsymbol{I}}) = \sum_i (\|\vec{\boldsymbol{x}}_i\|^2 \cdot \boldsymbol{1}_9 - \vec{\boldsymbol{x}}_i \otimes \vec{\boldsymbol{x}}_i) \in \mathbb{R}^9$. The first item is a scalar (0e) of 9 channels denoted as 9x0e, while the corresponding group representation of the second item $\sum_i \vec{\boldsymbol{x}}_i \otimes \vec{\boldsymbol{x}}_i$ is $\boldsymbol{R}_\mathfrak{r} \otimes \boldsymbol{R}_\mathfrak{r}$ (the inversion are canceled). According to the Clebsch–Gordan rule, this representation is similar to $\bigoplus_{l=0}^2 \boldsymbol{D}^{(l)}(\mathfrak{r})$, which can be transformed through a specific coefficient matrix, and $\sum_i \vec{\boldsymbol{x}}_i \otimes \vec{\boldsymbol{x}}_i$ can be decomposed into irreducible features of three types: 0e, 1e, 2e.[4]

### A.1.3 Discussion on Symmetric Graphs

A symmetric graph is a special kind of geometric graph. [49] points out that in a symmetric graph, steerable features of a particular degree will degenerate, so we list this case separately for analysis.

**Definition A.5** (Symmetric Graph)**.** *A geometric graph $\mathcal{G}$ is called a symmetric graph, if there exists a finite and nontrivial subgroup $\mathfrak{H} \leq \mathrm{O}(3), \mathfrak{H} \neq \{\mathfrak{e}\}$, satisfying that $\forall \mathfrak{h} \in \mathfrak{H}, \mathfrak{h} \cdot \mathcal{G} = \mathcal{G}$. All subgroups making $\mathcal{G}$ symmetric yields a set $\mathbb{H}(\mathcal{G})$, and all geometric graphs that are symmetric w.r.t. $\mathfrak{H}$ constitute a set denoted as $\mathbb{G}(\mathfrak{H})$. Moreover, all other graphs are referred to as* asymmetric *(geometric) graphs.*

It is straightforward to conclude that an asymmetric geometric graph must contain non-coplanar nodes; otherwise, if all the nodes lie in the same plane, the graph would coincide with itself after reflection across that plane. Therefore, the presence of non-coplanar nodes is a necessary condition for the asymmetry of the geometric graph.

**Lemma A.6** (Image Space of Functions on Symmetric Geometric Graphs)**.** *Given a symmetric graph $\mathcal{G}$, $\tilde{\boldsymbol{f}}^{(l)}(\mathcal{G}) \in \bigcap_{\mathfrak{H} \in \mathbb{H}(\mathcal{G})} [\boldsymbol{I}_{2l+1} - \rho^{(l)}(\mathfrak{H})]^\perp$, where $\rho^{(l)}(\mathfrak{H}) = \frac{1}{|\mathfrak{H}|} \sum_{\mathfrak{h} \in \mathfrak{H}} \rho^{(l)}(\mathfrak{h})$ denotes the group averaging.*

Here, we strengthen the conclusion of [49]. Below, we give an example that makes the steerable features confined to a subspace instead of degenerating into a zero vector.

**Example A.7.** Consider a cone with a square base, defined by the set $\mathcal{G} = \{(\pm 1, 0, -1), (0, \pm 1, -1), (0, 0, 4)\}$. For $l = 1$, it is straightforward to find that $\mathbb{F}^{(1)}(\mathcal{G}) = \{(0, 0, z)\} \subset \mathbb{R}^3$, and constructing a $\mathbb{F}^{(1)}(\mathcal{G})$-full-rank function is relatively easy. However, for cases where $l > 1$, while the analysis can still be performed using the orthogonal complement of the group average, constructing a $\mathbb{F}^{(l)}(\mathcal{G})$-full-rank function becomes significantly more challenging. We propose using tensor-product-based models, such as TFN [33], to construct a basis set. These models, despite their computational complexity, have been proven to be complete. Once constructed, this basis set can be applied with our method to achieve the desired result.

---

[4]Note that since it is a symmetric tensor, the components of 1e are actually zero.

## A.2 Connection between graph-level and subgraph-level functions

In practice, the focus of many problems lies in some local features, such as node-level or edge-level features, which we collectively refer to as subgraph-level. In this section, we will show the connection between graph-level functions and subgraph-level functions.

In fact, the local features also require the message from the whole geometric graph $\mathcal{G}$, thus for a subgraph $\mathcal{G}_{\text{sub}} \subset \mathcal{G}$, the $l$th-degree steerable feature could be denoted as $\tilde{\boldsymbol{f}}^{(l)}(\mathcal{G}_{\text{sub}}, \mathcal{G})$. Moreover, we denote $\mathbb{F}^{(l)}(\mathcal{G}_{\text{sub}}, \mathcal{G}) := \{\tilde{\boldsymbol{f}}^{(l)}(\mathcal{G}_{\text{sub}}, \mathcal{G}) | \tilde{\boldsymbol{f}}^{(l)} \in \mathbb{F}^{(l)}\} \subset \mathbb{R}^{2l+1}$. Note that the whole graph is also a subgraph, $\tilde{\boldsymbol{f}}^{(l)}(\mathcal{G})$ and $\mathbb{F}^{(l)}(\mathcal{G})$ are actually the simplified version of $\tilde{\boldsymbol{f}}^{(l)}(\mathcal{G}, \mathcal{G})$ and $\mathbb{F}^{(l)}(\mathcal{G}, \mathcal{G})$.

**Lemma A.8.** *Given a geometric graph $\mathcal{G}$, where $\mathcal{G}_{sub} \subset \mathcal{G}$ is a subgraph (e.g. node, edge), then we have $\mathbb{F}^{(l)}(\mathcal{G}) \subset \mathbb{F}^{(l)}(\mathcal{G}_{sub}, \mathcal{G})$.*

*Proof.* Note that for the subgraph-level function $\tilde{\boldsymbol{f}}^{(l)}(\mathcal{G}_{\text{sub}}, \mathcal{G})$, let the function ignore the previous input $\mathcal{G}_{\text{sub}}$. Therefore, for any graph-level function, there is always a subgraph-level function that is equal to it. $\qquad\square$

**Lemma A.9.** *Given a geometric graph $\mathcal{G}$, where $\mathcal{G}_{sub} \subset \mathcal{G}$ is a subgraph (e.g. node, edge). We have $\mathbb{F}(\mathcal{G}_{sub}, \mathcal{G}) = \mathbb{R}^{2l+1}$, if $\mathbb{F}(\mathcal{G}) = \mathbb{R}^{2l+1}$.*

*Proof.* It is obvious that $\mathbb{F}(\mathcal{G}_{\text{sub}}, \mathcal{G}) \subset \mathbb{R}^{2l+1}$. And we have $\mathbb{R}^{2l+1} = \mathbb{F}(\mathcal{G}) \subset \mathbb{F}(\mathcal{G}_{\text{sub}}, \mathcal{G})$, thus $\mathbb{F}(\mathcal{G}_{\text{sub}}, \mathcal{G}) = \mathbb{F}(\mathcal{G}) = \mathbb{R}^{2l+1}$. $\qquad\square$

**Proposition A.10.** *Given a geometric graph $\mathcal{G}$, where $\mathcal{G}_{sub} \subset \mathcal{G}$ is a subgraph (e.g. node, edge), suppose there is a matrix $\tilde{\boldsymbol{V}}^{(l)}(\mathcal{G}_{sub}, \mathcal{G})$ with $C$ channels of $l$th-degree steerable features denoted as $\tilde{\boldsymbol{v}}_c^{(l)}(\mathcal{G}_{sub}, \mathcal{G})$ satisfying $\operatorname{span} \tilde{\boldsymbol{V}}^{(l)}(\mathcal{G}_{sub}, \mathcal{G}) = \mathbb{F}^{(l)}(\mathcal{G}_{sub}, \mathcal{G})$. Then for any $l$th-degree steerable function $\tilde{\boldsymbol{t}}^{(l)} \in \mathbb{F}^{(l)}$, there exists $\tilde{\boldsymbol{w}}^{(0)}(\mathcal{G}_{sub}, \mathcal{G}) := [\tilde{w}_c^{(0)}(\mathcal{G}_{sub}, \mathcal{G})]_{c=1}^C$ with $C$-channel output scalars, such that*

$$\tilde{\boldsymbol{t}}^{(l)}(\mathcal{G}_{sub}, \mathcal{G}) = \textstyle\sum_{c=1}^C \tilde{w}_c^{(0)}(\mathcal{G}_{sub}, \mathcal{G})\tilde{\boldsymbol{v}}_c^{(l)}(\mathcal{G}_{sub}, \mathcal{G}). \tag{12}$$

*Proof.* Note that the proof of Theorem 3.2 does not depend on the input form of the function, so the generalization from Theorem 3.2 to this proposition still holds. $\qquad\square$

**Proposition A.11.** *Given a geometric graph $\mathcal{G}$, where $\mathcal{G}_{sub} \subset \mathcal{G}$ is a subgraph (e.g. node, edge) and $\mathbb{F}(\mathcal{G}_{sub}, \mathcal{G}) = \mathbb{F}(\mathcal{G})$. Suppose there is a matrix $\tilde{\boldsymbol{V}}^{(l)}(\mathcal{G})$ with $C$ channels of $l$th-degree steerable features denoted as $\tilde{\boldsymbol{v}}_c^{(l)}(\mathcal{G})$ satisfying $\operatorname{span} \tilde{\boldsymbol{V}}^{(l)}(\mathcal{G}) = \mathbb{F}^{(l)}(\mathcal{G})$. Then for any $l$th-degree steerable function $\tilde{\boldsymbol{t}}^{(l)} \in \mathbb{F}^{(l)}$, there exists $\tilde{\boldsymbol{w}}^{(0)}(\mathcal{G}_{sub}, \mathcal{G}) := [\tilde{w}_c^{(0)}(\mathcal{G}_{sub}, \mathcal{G})]_{c=1}^C$ with $C$-channel output scalars, such that*

$$\tilde{\boldsymbol{t}}^{(l)}(\mathcal{G}_{sub}, \mathcal{G}) = \textstyle\sum_{c=1}^C \tilde{w}_c^{(0)}(\mathcal{G}_{sub}, \mathcal{G})\tilde{\boldsymbol{v}}_c^{(l)}(\mathcal{G}). \tag{13}$$

*Proof.* Note that $\tilde{\boldsymbol{v}}_c^{(l)}(\mathcal{G}) \in \mathbb{F}^{(l)}(\mathcal{G}) \subset \mathbb{F}^{(l)}(\mathcal{G}_{\text{sub}}, \mathcal{G})$, so this proposition still holds. $\qquad\square$

Note that for invariant functions, if we can obtain the canonical form at the graph-level, then we can naturally obtain the canonical form at the subgraph-level. Therefore, the discussion of subgraph-level and graph-level is consistent. To simplify the writing, only the graph-level case will be discussed in the following.

## A.3 The full form of dynamic method

Before giving a complete theoretical analysis, let us review Theorem 3.2 as follows.

**Theorem 3.2** (Dynamic Method). *Given a geometric graph $\mathcal{G}$, suppose there is a matrix $\tilde{\boldsymbol{V}}^{(l)}(\mathcal{G})$ with $C$ channels of $l$th-degree steerable features denoted as $\tilde{\boldsymbol{v}}_c^{(l)}(\mathcal{G})$ satisfying $\operatorname{span}(\tilde{\boldsymbol{V}}^{(l)}(\mathcal{G})) = \mathbb{F}^{(l)}(\mathcal{G}) := \{\tilde{\boldsymbol{f}}^{(l)}(\mathcal{G}) \mid \tilde{\boldsymbol{f}}^{(l)} \in \mathbb{F}^{(l)}\} \subset \mathbb{R}^{2l+1}$. Then for any $l$th-degree steerable function $\tilde{\phi}^{(l)} \in \mathbb{F}^{(l)}$, there always exists $\tilde{\boldsymbol{w}}^{(0)}(\mathcal{G}) := [\tilde{w}_c^{(0)}(\mathcal{G})]_{c=1}^C$ with $C$-channel output scalars, such that*

$$\tilde{\phi}^{(l)}(\mathcal{G}) = \textstyle\sum_{c=1}^C \tilde{w}_c^{(0)}(\mathcal{G})\tilde{\boldsymbol{v}}_c^{(l)}(\mathcal{G}). \tag{4}$$

This form decomposes the problem of constructing a complete equivariant function into two components. However, determining the explicit forms of $\tilde{\boldsymbol{w}}^{(0)}(\mathcal{G})$ and $\tilde{\boldsymbol{V}}^{(l)}$ remains a highly nontrivial task. To provide insight into this construction, we present two corollaries of Theorem 3.2, corresponding to two special cases of the equivariant space $\mathbb{F}^{(l)}(\mathcal{G})$: one where it reduces to the trivial set consisting solely of the origin, $\theta := \{\boldsymbol{0}_{2l+1}\}$, and the other is the opposite, *i.e.* $\dim \mathbb{F}^{(l)}(\mathcal{G}) > 0$.

**Proposition A.12.** *Given a geometric graph $\mathcal{G}$ with $\mathbb{F}^{(l)}(\mathcal{G}) = \theta$, let $\mathbb{W}^{(0)} \subset \mathbb{F}^{(0)}$ denote all invariant functions that can be expressed by model, any lth-degree steerable function could be written into the form of Theorem 3.2, if $\mathbb{W}^{(0)}$ is not empty.*

*Proof.* Since $\mathbb{F}^{(l)}(\mathcal{G}) = \theta$, any equivariant output of degree $l$ must be the zero vector. In particular, both the target output $\tilde{\boldsymbol{t}}^{(l)}(\mathcal{G})$ and each basis component $\tilde{\boldsymbol{v}}_c^{(l)}(\mathcal{G})$ in the decomposition of Theorem 3.2 are identically zero.

As a result, the decomposition in Theorem 3.2 holds trivially for any choice of invariant scalar function $\tilde{\boldsymbol{w}}^{(0)}(\mathcal{G}) \in \mathbb{W}^{(0)}$, because both sides of the equation reduce to the zero vector. Hence, any $l$th-degree steerable function admits the desired representation, as long as $\mathbb{W}^{(0)}$ is non-empty. $\square$

**Proposition A.13.** *Given a geometric graph $\mathcal{G}$ with $\dim \mathbb{F}^{(l)}(\mathcal{G}) > 0$, let $\mathbb{W}^{(0)} \subset \mathbb{F}^{(0)}$ denote all invariant functions that can be expressed by model, any lth-degree steerable function could be written into the form of Theorem 3.2, iff $\mathbb{W}^{(0)} = \mathbb{F}^{(0)}$, i.e. the invariant model is complete.*

*Proof.* The sufficiency is evident. By Theorem 3.2, there exists an invariant function of the required form, and such a function is clearly contained in the space $\mathbb{F}^{(0)}$.

To prove necessity, assume by contradiction that $\mathbb{W}^{(0)} \subsetneq \mathbb{F}^{(0)}$ but that the decomposition in Theorem 3.2 still holds for all steerable functions. Let $\tilde{\boldsymbol{V}}^{(l)}(\mathcal{G})$ be a set of linearly independent equivariant vectors that spans $\mathbb{F}^{(l)}(\mathcal{G})$, *i.e.*, a basis of $\mathbb{F}^{(l)}(\mathcal{G})$. Now, consider any $\tilde{w}^{(0)} \in \mathbb{F}^{(0)} \setminus \mathbb{W}^{(0)}$. Then the function $\tilde{\boldsymbol{t}}^{(l)}(\mathcal{G}) = \sum_c \tilde{w}^{(0)}(\mathcal{G})\tilde{\boldsymbol{v}}_c^{(l)}(\mathcal{G})$ is a valid steerable function in $\mathbb{F}^{(l)}(\mathcal{G})$, as it results from a linear combination of basis elements with coefficients in $\mathbb{F}^{(0)}$. However, since $\tilde{w}^{(0)}$ is not expressible by the model, the decomposition cannot be achieved using elements in $\mathbb{W}^{(0)}$, contradicting the assumption that the decomposition is always possible.

Therefore, the assumption must be false, and we conclude that $\mathbb{W}^{(0)} = \mathbb{F}^{(0)}$ is necessary for the representation to hold for all steerable functions. $\square$

### A.4 The construction of canonical form

In order to losslessly encode the information of a geometric graph, we define the concept of canonical forms in this section. Then, we give a method to construct canonical forms in polynomial time. Here, we assume that the geometric graph is fully connected and use a double-loop set comparison method to estimate the worst time complexity of the algorithm.

#### A.4.1 An algorithm for determining point cloud isomorphism

Here, we give Algo. 1 for determining whether two geometric point clouds are isomorphic and prove its correctness. It is noteworthy that this algorithm resembles that of [104], centered on the principle of four-point positioning in three-dimensional space. Our subsequent designs will leverage this principle to adapt it for use in geometric graph scenarios and neural network applications.

For two given point clouds, represented as $\vec{\boldsymbol{X}}^{(\mathcal{G})}$ and $\vec{\boldsymbol{X}}^{(\mathcal{H})}$, we recognize that if these point clouds are isomorphic, there must exist a local substructure (composed of at least four points, *i.e.* $\vec{\boldsymbol{U}}_\alpha$ and $\vec{\boldsymbol{V}}_\beta$) that is also isomorphic. It is worth noting that this is the isomorphism judgment between two point clouds of a certain size, so it is always of constant complexity.

According to the four-point positioning principle, the coordinates of any point $\vec{\boldsymbol{u}}_i \in \vec{\boldsymbol{X}}^{(\mathcal{G})}$ or $\vec{\boldsymbol{v}}_i \in \vec{\boldsymbol{X}}^{(\mathcal{H})}$ can be expressed as distance vectors, *i.e.* $\boldsymbol{d}_i^{(\mathcal{G})}$ or $\boldsymbol{d}_i^{(\mathcal{H})}$, relative to these four foundational points. Consequently, sets of coordinates $\vec{\boldsymbol{X}}^{(\mathcal{G})}, \vec{\boldsymbol{X}}^{(\mathcal{H})}$ can be transformed into sets of distance vectors $\mathbb{D}^{(\mathcal{G})}, \mathbb{D}^{(\mathcal{H})}$. When there is a set of isomorphic substructures with matching distance vector

---

**Algorithm 1:** Check *point clouds isomorphic* with time complexity $\mathcal{O}(N^6)$.

---

**Data:** Two point clouds $\vec{X}^{(\mathcal{G})}$ and $\vec{X}^{(\mathcal{H})}$, each containing $N$ nodes.
**Result:** *True* or *False*, indicating whether the two point clouds are isomorphic.
`// O(N⁴), the worst case requires traversing all permutations.`

1 Find an ordered tuple containing four non-coplanar points $\vec{U}_{\boldsymbol{\alpha}} \leftarrow (\vec{u}^{(\mathcal{G})}_{\alpha_i})^4_{i=1}$ in $\mathcal{G}$;
    `// O(N⁴), the permutations of 4 elements from a set of size N.`

2 **for** *all ordered subsets of $\vec{X}^{(\mathcal{H})}$ containing four elements donated by $\vec{V}_{\boldsymbol{\beta}} \leftarrow (\vec{v}^{(\mathcal{H})}_{\beta_i})^4_{i=1}$* **do**
      `// O(1), comparison of two ordered sets of a certain size.`

3      **if** $\vec{U}_{\boldsymbol{\alpha}}$ *and* $\vec{V}_{\boldsymbol{\beta}}$ *is not point clouds isomorphic according to matching* $(\vec{u}^{(\mathcal{G})}_{\alpha_i}, \vec{v}^{(\mathcal{H})}_{\beta_i})$ **then**

4          **continue**;

5      **end**

6      **for** $\vec{u}_i \in \vec{X}^{(\mathcal{G})}$ **do**
        `// O(N), get a 4-channel scalar vector.`

7          $\boldsymbol{d}^{(\mathcal{G})}_i \leftarrow (\|\vec{u}_i - \vec{u}_{\alpha_1}\|, \|\vec{u}_i - \vec{u}_{\alpha_2}\|, \|\vec{u}_i - \vec{u}_{\alpha_3}\|, \|\vec{u}_i - \vec{u}_{\alpha_4}\|)$;

8      **end**

9      **for** $\vec{v}_i \in \vec{X}^{(\mathcal{H})}$ **do**
        `// O(N), get a 4-channel scalar vector.`

10         $\boldsymbol{d}^{(\mathcal{H})}_i \leftarrow (\|\vec{v}_i - \vec{v}_{\beta_1}\|, \|\vec{v}_i - \vec{v}_{\beta_2}\|, \|\vec{v}_i - \vec{v}_{\beta_3}\|, \|\vec{v}_i - \vec{v}_{\beta_4}\|)$;

11      **end**
      `// O(N), change the node set into scalar set.`

12      $\mathbb{D}^{(\mathcal{G})}, \mathbb{D}^{(\mathcal{H})} \leftarrow \texttt{set}([\boldsymbol{d}^{(\mathcal{G})}_i]^N_{i=1}), \texttt{set}([\boldsymbol{d}^{(\mathcal{H})}_i]^N_{i=1})$;
      `// O(N²), the complexity of comparing two sets varies depending on`
          `the choice, such as double loop comparison.`

13      **if** $\mathbb{D}^{(\mathcal{G})} = \mathbb{D}^{(\mathcal{H})}$ **then**

14         **return** True;

15      **end**

16 **end**

17 **return** False;

---

sets, the point clouds are necessarily isomorphic. Conversely, if no isomorphic substructure can be identified or if the distance vector sets cannot be rendered equivalent for any potential isomorphic substructure, then the point clouds must not be isomorphic.

### A.4.2 An algorithm for determining geometric graph isomorphism

Furthermore, we discuss how to design an algorithm to distinguish the isomorphism of two geometric graphs. Note that at this point, all node coordinates have been converted to distance vectors, so it is also easy to transform the edge set as follows.

### A.4.3 A general canonical form

Now, we further give a method to construct the canonical form of a geometric graph. From Algos. 1 and 2, it is not difficult to see that the core of the algorithm is to transform a set of 3D coordinates into a set of distance vectors. Here we give Algo. 3 as follows.

To encode the set, we employ DeepSet [65] for encoding, which losslessly encodes set data. Here, we ignore the complexity of the neural network (because this is a function of the hidden layer dimension $H$), and only consider the complexity of the set size. It is not difficult to see from Eq. (14) that for a set of size $M$, the complexity of DeepSet is $\mathcal{O}(M)$.

**Lemma A.14** (DeepSet). *A set function could be always written into such form:*

$$f(\{x_i\}^N_{i=1}) = \rho(\textstyle\sum^N_{i=1} \phi(x_i)). \tag{14}$$

---
**Algorithm 2:** Check *geometric graph isomorphism* with time complexity $\mathcal{O}(N^8)$.
---

   **Data:** Two geometric graphs $\mathcal{G}$ and $\mathcal{H}$, each containing $N$ nodes.

   **Result:** *True* or *False*, indicating whether the two point clouds are isomorphic.

   `// ` $\mathcal{O}(N^4)$`, the worst case requires traversing all permutations.`

**1** Find an ordered tuple containing four non-coplanar points $\vec{U}_{\boldsymbol{\alpha}} \leftarrow (\vec{u}_{\alpha_i}^{(\mathcal{G})})_{i=1}^4$ in $\mathcal{G}$;

   `// ` $\mathcal{O}(N^4)$`, the permutations of 4 elements from a set of size N.`

**2** **for** *all ordered subsets of* $\vec{X}^{(\mathcal{H})}$ *containing four elements donated by* $\vec{V}_{\boldsymbol{\beta}} \leftarrow (\vec{v}_{\beta_i}^{(\mathcal{H})})_{i=1}^4$ **do**

      `// ` $\mathcal{O}(1)$`, comparison of two ordered sets of a certain size.`

**3**    **if** $\vec{U}_{\boldsymbol{\alpha}}$ *and* $\vec{V}_{\boldsymbol{\beta}}$ *is not point clouds isomorphic according to matching* $(\vec{u}_{\alpha_i}^{(\mathcal{G})}, \vec{v}_{\beta_i}^{(\mathcal{H})})$ **then**

**4**        | **continue**;

**5**    **end**

**6**    **for** $\vec{u}_i \in \vec{X}^{(\mathcal{G})}$ **do**

        | `// ` $\mathcal{O}(N)$`, get a 4-channel scalar vector.`

**7**        | $\boldsymbol{d}_i^{(\mathcal{G})} \leftarrow (\|\vec{u}_i - \vec{u}_{\alpha_1}\|, \|\vec{u}_i - \vec{u}_{\alpha_2}\|, \|\vec{u}_i - \vec{u}_{\alpha_3}\|, \|\vec{u}_i - \vec{u}_{\alpha_4}\|)$;

**8**    **end**

**9**    **for** $\vec{v}_i \in \vec{X}^{(\mathcal{H})}$ **do**

        | `// ` $\mathcal{O}(N)$`, get a 4-channel scalar vector.`

**10**     | $\boldsymbol{d}_i^{(\mathcal{H})} \leftarrow (\|\vec{v}_i - \vec{v}_{\beta_1}\|, \|\vec{v}_i - \vec{v}_{\beta_2}\|, \|\vec{v}_i - \vec{v}_{\beta_3}\|, \|\vec{v}_i - \vec{v}_{\beta_4}\|)$;

**11**    **end**

     `// ` $\mathcal{O}(N + N^2)$`, change the node set and edge set into scalar set.`

**12**    $\mathbb{D}^{(\mathcal{G})}, \mathbb{D}^{(\mathcal{H})} \leftarrow \mathtt{set}([\boldsymbol{d}_i^{(\mathcal{G})}]_{i=1}^N), \mathtt{set}([\boldsymbol{d}_i^{(\mathcal{H})}]_{i=1}^N)$;

**13**    $\mathbb{E}^{(\mathcal{G})}, \mathbb{E}^{(\mathcal{H})} \leftarrow \mathtt{set}([\boldsymbol{d}_i^{(\mathcal{G})}, \boldsymbol{d}_j^{(\mathcal{G})}, \boldsymbol{e}_{ij}^{(\mathcal{G})}]_{\langle i,j \rangle \in \mathcal{E}^{(\mathcal{G})}}), \mathtt{set}([\boldsymbol{d}_i^{(\mathcal{H})}, \boldsymbol{d}_j^{(\mathcal{H})}, \boldsymbol{e}_{ij}^{(\mathcal{H})}]_{\langle i,j \rangle \in \mathcal{E}^{(\mathcal{H})}})$;

     `// ` $\mathcal{O}(N^2 + N^4)$`, the complexity of comparing four sets (node-node,`
        `edge-edge) varies depending on the choice, such as double loop`
        `comparison.`

**14**    **if** $\mathbb{D}^{(\mathcal{G})} = \mathbb{D}^{(\mathcal{H})}$ *and* $\mathbb{E}^{(\mathcal{G})} = \mathbb{E}^{(\mathcal{H})}$ **then**

**15**     | **return** True;

**16**    **end**

**17** **end**

**18** **return** False;

---

*Proof.* This proof is a simplification of Theorem 7 in Deepset [65]. The necessity, that is, that this form satisfies the permutation invariance of sets is obvious, and its sufficiency is proved below. Consider an $N$th degree polynomial $P(t) = \sum_{i=0}^N a_i t^i$, which satisfies the set $\{x_i\}_{i=1}^N$ of all the $N$ roots of $P(t) = 0$. Thus, the set can be mapped to ordered polynomial coefficients. Vieta's theorem and Newton's identity show that the space of polynomial coefficients is consistent with the space of sum-of-power functions, so the universal approximation theorem of MLP can be used to prove that such a mapping can be found. □

**Theorem 3.5** (General Canonical Form)**.** *Given any geometric graph* $\mathcal{G}$*, Algo. 3 provides a method to create canonical form with time complexity* $\mathcal{O}(N^6)$*.*

*Proof.* Due to the completeness of Deepset's encoding, Algo. 3 can naturally distinguish arbitrary geometric graphs. For a detailed time complexity analysis, see the comments in the algorithm block. □

### A.4.4 A faster method to construct canonical form

In practical applications, it is impractical to traverse all ordered four-point sets because the complexity is too high. In order to reduce the complexity, a simple and intuitive way is to reduce the number of four-point sets. Here, a very clever change of mind is that we change the ordered four-point set from being selected from a geometric graph to being generated from a geometric graph. We call these generated ordered four-point sets virtual nodes. In order to ensure the correctness of the algorithm, we

---

**Algorithm 3:** A canonical form of geometric graphs.

**Data:** A geometric graph $\mathcal{G}$, and $\psi_{\text{node}}, \psi_{\text{edge}}, \psi_{\text{graph}}$ are DeepSet models.
**Result:** The canonical form $\Gamma \in \mathbb{R}^H$ of point clouds $\mathcal{G}$.
// $\mathcal{O}(N^4)$, traverse all permutations.

1   $\mathbb{T} \leftarrow \varnothing$;
2   **for** *any ordered set containing four non-coplanar points* $\vec{U}_{\boldsymbol{\alpha}} \leftarrow \{\vec{u}_{\alpha_i}\}_{i=1}^4$ *in* $\mathcal{G}$ **do**
3     **for** $\vec{u}_i \in \vec{X}^{(\mathcal{G})}$ **do**
      // $\mathcal{O}(N)$, get a 4-channel scalar vector.
4      $\boldsymbol{d}_i \leftarrow (\|\vec{u}_i - \vec{u}_{\alpha_1}\|, \|\vec{u}_i - \vec{u}_{\alpha_2}\|, \|\vec{u}_i - \vec{u}_{\alpha_3}\|, \|\vec{u}_i - \vec{u}_{\alpha_4}\|)$;
5     **end**
     // $\mathcal{O}(N + N^2)$, convert point sets and edge sets into scalar sets.
6     $\mathbb{D} \leftarrow \texttt{set}([\boldsymbol{d}_i]_{i=1}^N)$, $\mathbb{E} \leftarrow \texttt{set}([\boldsymbol{d}_i, \boldsymbol{d}_j, \boldsymbol{e}_{ij}]_{\langle i,j \rangle \in \mathcal{E}})$;
     // $\mathcal{O}(1)$, decentralization of the four reference points.
7     $\vec{U}_{\boldsymbol{\alpha}} \leftarrow (\boldsymbol{I}_{4\times 4} - \frac{1}{4}\boldsymbol{1}_{4\times 4})\vec{U}_{\boldsymbol{\alpha}}$;
     // $\mathcal{O}(N + N^2)$, get the embedding based on current four points.
8     $\Gamma_{\boldsymbol{\alpha}} \leftarrow \texttt{concat}(\vec{U}_{\boldsymbol{\alpha}}^\top \vec{U}_{\boldsymbol{\alpha}}, \psi_{\text{node}}(\mathbb{D}), \psi_{\text{edge}}(\mathbb{E}))$;
9     $\mathbb{T} \leftarrow \mathbb{T} \cup \{\Gamma_{\boldsymbol{\alpha}}\}$;
10 **end**
11 $\Gamma \longleftarrow \psi_{\text{graph}}(\mathbb{T})$;
12 **return** $\Gamma$;

---

need to use a E(3)-equivariant method to implement the generation process. We call this generating function $\zeta$, thereby simplifying Algo. 3 to Algo. 4.

---

**Algorithm 4:** A faster method to construct canonical form.

**Data:** A geometric graph $\mathcal{G}$, and $\psi_{\text{node}}, \psi_{\text{edge}}$ are DeepSet models.
**Result:** The canonical form $\Gamma \in \mathbb{R}^H$ of point clouds $\mathcal{G}$.
// Get four non-coplanar reference points via generation.

1   $\vec{V} \leftarrow \zeta(\mathcal{G})$;
2   **for** $\vec{u}_i \in \vec{X}^{(\mathcal{G})}$ **do**
    // $\mathcal{O}(N)$, get a 4-channel scalar vector.
3     $\boldsymbol{d}_i \leftarrow (\|\vec{u}_i - \vec{v}_1\|, \|\vec{u}_i - \vec{v}_2\|, \|\vec{u}_i - \vec{v}_3\|, \|\vec{u}_i - \vec{v}_4\|)$;
4   **end**
    // $\mathcal{O}(N + N^2)$, convert point sets and edge sets into scalar sets.
5   $\mathbb{D} \leftarrow \texttt{set}([\boldsymbol{d}_i]_{i=1}^N)$, $\mathbb{E} \leftarrow \texttt{set}([\boldsymbol{d}_i, \boldsymbol{d}_j, \boldsymbol{e}_{ij}]_{\langle i,j \rangle \in \mathcal{E}})$;
    // $\mathcal{O}(1)$, decentralization of the four reference points.
6   $\vec{V} \leftarrow (\boldsymbol{I}_{4\times 4} - \frac{1}{4}\boldsymbol{1}_{4\times 4})\vec{V}$;
    // $\mathcal{O}(N + N^2)$, get the embedding based on current four points.
7   $\Gamma \leftarrow \texttt{concat}(\vec{V}^\top \vec{V}, \psi_{\text{node}}(\mathbb{D}), \psi_{\text{edge}}(\mathbb{E}))$;
8   **return** $\Gamma$;

---

It is worth noting that Algo. 4 is applicable only when the four virtual nodes generated by $\zeta$ are not coplanar. We summarize it as:

**Theorem 3.6** (Faster Canonical Form). *Given an* E(3)-*equivariant function* $\zeta$ *that generates four non-coplanar points on* $\mathcal{G}$, *Algo. 4 is able to create canonical form with time complexity* $\mathcal{O}(N^2)$.

*Proof.* Let $\Gamma$ be the function of Algo. 4, we only need to prove that $\mathcal{G} \cong \mathcal{H} \iff \Gamma(\mathcal{G}) = \Gamma(\mathcal{H})$. We first prove sufficiency, let $\mathcal{G} = \mathfrak{g} \cdot \mathcal{H}$ with $\mathfrak{g} = (\boldsymbol{O}, \vec{t}) \in$ E(3) be a group element. Since $\zeta$ is an E(3)-equivariant function, we have $\vec{V}^{(\mathcal{G})} = \zeta(\mathcal{G}) = \zeta(\mathfrak{g} \cdot \mathcal{H}) = \boldsymbol{O}\vec{V}^{(\mathcal{G})} + \vec{t} = \vec{V}^{(\mathcal{H})}$. After decentralization, we have $\boldsymbol{O}\vec{V}^{(\mathcal{G})} = \vec{V}^{(\mathcal{H})} \implies (\vec{V}^{(\mathcal{G})})^\top \vec{V}^{(\mathcal{G})} = (\vec{V}^{(\mathcal{H})})^\top \vec{V}^{(\mathcal{H})}$. In addition, the encoding of nodes and edges depends on the relative distance to the virtual node and is therefore

invariant, *i.e.* $\mathbb{D}^{(\mathcal{G})} = \mathbb{D}^{(\mathcal{H})}, \mathbb{E}^{(\mathcal{G})} = \mathbb{E}^{(\mathcal{H})}$. Thus, we get $\Gamma(\mathcal{G}) = \Gamma(\mathcal{H})$. Next, we prove the necessity. Note that when we take the reference four points as $\zeta(\mathcal{G})$ and $\zeta(\mathcal{H})$, we can directly get $\mathcal{G} \cong \mathcal{H}$ in Algo. 2, thus completing the proof. $\qquad\square$

Table 8: Comparison of time complexity.

| | General Method (Algo. 3) | | Faster Method | |
| | Algorithm Details | Time Complexity | Algorithm Details | Time Complexity |
|---|---|---|---|---|
| Possible chosen of VNs | Traverse all possible quadruplets ($\binom{N}{4}$ types). | $\mathcal{O}(N^4)$ | Generate directly. | $\mathcal{O}(1)$ |
| Construct VNs | Use real nodes as virtual nodes. | $\mathcal{O}(1)$ | Aggregate by all nodes. | $\mathcal{O}(N)$ |
| Get distance vector | Calculate $N$ nodes to possible VNs. | $\mathcal{O}(N^4 \cdot N)$ | Calculate $N$ nodes to possible VNs. | $\mathcal{O}(1 \cdot N)$ |
| Set of distance vector | Each chosen of VN prodive a set. | $\mathcal{O}(N^4)$ | Each chosen of VN prodive a set. | $\mathcal{O}(1)$ |
| Embed set with Deepset | $N$ for node and $N^2$ for edge | $\mathcal{O}(N^4 \cdot (N + N^2))$ | $N$ for node and $N^2$ for edge | $\mathcal{O}(1 \cdot (N + N^2))$ |
| Total | | $\mathcal{O}(N^6)$ | | $\mathcal{O}(N^2)$ |

## A.5 Construction of full-rank basis set

In this section, we discuss the full-rank basis set, including its existence and construction method on asymmetric graphs.

**Theorem 3.7** (Coloring on Asymmetric Graph). *In an asymmetric geometric graph, each point can be assigned a unique color. This implies the existence of an* $\mathrm{E}(3)$ *invariant function that maps the features of the $i$th node to distinct values $\boldsymbol{h}_i$. Similarly, for directed edges $ij$, their features are mapped to distinct values $\boldsymbol{e}_{ij}$.*

*Proof.* The theorem can be described in the following formal language, given an asymmetric point cloud $\vec{\boldsymbol{X}} \in \mathbb{R}^{3 \times N}$, and all function here discussed are node-level scalar function. This theorem means

$$\exists f : \vec{\boldsymbol{X}} \mapsto \boldsymbol{y} \in \mathbb{N}_+^N, \forall i \neq j, \boldsymbol{y}_i \neq \boldsymbol{y}_j. \tag{15}$$

First, we prove that it can be expressed as an equivalent proposition, which is

$$\forall i \neq j, \exists f : \vec{\boldsymbol{X}} \mapsto \boldsymbol{y} \in \mathbb{N}_+^N, \boldsymbol{y}_i \neq \boldsymbol{y}_j. \tag{16}$$

According to the logical relationship, it is obvious that Eq. (15) leads to Eq. (16). We will prove below that Eq. (16) can also lead to Eq. (15). Let $f^{ij} : \vec{\boldsymbol{X}} \mapsto \boldsymbol{y}^{ij}$ denote a function let $\boldsymbol{y}_i^{ij} \neq \boldsymbol{y}_j^{ij}$, then consider map $g : \vec{\boldsymbol{X}} \mapsto \boldsymbol{Y} := \oplus_{i,j} \boldsymbol{y}^{ij}$. We can find that $\boldsymbol{Y}_i \neq \boldsymbol{Y}_j$, and notice each $f^{ij}$ only uses finite colors, which means we can further assign a new color to each possible output of $g$, which is the mapping we want in Eq. (16).

Then we prove it by contradiction Eq. (16), assuming that the following proposition is true:

$$\exists i \neq j, \forall f : \vec{\boldsymbol{X}} \mapsto \boldsymbol{y} \in \mathbb{N}_+^N, \boldsymbol{y}_i = \boldsymbol{y}_j. \tag{17}$$

We now construct a contradiction, hoping to prove that Eq. (17) will lead to point cloud symmetry. Let the permutation matrix $\boldsymbol{P} \in S_N$ represent the exchange between node $i$ and node $j$, so we can get:

$$f(\boldsymbol{P}\vec{\boldsymbol{X}}) = \boldsymbol{P}\boldsymbol{y} = \boldsymbol{y} = f(\vec{\boldsymbol{X}}). \tag{18}$$

Since $f$ is an $\mathrm{E}(3)$-invarinat function, we guess $\boldsymbol{P}\vec{\boldsymbol{X}}$ will fall in set $\mathrm{E}(3) \cdot \vec{\boldsymbol{X}}$. And now we prove that $\boldsymbol{P}\vec{\boldsymbol{X}}$ must fall in such set. And the canonical form $\Gamma$ is obvious an $\mathrm{E}(3)$-invarinat function and could distinguish whether two point clouds are isomorphic, which means $\boldsymbol{P}\vec{\boldsymbol{X}}$ must be point cloud isomorphic with $\vec{\boldsymbol{X}}$. And notice that $f$ is a node-level function, which means

$$\vec{\boldsymbol{x}}_i = \mathfrak{g} \cdot \vec{\boldsymbol{x}}_j, \vec{\boldsymbol{x}}_j = \mathfrak{g} \cdot \vec{\boldsymbol{x}}_i.$$

And we have assumed there are no overlapping points, *i.e.* $\vec{\boldsymbol{x}}_i \neq \vec{\boldsymbol{x}}_j$, so $\mathfrak{g} \neq \mathfrak{e}$. And it shows that such a point cloud is symmetric, which constitutes a contradiction and the original proposition is proved. $\qquad\square$

**Theorem 3.8** (Existence of Full-Rank Basis Set). *For any given asymmetric graph $\mathcal{G}$, an $\mathbb{F}^{(l)}(\mathcal{G})$-full-rank $\tilde{\boldsymbol{V}}^{(l)}(\mathcal{G})$ can always be constructed for any degree $l$.*

*Proof.* We give the proof via a constructive method. According to Theorem 3.7, there exists a mapping let all node be colored differently, and without loss of generality, we denote the color of node $i$ as $c_i$. Now we construct function:

$$f_k(\vec{\boldsymbol{X}}) = \sum_{i=1}^{N} \delta_{k,c_i} \cdot \vec{\boldsymbol{x}}_i = \vec{\boldsymbol{x}}_k, \tag{19}$$

It is obviously that such a funtion is E(3)-equvariant, and we and further combine $f_k$ of different $k$, which can construct a full-rank $\mathbb{V}^{(1)}$ since the input point cloud is non-coplanar. $\square$

We just give an existence proof here, which is still affected by the structure of the model itself and the form of message passing. We give an example in Example A.15.

## A.6 Comparison of different coloring methods

Node coloring necessitates updating node features. Here, we propose two alternative solutions:

1. **Distance to Center**: A simple preliminary coloring can be achieved by calculating the distance $\|\vec{\boldsymbol{x}}_i - \vec{\boldsymbol{x}}_c\|$ from each node to the coordinate center. While this method is straightforward, it lacks practical significance and may not perform well in real-world problems. Additionally, there are counterexamples, such as in Example A.15, where nodes cannot be distinguished.

2. **Tensor Product**: This method extends the previous approach. First, we construct a global feature $\sum_{i=1}^{N} Y^{(l)}(\vec{\boldsymbol{x}}_i - \vec{\boldsymbol{x}}_c)$, which is then used to create the coloring through moments. While this method is more complex and incurs higher computational costs, it enhances expressive power.

Table 9: Different Color Methods. For the tensor product method, the set $\mathbb{L} = \{0, \dots, L\}$ denotes all degrees and the learnable weights of tensor product are omitted.

| Nothing ($\varnothing$) | Distance to Center ($\oplus$) | Tensor Product ($\otimes$) |
|---|---|---|
| | Calculation Formula | |
| $\varnothing$ | $\boldsymbol{h}_i \leftarrow \boldsymbol{h}_i + \sigma(\|\vec{\boldsymbol{x}}_{ic}\|)$ | $\tilde{\boldsymbol{h}}_{\text{global}}^{(\mathbb{L})} \leftarrow \sum_{i=1}^{N} \sigma(\|\vec{\boldsymbol{x}}_{ic}\|) \cdot Y^{(\mathbb{L})}(\vec{\boldsymbol{x}}_{ic})$ $\boldsymbol{h}_i \leftarrow \boldsymbol{h}_i + \tilde{\boldsymbol{h}}_{\text{global}}^{(\mathbb{L})} \otimes_{\text{cg}} \cdots \otimes_{\text{cg}} Y^{(\mathbb{L})}(\vec{\boldsymbol{x}}_{ic})$ |
| | Scenarios to Avoid | |
| Sparse Graphs | Counterexample like Fig. 4(c) | Not Founded |

Below, we present two examples to illustrate potential issues without coloring.

**Example A.15** (Fig. 4(c)). We use the center pooling method to construct such an example. First, we ensure that the center of gravity of the image is at the origin, and secondly, make it asymmetric. The construction here requires a basic point cloud:

$$\vec{\boldsymbol{X}}_0 = \left\{ (-1, 0, 0), \left( \frac{1}{3}, \pm\frac{2\sqrt{2}}{3}, 0 \right), \left( \frac{1}{6}, \pm\frac{\sqrt{35}}{6}, 0 \right) \right\}.$$

All nodes in $\vec{\boldsymbol{X}}_0$ are in the unit circle, which means the distance between each node and the center of mass is the same. And the final point could is

$$\vec{\boldsymbol{X}} = \{\vec{\boldsymbol{X}}_0 \boldsymbol{R}_i\},$$

where we sample some random rotation matrix $\boldsymbol{R}_i$. And then it is obviously only using global pooling will lead all virtual node in the center of mass if we do not use any edge to color this nodes.

However, it could also be solve by high-degree steerable feature. The global steerable features can construct different virtual nodes with tensor product method in Table 9.

**Example A.16** (Degeneration in Uncolored Models). There are specific scenarios in which message passing on uncolored geometric graphs can result in degenerate phenomena. A particularly illustrative example is found in geometric graphs where all node features are identical and all edges are bidirectional. Considering a single-layer EGNN as an example, it may degenerate because the

condition $\boldsymbol{h}_i = \boldsymbol{h}_j$ implies $\boldsymbol{m}_{ij} = \boldsymbol{m}_{ji}$. Consequently, we have $\sum_{\langle i,j \rangle \in \mathcal{E}} \varphi_{\vec{\boldsymbol{x}}}(\boldsymbol{m}_{ij}) \cdot \vec{\boldsymbol{x}}_{ij} = \vec{\boldsymbol{0}}_3$. As a result, Eq. (7) can only yield the coordinate center $\vec{\boldsymbol{x}}_c$. This phenomenon can be observed in the tetrahedron center experiment in § 4, where besides EGNN, advanced models such as HEGNN [49], TFN [33], and MACE [35] also degenerate.

# B  Experiment Details

Our code is available at `https://github.com/GLAD-RUC/Uni-EGNN`.

## B.1  Expressivity Tests

### B.1.1  Completeness Test

In this section, we provide additional details that were omitted in § 4.1, focusing on the following points:

1. The 4-body non-chiral counterexample implemented by the GWL-test (Fig. 2(f) in IASR-test) and the 4-body chiral counterexample (Fig. 2(e) in IASR-test) exhibit a significant issue: the geometric graphs presented by [32] are geometrically isomorphic.

2. Details regarding data construction and model design for the chirality test, which primarily include the coloring strategy and specifics of the construction of 0o-type features.

**Metric.** Referring to the settings of the GWL-test [32], we utilize the average accuracy across ten tests as our evaluation metric. The detailed experimental design is based on the implementation of the GWL-test [32], available in the code repository[5]. In this setup, a single GNN layer is employed to encode the geometric graph, followed by a simple classifier that predicts the label. If a GNN fails to distinguish between two geometric graphs, the output embeddings will be identical, resulting in one graph being classified correctly while the other is misclassified, leading to an accuracy of 50%. Conversely, an accuracy rate exceeding 50% indicates that the GNN can generate distinct embeddings and successfully differentiate between the two geometric graphs.

Additionally, the accuracy of the Basic model in Table 2 does not reach 100% due to a limited number of training epochs for the classifier (set to 100). Increasing the number of epochs (e.g., to 200) can achieve 100% classification accuracy. Notably, the EGNN model, with the same settings, also achieves 100% accuracy. We speculate that EGNN improves the embedding of geometric graphs during the message passing process.

**Results of $k$-chain test.** This experiment comes from GWL-test. The canonical form introduced by our model can easily obtain global information, so only one layer is needed to identify the geometric graphs differentiation task that other models require multiple layers.

Table 10: $k$-chain Test.

| ($k = 4$-chains) GNN Layer | **Number of layers** | | | | |
|---|---|---|---|---|---|
| | $\lfloor \frac{k}{2} \rfloor$ | $\lfloor \frac{k}{2} \rfloor + 1 = \mathbf{3}$ | $\lfloor \frac{k}{2} \rfloor + 2$ | $\lfloor \frac{k}{2} \rfloor + 3$ | $\lfloor \frac{k}{2} \rfloor + 4$ |
| EGNN | 50.0 ± 0.0 | 50.0 ± 0.0 | 50.0 ± 0.0 | 50.0 ± 0.0 | **100.0** ± 0.0 |
| GVP-GNN | 50.0 ± 0.0 | **100.0** ± 0.0 | **100.0** ± 0.0 | **100.0** ± 0.0 | **100.0** ± 0.0 |
| TFN | 50.0 ± 0.0 | 50.0 ± 0.0 | 50.0 ± 0.0 | **100.0** ± 0.0 | **100.0** ± 0.0 |
| MACE | 50.0 ± 0.0 | **100.0** ± 0.0 | **100.0** ± 0.0 | **100.0** ± 0.0 | **100.0** ± 0.0 |
| Basic$_{cpl}$ | **100.0** ± 0.0 | **100.0** ± 0.0 | **100.0** ± 0.0 | **100.0** ± 0.0 | **100.0** ± 0.0 |
| SchNet$_{cpl}$ | **100.0** ± 0.0 | **100.0** ± 0.0 | **100.0** ± 0.0 | **100.0** ± 0.0 | **100.0** ± 0.0 |
| EGNN$_{cpl}$ | **100.0** ± 0.0 | **100.0** ± 0.0 | **100.0** ± 0.0 | **100.0** ± 0.0 | **100.0** ± 0.0 |
| GVP-GNN$_{cpl}$ | **100.0** ± 0.0 | **100.0** ± 0.0 | **100.0** ± 0.0 | **100.0** ± 0.0 | **100.0** ± 0.0 |
| TFN$_{cpl}$ | **100.0** ± 0.0 | **100.0** ± 0.0 | **100.0** ± 0.0 | **100.0** ± 0.0 | **100.0** ± 0.0 |

**Geometrical isomorphism of 4-body counterexample.** In the open source notebook provided by GWL-test [32], four tasks are presented, the first two of which are the 2-body and 3-body tasks tested

---

[5]`https://github.com/chaitjo/geometric-gnn-dojo`.

in Table 1. The other two tasks are termed the *4-body non-chiral counterexample* and the *4-body chiral counterexample*, both originating from IASR-test [69]. According to the experimental results provided in GWL-test, only $MACE_{5\text{-body}}$ successfully passed the 4-body non-chiral counterexample; moreover, the results for the 4-body chiral counterexample were not presented.

We successfully reproduced and cited the experimental results of the 2-body and 3-body from GWL-test [32]. However, we were unable to reproduce the results reported in the article despite multiple testing attempts, consistently achieving an accuracy of only 50%, indicating that $MACE_{5\text{-body}}$ failed this task. Furthermore, recent literature [77] also reported similar failures in the test. Upon thorough investigation, we discovered that the geometric graphs used in the two tests were geometrically isomorphic. We suspect that there may be a minor mistake in the GWL-test.

Specifically, the two graphs $\mathcal{G}_1, \mathcal{G}_2$ of the 4-body non-chiral counterexample are constructed as follows:

1. Consider three sub-graphs: $\mathcal{H}_1 = \{(3, 2, -4), (0, 2, 5), (-3, 2, -4)\}$, $\mathcal{H}_2 = \{(3, -2, -4), (0, -2, 5), (-3, -2, -4)\}$, and $\mathcal{H}_3 = \{(0, 5, 0)\}$. $\boldsymbol{R}_y$ is a random matrix for rotation around the $Oy$-axis. Let $\boldsymbol{M}_x, \boldsymbol{M}_y$ denote the reflection about $yOz$-plane and $zOx$-plane.

2. $\mathcal{G}_1 = \mathcal{H}_1 \cup (\boldsymbol{R}_y \cdot \mathcal{H}_2) \cup \mathcal{H}_3$ and $\mathcal{G}_2 = \mathcal{H}_1 \cup (\boldsymbol{R}_y \cdot \mathcal{H}_2) \cup (\boldsymbol{M}_y \cdot \mathcal{H}_3)$.

Now we show $\mathcal{G}_1 \cong \mathcal{H}_2$, notice there are serval relation:

- Internal symmetry of geometric graph: $\mathcal{H}_i = \boldsymbol{M}_x \cdot \mathcal{H}_i (i = 1, 2)$, $\mathcal{H}_3 = \boldsymbol{R}_y \mathcal{H}_3$;
- Symmetry between geometric graphs: $\mathcal{H}_1 = \boldsymbol{M}_y \cdot \mathcal{H}_2$;
- Properties between geometric transformations: $\boldsymbol{M}_x^2 = \boldsymbol{M}_y^2 = I$, $\boldsymbol{R}_y \boldsymbol{M}_x = \boldsymbol{M}_x \boldsymbol{R}_y$, $\boldsymbol{R}_y \boldsymbol{M}_y = \boldsymbol{M}_y \boldsymbol{R}_y^\top$.

Then we have:

$$
\begin{aligned}
\mathcal{G}_1 =& \mathcal{H}_1 \cup (\boldsymbol{R}_y \cdot \mathcal{H}_2) \cup \mathcal{H}_3 = (\boldsymbol{M}_y \cdot \mathcal{H}_2) \cup (\boldsymbol{R}_y \boldsymbol{M}_y \cdot \mathcal{H}_1) \cup (\boldsymbol{M}_y \cdot \mathcal{H}_3) \\
=& (\boldsymbol{R}_y \boldsymbol{M}_y \cdot \mathcal{H}_1) \cup (\boldsymbol{M}_y \cdot \mathcal{H}_2) \cup (\boldsymbol{M}_y \cdot \mathcal{H}_3) \\
=& (\boldsymbol{M}_y \boldsymbol{R}_y^\top \cdot \mathcal{H}_1) \cup (\boldsymbol{M}_y \cdot \mathcal{H}_2) \cup (\boldsymbol{M}_y \cdot \mathcal{H}_3) \\
=& (\boldsymbol{M}_y \boldsymbol{R}_y^\top) \cdot [\mathcal{H}_1 \cup \boldsymbol{R}_y \mathcal{H}_2 \cup \boldsymbol{R}_y \mathcal{H}_3] \\
=& (\boldsymbol{M}_y \boldsymbol{R}_y^\top) \cdot [\mathcal{H}_1 \cup \boldsymbol{R}_y \mathcal{H}_2 \cup \mathcal{H}_3] \\
=& (\boldsymbol{M}_y \boldsymbol{R}_y^\top \boldsymbol{M}_x) \cdot [\boldsymbol{M}_x \mathcal{H}_1 \cup \boldsymbol{M}_x \boldsymbol{R}_y \mathcal{H}_2 \cup \boldsymbol{M}_x \mathcal{H}_3] \\
=& (\boldsymbol{M}_y \boldsymbol{R}_y^\top \boldsymbol{M}_x) \cdot [\mathcal{H}_1 \cup \boldsymbol{R}_y \boldsymbol{M}_x \mathcal{H}_2 \cup \boldsymbol{M}_x \mathcal{H}_3] \\
=& (\boldsymbol{M}_y \boldsymbol{R}_y^\top \boldsymbol{M}_x) \cdot [\mathcal{H}_1 \cup \boldsymbol{R}_y \mathcal{H}_2 \cup \boldsymbol{M}_x \mathcal{H}_3] \\
=& (\boldsymbol{M}_y \boldsymbol{R}_y^\top \boldsymbol{M}_x) \cdot \mathcal{G}_2 \\
=& (\boldsymbol{M}_x \boldsymbol{R}_y \boldsymbol{M}_y)^\top \cdot \mathcal{G}_2.
\end{aligned}
$$

Moreover, the 4-body chiral counterexample also contains two geometric graphs $\mathcal{G}_3, \mathcal{G}_4$, where $\mathcal{G}_3 \cong \mathcal{G}_4$:

1. Consider two sub-graphs: $\mathcal{H}_4 = \{(3, 0, -4), (0, 0, 5), (-3, 0, -4)\}$ and $\mathcal{H}_5 = \{(0, 5, 0)\}$. $\boldsymbol{M}_x, \boldsymbol{M}_y$ denote the reflection about $yOz$-plane and $zOx$-plane.

2. $\mathcal{G}_3 = \mathcal{H}_4 \cup \mathcal{H}_5$ and $\mathcal{G}_4 = \mathcal{H}_4 \cup (\boldsymbol{M}_y \mathcal{H}_5)$.

Now we show $\mathcal{G}_3 \cong \mathcal{H}_3$, notice there are serval relation:

- Internal symmetry of geometric graph: $\mathcal{H}_3 = \boldsymbol{M}_x \cdot \mathcal{H}_3$, $\mathcal{H}_3 = \boldsymbol{M}_y \mathcal{H}_3$ and $\mathcal{H}_4 = \boldsymbol{M}_x \mathcal{H}_4$;
- Properties between geometric transformations: $\boldsymbol{M}_x^2 = \boldsymbol{M}_y^2 = I$.

Then we have:

$$
\mathcal{G}_3 = \mathcal{H}_4 \cup \mathcal{H}_5 = (\boldsymbol{M}_x \boldsymbol{M}_y) \cdot [\mathcal{H}_4 \cup \boldsymbol{M}_y \mathcal{H}_5] = (\boldsymbol{M}_x \boldsymbol{M}_y) \cdot \mathcal{G}_4.
$$

### B.1.2 Chirality Test

**Details of the chirality test.**

1. Fig. 4(a): Choose $\mathcal{G}_1$ from the 4-body non-chiral counterexample, and another geometric graph is $-\mathcal{G}_1$.

2. Fig. 4(b): Since $\mathcal{G}_3, \mathcal{G}_4$ are symmetric, we modify it to $\mathcal{G}_5 = \{(3, 0, -4), \boldsymbol{R}_y \cdot (0, 0, 5), (-3, 0, -4), (0, 5, 0)\}$, where $\boldsymbol{R}_y$ is a random matirx for rotation around the $Oy$-axis.

3. Fig. 4(c): Similar to Example A.15.

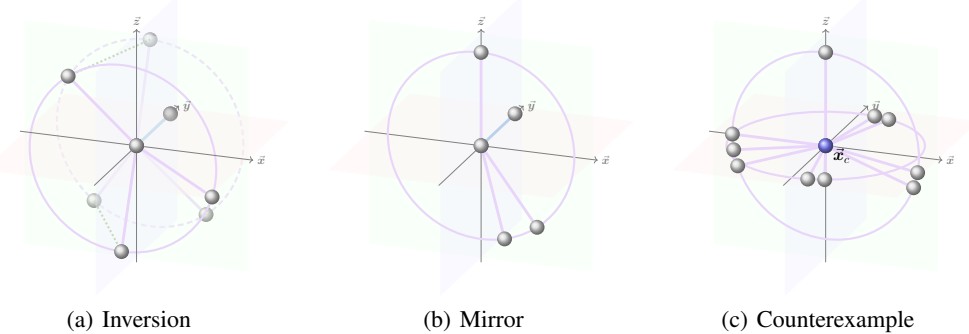

|  (a) Inversion | (b) Mirror | (c) Counterexample |

Figure 4: Examples to test whether the model can recognize chirality. Figs. 4(a) and 4(b) are modified from GWL-test's implementation [36] of IASR-test [69], while Fig. 4(c) is designed by ourselves. Fig. 4(a): Test whether models could distinguish this geometric graph with itself after *inversion*. Fig. 4(b): Test whether models could distinguish this geometric graph with itself after *mirror* about the $xOz$-plane. Fig. 4(c): The task is same as Fig. 4(a), while all nodes are on the unit sphere, and the center point coincides with the center of the sphere. Such counterexample cannot produce separable virtual nodes by simple center distance coloring.

**Details of construction of** `0o`**-type features.** There are two ways to construct `0o`-type features:

1. **Determinant**: Calculate $\det([\vec{v}_2 - \vec{v}_1, \vec{v}_3 - \vec{v}_1, \vec{v}_4 - \vec{v}_1])$;

2. **Tensor Product**: We still begin by constructing $\sum_{i=1}^{N} Y^{(l)}(\vec{x}_i - \vec{x}_c)$, and employing moments, specifically at least 3rd-order moments, to derive `0o`-type features. This approach clearly serves as an extension of the previously established method.

### B.2 Performance Tests

We conducted four experiments:

1. **Tetrahedron center prediction**, which entails predicting graph-level targets, *i.e.* Monge points, twelve-point centers, and incenters, given the coordinates of four points (with the same node features) of a given tetrahedron.

2. 5-**body system** [72] involves predicting node-level targets, *i.e.* the coordinates of each charged particle at a specific moment in the future, given its initial coordinates and velocity.

3. 100-**body system** [66], a generalization of the 5 body system, is used to test the performance of the model in large-scale systems.

4. **MD17 dataset** [68], containing eight molecules, is used to predict node-level targets, *i.e.* the future coordinates of each atom in the molecular trajectory. This dataset can be used to test the performance of the model on real datasets.

5. **Water-3D mini dataset** [66; 67], is used to test the performance of the model in large-scale systems and real datasets. Due to time constraints, we only evaluate a substantial subset (Water-3D-mini: 3,000/300/300 for train/val/test).

Table 11: Details of each dataset.

| Dataset | Target | Type | #sample (train / valid / test) | Node Number | Edge connection |
|---|---|---|---|---|---|
| Tetrahedron | graph-level | Synthetic Dataset | 500 / 2,000 / 2,000 | 4 | `fully_connected` |
| 5-body | node-level | Synthetic Dataset | 5,000 / 2,000 / 2,000 | 5 | `fully_connected` |
| 100-body | node-level | Synthetic Dataset | 5,000 / 2,000 / 2,000 | 100 | `fully_connected` |
| MD17 | node-level | Real-world Dataset | 500 / 2,000 / 2,000 | 5-13 | `distance_cutoff` |
| Water-3D mini | node-level | Real-world Dataset | 3,000 / 300 / 300 | $\sim$8,000 | `distance_cutoff` |

### B.2.1 Tetrahedron Center Prediction and N-body System

We conducted experiments on a single NVIDIA H20 GPU. In both experiments, we configured the batch size to 100, the learning rate to $5 \times 10^{-4}$, and the weight decay to $10^{-12}$. For the tetrahedron center prediction task, the maximum number of training epochs was set to 300. In the case of the $N$-body system task, we implemented early stopping with a patience of 100 steps.

**Metric.** We use Mean Squared Error (MSE) to measure the accuracy of the prediction results in both experiments.

**Tetrahedron Center Prediction.** We generated our dataset using the calculation formulas for the Monge point $\vec{x}_M$, twelve-point center $\vec{x}_T$ and incenter $\vec{x}_I$ from [105]. Given a tetrahedron defined by four points $\vec{x}_O, \vec{x}_A, \vec{x}_B, \vec{x}_C$, with the vectors $\vec{a} = \vec{x}_A - \vec{x}_O, \vec{b} = \vec{x}_B - \vec{x}_O, \vec{c} = \vec{x}_C - \vec{x}_O$, we can compute the desired points as follows:

$$
\begin{aligned}
\vec{x}_M &= \vec{x}_O + \frac{\vec{a} \cdot (\vec{b} + \vec{c})(\vec{b} \times \vec{c}) + \vec{b} \cdot (\vec{c} + \vec{a})(\vec{c} \times \vec{a}) + \vec{c} \cdot (\vec{a} + \vec{b})(\vec{a} \times \vec{b})}{2\vec{a} \cdot (\vec{b} \times \vec{c})}, \\
\vec{x}_T &= \vec{x}_M + \frac{1}{3}(\vec{x}_O - \vec{x}_M), \\
\vec{x}_I &= \frac{\|\vec{b} \times \vec{c}\|\vec{a} + \|\vec{c} \times \vec{a}\|\vec{b} + \|\vec{a} \times \vec{b}\|\vec{c}}{\|\vec{b} \times \vec{c}\| + \|\vec{c} \times \vec{a}\| + \|\vec{a} \times \vec{b}\| + \|\vec{b} \times \vec{c} + \vec{c} \times \vec{a} + \vec{a} \times \vec{b}\|}.
\end{aligned}
\tag{20}
$$

We utilize 500/2000/2000 samples for training, validation, and testing, respectively. $\epsilon = 10^{-6}$ was added to the denominator to avoid division by zero. In addition, to avoid numerical problems caused by the center point being extremely large, we generated the circumscribed sphere of the tetrahedron when generating the tetrahedron, ensuring that the radius of the sphere is less than or equal to 6.

$N$**-body system.** $N$-body system [72] is a dataset generated from simulations. In our simulations, each system consists of 5 charged particles with random charges $c_i \in \{\pm 1\}$, whose movements are governed by Coulomb forces. The task is to estimate, a node-level target, *i.e.* the coordinates of the $N$ particles after 1000 timesteps. The initial node feature is set as the kinetic energy, while the edge features include the distance and the product of the charges. We use 5000/2000/2000 samples for training, validation, and testing like the setting in HEGNN [49]. Since the particles have different initial velocities, they initially have different characteristics (coloring).

### B.2.2 Implementation of Baselines

We undertook further development based on the codes of EGNN [43], TFN [33], and MACE [35] provided in the GWL-test repository. Following our testing, we eliminated the `Gate` modules from both TFN and MACE, as their inclusion resulted in significantly degraded performance. Additionally, for HEGNN [49], we utilized the code[6] from the original paper and opted for the `inner product normalization` option.

Each model utilizes at most 4 layers, and the dimension of the hidden embedding is fixed at 64. The last three models employ at most 2nd-degree steerable features. Both TFN and MACE utilize 8 channels for steerable features, with the correlation parameter for MACE set to 4.

### B.2.3 Implementation of Our Models

We provide two complete implementations of EGNN [43] and TFN [33] as follows.

Unlike FastEGNN [66], the virtual nodes in this approach are not updated following their initialization during the message-passing layers. We define the normalization coefficient as $\alpha_i \coloneqq 1/|\mathcal{N}(i)|$. For

---

[6] https://github.com/GLAD-RUC/HEGNN/.

the $i$th node, we denote its features as $\boldsymbol{h}_i$ and its coordinates as $\vec{\boldsymbol{x}}_i$. Here, $C$ represents the channels of the virtual nodes, and $\vec{\boldsymbol{Z}} \in \mathbb{R}^{C \times 3}$ denotes all coordinates. Specifically, for TFN$_{\mathrm{cpl}}$-global, we employ the canonical form to build a new feature $\boldsymbol{w}_i$ for each node, thereby rescaling the norm of steerable features in a manner analogous to HEGNN [49], which is consistent with Theorem 3.2.

Table 12: Complete global models.

| | EGNN$_{\mathrm{cpl}}$-global | TFN$_{\mathrm{cpl}}$-global |
|---|---|---|
| Node Coloring | $\boldsymbol{h}_i = \texttt{Color}(\boldsymbol{h}_i, \texttt{distance\_to\_center})$ | $\tilde{\boldsymbol{h}}_i^{(0)} = \texttt{Color}(\tilde{\boldsymbol{h}}_i^{(0)}, \texttt{distance\_to\_center})$ |
| Generate VNs | $\vec{\boldsymbol{Z}} = \mathbf{1}_C \vec{\boldsymbol{x}}_c + \alpha_i \sum_{i=1}^N \varphi_Z(\boldsymbol{h}_i) \cdot \vec{\boldsymbol{x}}_{ic}$ | $\vec{\boldsymbol{Z}} = \mathbf{1}_C \vec{\boldsymbol{x}}_c + \alpha_i \sum_{i=1}^N \varphi_Z(\tilde{\boldsymbol{h}}_i^{(0)}) \cdot \vec{\boldsymbol{x}}_{ic}$ |
| Create Canonical Form | $\vec{\boldsymbol{M}}_i = \mathbf{1}_C \vec{\boldsymbol{x}}_i - \vec{\boldsymbol{Z}}, \; \boldsymbol{m}_{i,\mathrm{cpl}} = \vec{\boldsymbol{M}}_i^\top \vec{\boldsymbol{M}}_i / \|\vec{\boldsymbol{M}}_i^\top \vec{\boldsymbol{M}}_i\|_{\mathrm{F}}$ | $\vec{\boldsymbol{M}}_i = \mathbf{1}_C \vec{\boldsymbol{x}}_i - \vec{\boldsymbol{Z}}, \; \boldsymbol{m}_{i,\mathrm{cpl}} = \vec{\boldsymbol{M}}_i^\top \vec{\boldsymbol{M}}_i / \|\vec{\boldsymbol{M}}_i^\top \vec{\boldsymbol{M}}_i\|_{\mathrm{F}}$ |
| Update Node Feature | $\boldsymbol{h}_i = \boldsymbol{h}_i + \varphi_{\mathrm{cpl}}(\boldsymbol{m}_{i,\mathrm{cpl}}), \boldsymbol{h}_i = \texttt{Message\_Passing}(\boldsymbol{h}_i, \mathcal{G})$ | $\boldsymbol{w}_i = \tilde{\boldsymbol{h}}_i^{(0)} + \varphi_{\mathrm{cpl}}(\boldsymbol{m}_{i,\mathrm{cpl}})$ |
| Message Passing ($\times L$) | $\boldsymbol{m}_{ij} = \varphi_{\boldsymbol{m}}(\boldsymbol{h}_i, \boldsymbol{h}_j, \boldsymbol{e}_{ij}, \|\vec{\boldsymbol{x}}_{ij}\|^2), \; \vec{\boldsymbol{m}}_{ij} = \varphi_{\vec{\boldsymbol{x}}}(\boldsymbol{m}_{ij}) \cdot \vec{\boldsymbol{x}}_{ij}$
$\boldsymbol{m}_i = \alpha_i \sum_{j \in \mathcal{N}(i)} \boldsymbol{m}_{ij}, \; \vec{\boldsymbol{m}}_i = \alpha_i \sum_{j \in \mathcal{N}(i)} \vec{\boldsymbol{m}}_{ij}$
$\boldsymbol{h}_i = \boldsymbol{h}_i + \varphi_{\boldsymbol{h}}(\boldsymbol{h}_i, \boldsymbol{m}_i), \; \vec{\boldsymbol{x}}_i = \vec{\boldsymbol{x}}_i + \vec{\boldsymbol{m}}_i$ | $\tilde{\boldsymbol{m}}_{ij}^{(\mathbb{L})} = \tilde{\boldsymbol{h}}_i^{(\mathbb{L})} \otimes_{\mathrm{cg}} Y^{(\mathbb{L})}(\vec{\boldsymbol{x}}_{ij})$
$\tilde{\boldsymbol{h}}_i^{(\mathbb{L})} = \tilde{\boldsymbol{h}}_i^{(\mathbb{L})} + \alpha_i \sum_{j \in \mathcal{N}(i)} \tilde{\boldsymbol{m}}_{ij}^{(\mathbb{L})}$
$\tilde{\boldsymbol{h}}_i^{(l)} = \varphi_{\mathrm{cpl}}(\tilde{\boldsymbol{h}}_i^{(0)}, \boldsymbol{w}_i) \cdot \tilde{\boldsymbol{h}}_i^{(l)}, \; \boldsymbol{w}_i = \boldsymbol{w}_i + \varphi_{\boldsymbol{w}}(\boldsymbol{w}_i, \tilde{\boldsymbol{h}}_i^{(0)})$ |
| Readout (node-level) | $\boldsymbol{h}_i = \varphi_{\mathrm{out}}(\boldsymbol{h}_i), \; \vec{\boldsymbol{x}}_i = \vec{\boldsymbol{x}}_i$ | $\boldsymbol{h}_i = \varphi_{\mathrm{out}}(\tilde{\boldsymbol{h}}_i^{(0)}), \; \vec{\boldsymbol{x}}_i = \vec{\boldsymbol{x}}_i + \texttt{e3nn.o3.Linear}(\tilde{\boldsymbol{h}}_i^{(1)})$ |
| Readout (graph-level) | $\boldsymbol{h}_{\mathcal{G}} = \varphi_{\mathrm{pool}}(\frac{1}{N}\sum_{i=1}^N \boldsymbol{h}_i), \; \vec{\boldsymbol{x}}_{\mathcal{G}} = \frac{1}{N}\sum_{i=1}^N \vec{\boldsymbol{x}}_i$ | $\boldsymbol{h}_{\mathcal{G}} = \varphi_{\mathrm{pool}}(\frac{1}{N}\sum_{i=1}^N \boldsymbol{h}_i), \; \vec{\boldsymbol{x}}_{\mathcal{G}} = \frac{1}{N}\sum_{i=1}^N \vec{\boldsymbol{x}}_i$ |

To enhance the model's performance, we permit the virtual nodes to be updated layer by layer, similar to the approach taken in FastEGNN [66]. We denote the normalization coefficient $\alpha_i := 1/|\mathcal{N}(i)|$. We define the normalization coefficient as $\alpha_i := 1/|\mathcal{N}(i)|$. Notably, we allocate $C$ virtual nodes to each node, denoted as $\vec{\boldsymbol{Z}}_i \in \mathbb{R}^{C \times 3}$, and these $C$ virtual nodes share a scalar feature represented by $\boldsymbol{s}_i$.

Table 13: Complete local models.

| | EGNN$_{\mathrm{cpl}}$-local | TFN$_{\mathrm{cpl}}$-local |
|---|---|---|
| Generate VNs | $\boldsymbol{m}_{ij} = \varphi_{\boldsymbol{m}}(\boldsymbol{h}_i, \boldsymbol{h}_j, \boldsymbol{e}_{ij}, \|\vec{\boldsymbol{x}}_{ij}\|^2)$
$\boldsymbol{s}_i = \varphi_{\boldsymbol{s}}(\varphi_{\mathrm{init}}(\boldsymbol{h}_i), \boldsymbol{m}_{ij})$
$\vec{\boldsymbol{Z}}_i = \mathbf{1}_C \vec{\boldsymbol{x}}_i + \alpha_i \sum_{j \in \mathcal{N}(i)}^N \varphi_Z(\boldsymbol{h}_i) \cdot \vec{\boldsymbol{x}}_{ij}$ | $\boldsymbol{m}_{ij} = \varphi_{\boldsymbol{m}}(\tilde{\boldsymbol{h}}_i^{(0)}, \tilde{\boldsymbol{h}}_j^{(0)}, \boldsymbol{e}_{ij}, \|\vec{\boldsymbol{x}}_{ij}\|^2)$
$\boldsymbol{s}_i = \varphi_{\boldsymbol{s}}(\varphi_{\mathrm{init}}(\tilde{\boldsymbol{h}}_i^{(0)}), \boldsymbol{m}_{ij})$
$\vec{\boldsymbol{Z}}_i = \mathbf{1}_C \vec{\boldsymbol{x}}_i + \alpha_i \sum_{j \in \mathcal{N}(i)}^N \varphi_Z(\boldsymbol{h}_i) \cdot \vec{\boldsymbol{x}}_{ij}$ |
| Message Passing ($\times L$)
(real & virtual) | $\boldsymbol{m}_{ij} = \varphi_{\boldsymbol{m}}(\boldsymbol{h}_i, \boldsymbol{h}_j, \boldsymbol{e}_{ij}, \|\vec{\boldsymbol{x}}_{ij}\|^2), \; \vec{\boldsymbol{m}}_{ij} = \varphi_{\vec{\boldsymbol{x}}}(\boldsymbol{m}_{ij}) \cdot \vec{\boldsymbol{x}}_{ij}$
$\boldsymbol{m}_i = \alpha_i \sum_{j \in \mathcal{N}(i)} \boldsymbol{m}_{ij}, \; \vec{\boldsymbol{m}}_i = \alpha_i \sum_{j \in \mathcal{N}(i)} \vec{\boldsymbol{m}}_{ij}$
$\boldsymbol{h}_i = \boldsymbol{h}_i + \varphi_{\boldsymbol{h}}(\boldsymbol{h}_i, \boldsymbol{m}_i), \; \vec{\boldsymbol{x}}_i = \vec{\boldsymbol{x}}_i + \vec{\boldsymbol{m}}_i$<hr>$\vec{\boldsymbol{M}}_{ij} = \mathbf{1}_C \vec{\boldsymbol{x}}_j - \vec{\boldsymbol{Z}}_i, \boldsymbol{m}_{ij,\mathrm{cpl}} = \vec{\boldsymbol{M}}_{ij}^\top \vec{\boldsymbol{M}}_{ij} / \|\vec{\boldsymbol{M}}_{ij}^\top \vec{\boldsymbol{M}}_{ij}\|_{\mathrm{F}}$
$\boldsymbol{m}_{ij,\mathrm{cpl}} = \varphi_{\boldsymbol{m},\mathrm{cpl}}(\boldsymbol{s}_i, \boldsymbol{h}_j, \boldsymbol{m}_{ij,\mathrm{cpl}})$
$\boldsymbol{h}_i = \boldsymbol{h}_i + \varphi_{\boldsymbol{h},\mathrm{cpl}}(\boldsymbol{m}_{ij,\mathrm{cpl}}), \; \boldsymbol{s}_i = \boldsymbol{s}_i + \varphi_{\boldsymbol{s},\mathrm{cpl}}(\boldsymbol{m}_{ij,\mathrm{cpl}})$
$\vec{\boldsymbol{x}}_i = \vec{\boldsymbol{x}}_i + \alpha_i \sum_{j \in \mathcal{N}(i)} \varphi_{\vec{\boldsymbol{x}}}^{(v)}(\boldsymbol{m}_{ij,\mathrm{cpl}}) \cdot \vec{\boldsymbol{M}}_{ij}$
$\vec{\boldsymbol{Z}}_i = \vec{\boldsymbol{Z}}_i + \alpha_i \sum_{j \in \mathcal{N}(i)} \varphi_{\vec{\boldsymbol{Z}}}^{(v)}(\boldsymbol{m}_{ij,\mathrm{cpl}}) \cdot (-\vec{\boldsymbol{M}}_{ij})$ | $\tilde{\boldsymbol{m}}_{ij}^{(\mathbb{L})} = \tilde{\boldsymbol{h}}_i^{(\mathbb{L})} \otimes_{\mathrm{cg}} Y^{(\mathbb{L})}(\vec{\boldsymbol{x}}_{ij})$
$\tilde{\boldsymbol{h}}_i^{(\mathbb{L})} = \tilde{\boldsymbol{h}}_i^{(\mathbb{L})} + \alpha_i \sum_{j \in \mathcal{N}(i)} \tilde{\boldsymbol{m}}_{ij}^{(\mathbb{L})}$
$\tilde{\boldsymbol{h}}_i^{(l)} = \varphi_{\mathrm{cpl}}(\tilde{\boldsymbol{m}}_i^{(0)}) \cdot \tilde{\boldsymbol{h}}_i^{(l)}$<hr>$\vec{\boldsymbol{M}}_{ij} = \mathbf{1}_C \vec{\boldsymbol{x}}_j - \vec{\boldsymbol{Z}}_i, \boldsymbol{m}_{ij,\mathrm{cpl}} = \vec{\boldsymbol{M}}_{ij}^\top \vec{\boldsymbol{M}}_{ij} / \|\vec{\boldsymbol{M}}_{ij}^\top \vec{\boldsymbol{M}}_{ij}\|_{\mathrm{F}}$
$\boldsymbol{m}_{ij,\mathrm{cpl}} = \varphi_{\boldsymbol{m},\mathrm{cpl}}(\boldsymbol{s}_i, \tilde{\boldsymbol{h}}_j^{(0)}, \boldsymbol{m}_{ij,\mathrm{cpl}})$
$\tilde{\boldsymbol{h}}_i^{(0)} = \tilde{\boldsymbol{h}}_i^{(0)} + \varphi_{\tilde{\boldsymbol{h}}^{(0)},\mathrm{cpl}}(\boldsymbol{m}_{ij,\mathrm{cpl}}), \; \boldsymbol{s}_i = \boldsymbol{s}_i + \varphi_{\boldsymbol{s},\mathrm{cpl}}(\boldsymbol{m}_{ij,\mathrm{cpl}})$
$\vec{\boldsymbol{x}}_i = \vec{\boldsymbol{x}}_i + \alpha_i \sum_{j \in \mathcal{N}(i)} \varphi_{\vec{\boldsymbol{x}}}^{(v)}(\boldsymbol{m}_{ij,\mathrm{cpl}}) \cdot \vec{\boldsymbol{M}}_{ij}$
$\vec{\boldsymbol{Z}}_i = \vec{\boldsymbol{Z}}_i + \alpha_i \sum_{j \in \mathcal{N}(i)} \varphi_{\vec{\boldsymbol{Z}}}^{(v)}(\boldsymbol{m}_{ij,\mathrm{cpl}}) \cdot (-\vec{\boldsymbol{M}}_{ij})$ |
| Readout (node-level) | $\boldsymbol{h}_i = \varphi_{\mathrm{out}}(\boldsymbol{h}_i), \; \vec{\boldsymbol{x}}_i = \vec{\boldsymbol{x}}_i$ | $\boldsymbol{h}_i = \varphi_{\mathrm{out}}(\tilde{\boldsymbol{h}}_i^{(0)}), \; \vec{\boldsymbol{x}}_i = \vec{\boldsymbol{x}}_i + \texttt{e3nn.o3.Linear}(\tilde{\boldsymbol{h}}_i^{(1)})$ |
| Readout (graph-level) | $\boldsymbol{h}_{\mathcal{G}} = \varphi_{\mathrm{pool}}(\frac{1}{N}\sum_{i=1}^N \boldsymbol{h}_i), \; \vec{\boldsymbol{x}}_{\mathcal{G}} = \frac{1}{N}\sum_{i=1}^N \vec{\boldsymbol{x}}_i$ | $\boldsymbol{h}_{\mathcal{G}} = \varphi_{\mathrm{pool}}(\frac{1}{N}\sum_{i=1}^N \boldsymbol{h}_i), \; \vec{\boldsymbol{x}}_{\mathcal{G}} = \frac{1}{N}\sum_{i=1}^N \vec{\boldsymbol{x}}_i$ |

**Parameter Quantities.** Here, we present the number of parameters for each model in the tasks shown in § 4.2. Since the inputs for the two tasks differ, the number of parameters for models with the same number of layers also varies.

Table 14: Tetrahedron dataset.

| | Total Parameters | | | |
|---|---|---|---|---|
| | 1-layer | 2-layer | 3-layer | 4-layer |
| EGNN | 33.7k | 67.1k | 100.6k | 134.1k |
| FastEGNN | 67.4k | 134.4k | 201.7k | 268.8k |
| HEGNN | 46.9k | 84.9k | 122.9k | 160.9k |
| TFN | 46.5k | 93.0k | 139.4k | 185.9k |
| MACE | 48.9k | 97.8k | 126.6k | 195.5k |
| Equiformer | 313.5k | 340.5k | 367.5k | 349.4k |
| EGNN$_{\mathrm{cpl}}$-global | 181.7k | 215.2k | 248.7k | 282.1k |
| EGNN$_{\mathrm{cpl}}^*$-global | 46.8k | 55.4k | 63.9k | 72.5k |
| EGNN$_{\mathrm{cpl}}$-local | 107.5k | 180.7k | 253.8k | 327.0k |
| EGNN$_{\mathrm{cpl}}^*$-local | 28.2k | 47.3k | 66.5k | 85.7k |
| TFN$_{\mathrm{cpl}}$-global | 215.2k | 291.1k | 267.1k | 443.1k |
| TFN$_{\mathrm{cpl}}^*$-global | 70.3k | 104.6k | 138.9k | 173.3k |
| TFN$_{\mathrm{cpl}}$-local | 138.0k | 241.7k | 345.5k | 449.2k |
| TFN$_{\mathrm{cpl}}^*$-local | 51.0k | 93.1k | 135.2k | 177.3k |

Table 15: $N$-body dataset.

| | Total Parameters | | | |
|---|---|---|---|---|
| | 1-layer | 2-layer | 3-layer | 4-layer |
| EGNN | 33.5k | 66.9k | 100.4k | 133.8k |
| FastEGNN | 67.5k | 134.6k | 201.4k | 268.4k |
| HEGNN | 46.7k | 84.6k | 122.6k | 160.5k |
| TFN | 46.4k | 92.8k | 139.2k | 185.6k |
| MACE | 47.2k | 94.3k | 141.5k | 188.6k |
| Equiformer | 313.6k | 340.6k | 367.5k | 394.5k |
| EGNN$_{\mathrm{cpl}}$-global | 181.3k | 214.7k | 248.1k | 281.5k |
| EGNN$_{\mathrm{cpl}}^*$-global | 46.6k | 55.1k | 63.6k | 72.1k |
| EGNN$_{\mathrm{cpl}}$-local | 107.3k | 180.4k | 253.5k | 326.6k |
| EGNN$_{\mathrm{cpl}}^*$-local | 28k | 47.2k | 66.4k | 85.5k |
| TFN$_{\mathrm{cpl}}$-global | 215.6k | 291.6k | 367.7k | 443.7k |
| TFN$_{\mathrm{cpl}}^*$-global | 70.5k | 104.8k | 139.2k | 173.6k |
| TFN$_{\mathrm{cpl}}$-local | 138.2k | 242.0k | 345.8k | 449.7k |
| TFN$_{\mathrm{cpl}}^*$-local | 51.1k | 93.3k | 135.4k | 177.5k |

## C Rethinking the Equivariance and General Functions

In this section, we give a Fourier series-like approach to understanding equivariant GNNs.

### C.1 Rethinking equivariance: A perspective from Fourier expansion.

According to [106], the Fourier expansion of a multivariate function necessitates the simultaneous consideration of the basis functions associated with all variables, as demonstrated below:

$$f(t) = \sum_{n \in \mathbb{Z}} f_n \exp(-2\pi j n t) \longrightarrow f(\boldsymbol{t}) = \sum_{\boldsymbol{n} \in \mathbb{Z}^d} f_{\boldsymbol{n}} \exp(-2\pi j \boldsymbol{n}^\top \boldsymbol{t}) = f(\boldsymbol{t}) = \sum_{\boldsymbol{n} \in \mathbb{Z}^d} f_{\boldsymbol{n}} \prod_{k=1}^d \exp(-2\pi j n_k t_k),$$

(21)

represents a multi-dimensional variable. In fact, any function defined on the unit sphere can be expressed as a summation of spherical harmonics. Let $\hat{\boldsymbol{x}}$ denote a point on the unit sphere.

$$f(\hat{\boldsymbol{x}}) = \sum_{l=0}^{\infty} \sum_{m=-l}^{l} f_m^{(l)} Y_m^{(l)}(\hat{\boldsymbol{x}}) = \sum_{l=0}^{\infty} \langle \boldsymbol{a}^{(l)}, Y^{(l)}(\hat{\boldsymbol{x}}) \rangle,$$

(22)

This formulation offers two representations. The fully expanded form aligns more closely with the Fourier series, while the inner product formulation facilitates a connection to equivariance. At this juncture, we introduce a second point $\hat{\boldsymbol{y}}$ on the unit sphere and consider the expansion of the function with two inputs, $\hat{\boldsymbol{x}}$ and $\hat{\boldsymbol{y}}$.

$$f(\hat{\boldsymbol{x}}, \hat{\boldsymbol{y}}) = \sum_{l_1=0}^{\infty} \sum_{l_2=0}^{\infty} \sum_{m_1=-l_1}^{l_1} \sum_{m_2=-l_2}^{l_2} f_{m_1,m_2}^{(l_1,l_2)} Y_{m_1}^{(l_1)}(\hat{\boldsymbol{x}}) Y_{m_2}^{(l_2)}(\hat{\boldsymbol{y}}).$$

(23)

We observe that the product $Y_{m_1}^{(l_1)}(\hat{\boldsymbol{x}}) Y_{m_2}^{(l_2)}(\hat{\boldsymbol{y}})$ constructs $(2l_1 + 1) \cdot (2l_2 + 1)$ basis functions, *i.e.* elements in $Y^{(l_1)}(\hat{\boldsymbol{x}}) \otimes Y^{(l_2)}(\hat{\boldsymbol{y}}) \in \mathbb{R}^{(2l_1+1)\cdot(2l_2+1)}$. Let $\boldsymbol{f}^{(l_1 \otimes l_2)} \in \mathbb{R}^{(2l_1+1)\cdot(2l_2+1)}$ be the coefficients. Consequently, the expansion can also be expressed in an inner product form as follows:

$$f(\hat{\boldsymbol{x}}, \hat{\boldsymbol{y}}) = \sum_{l_1=0}^{\infty} \sum_{l_2=0}^{\infty} \langle \boldsymbol{f}^{(l_1 \otimes l_2)}, Y^{(l_1)}(\hat{\boldsymbol{x}}) \otimes Y^{(l_2)}(\hat{\boldsymbol{y}}) \rangle.$$

(24)

According to the notation of tensor product, we denote the $l$th-degree steerable feature derived from the tensor product $Y^{(l_1)}(\hat{\boldsymbol{x}}) \otimes Y^{(l_2)}(\hat{\boldsymbol{y}})$ as $\tilde{\boldsymbol{b}}^{(l_1 \otimes l_2 \to l)}(\hat{\boldsymbol{x}}, \hat{\boldsymbol{y}})$, which can be expressed as follows:

$$\bigoplus_{l=|l_1-l_2|}^{l_1+l_2} \tilde{\boldsymbol{b}}^{(l_1 \otimes l_2 \to l)}(\hat{\boldsymbol{x}}, \hat{\boldsymbol{y}}) = \boldsymbol{Q}^{(l_1 \otimes l_2)} \cdot \left( Y^{(l_1)}(\hat{\boldsymbol{x}}) \otimes Y^{(l_2)}(\hat{\boldsymbol{y}}) \right).$$

(25)

Let us define $\bigoplus_{l=|l_1-l_2|}^{l_1+l_2} \boldsymbol{w}^{(l_1 \otimes l_2 \to l)} = (\boldsymbol{Q}^{(l_1 \otimes l_2)})^\top \boldsymbol{f}^{(l_1 \otimes l_2)}$. This leads us to the following expression:

$$f(\hat{\boldsymbol{x}}, \hat{\boldsymbol{y}}) = \sum_{l=0}^{\infty} \sum_{|l_1-l_2| \le l \le l_1+l_2} \langle \boldsymbol{w}^{(l_1 \otimes l_2 \to l)}, \tilde{\boldsymbol{b}}^{(l_1 \otimes l_2 \to l)}(\hat{\boldsymbol{x}}, \hat{\boldsymbol{y}}) \rangle,$$

(26)

where a $l$th-degree steerable function $\tilde{\boldsymbol{t}}^{(l)}(\hat{\boldsymbol{x}}, \hat{\boldsymbol{y}})$ incorporates only the $l$th-degree components in Eq. (26), and can be expressed as follows:

$$\tilde{\boldsymbol{t}}^{(l)}(\hat{\boldsymbol{x}}, \hat{\boldsymbol{y}}) = \sum_{|l_1-l_2| \le l \le l_1+l_2} w^{(l_1 \otimes l_2 \to l)} \cdot \tilde{\boldsymbol{b}}^{(l_1 \otimes l_2 \to l)}(\hat{\boldsymbol{x}}, \hat{\boldsymbol{y}}),$$

(27)

Eq. (27) offers a more concrete understanding of the framework. It is important to note, however, the formula in Eq. (3) is invariant to permutations of the inputs, while the inputs in Eq. (27) are presented with a specific order. Nevertheless, this specificity does not preclude us from conducting an analysis based on this formulation:

1. Since $l_1$ and $l_2$ can take any values, there will be infinite basis functions in $\mathbb{B}^{(l)}$ when the geometric graph accommodates interactions involving more than two bodies.

2. Not all bases in Eq. (27) will be meaningful; that is, some bases may not emerge in the objective function, although they satisfy the equivariance constraints.

From this perspective, the advantages and disadvantages of the equivariant model become evident. By incorporating an equivariance prior into the model design, it effectively eliminates components that cannot exist within the framework. However, this constraint necessitates a limited selection of operators in the model design. In the current setup, only the tensor product exhibits strict equivariance (while others can be transformed and expressed using tensor products, *e.g.* tensor decomposition in frame averaging [56; 84]), whereas other operators, such as group convolution, require approximations under current floating-point operation systems and can achieve only partial equivariance. This limitation, in turn, constrains the model's capacity for learning.

## C.2 Rethinking General Functions: the Equivariant Decomposition

Generally, the input of a geometric graph could be denoted as $\{\vec{x}_i\}_{i=1}^N$, without loss of generality, we assume that these coordinates are decentralized. Although in practice, people generally use edges as input (*i.e.* $\{\vec{x}_i - \vec{x}_j\}_{i,j=1}^N$), for the sake of convenience we still discuss the vectors of points here, and it is easy to verify that they are equivalent. Then a normal function $\boldsymbol{f} : \mathbb{R}^{3\times N} \to \mathbb{R}^{2l+1}$ we want in a $l$-degree problem could be rewritten as:

$$\boldsymbol{f}^{(l)}(\mathcal{G}) = \sum_{\tilde{\boldsymbol{b}}_\alpha^{(l)}(\mathcal{G})\in\mathbb{B}^{(l)}} w_\alpha \tilde{\boldsymbol{b}}_\alpha^{(l)}(\mathcal{G}) + \boldsymbol{f}_{\text{else}}^{(l)}. \tag{28}$$

To completely give the expansion formula of such $\boldsymbol{f}^{(l)}$, we need to analyze each dimension separately. And we denote the $m$th-dimension ($-l \le m \le l$) of $\boldsymbol{f}^{(l)}$ as $f^{(l,m)}$ (similarly, $b^{(l,m)}$ represents the $m$th dimension in $\tilde{\boldsymbol{b}}^{(l)}$), which is:

$$f^{(l,m)}(\mathcal{G}) = \sum_{\tilde{\boldsymbol{b}}_\alpha^{(l)}(\mathcal{G})\in\mathbb{B}^{(l)}} w_\alpha^{(l)} \tilde{\boldsymbol{b}}_\alpha^{(l,m)}(\mathcal{G}) + \sum_{s=0}^\infty \sum_{\tilde{\boldsymbol{b}}_\beta^{(s)}(\mathcal{G})\in\mathbb{B}^{(s)}} \sum_{t=-s}^s w_\beta^{(s,t)} \tilde{\boldsymbol{b}}_\beta^{(s,t)}(\mathcal{G}). \tag{29}$$

And it could find that he $m$th-dimension $f_{\text{else}}^{(l,m)}$ of $\boldsymbol{f}_{\text{else}}^{(l)}$ could be donated as:

$$f_{\text{else}}^{(l,m)}(\mathcal{G}) = \sum_{s=0}^\infty \sum_{\tilde{\boldsymbol{b}}_\beta^{(s)}(\mathcal{G})\in\mathbb{B}^{(s)}} \sum_{t=-s}^s w_\beta^{(s,t)} b_\beta^{(s,t)}(\mathcal{G}). \tag{30}$$

Here $\alpha$ and $\beta$ are dummy index for enumeration and both $w_\alpha^{(l)}$ and $w_\beta^{(l,m)}$ are *constant* (or only the function about all radials $\{\|\vec{x}_i\|\}_{i=1}^N$). Notice that the same basis may appear in both two items in the equation, to ensure uniqueness, we set

$$w_\alpha^{(l)} = \underset{w_\alpha^{(l)}}{\arg\min} \|\boldsymbol{f}_{\text{else}}^{(l)}(\mathcal{G})\|^2 = \underset{w_\alpha^{(l)}}{\arg\min} \sum_{\tilde{\boldsymbol{b}}_\beta^{(t)}(\mathcal{G})\in\mathbb{B}^{(t)}} \sum_{s=-t}^t (w_\beta^{(s,t)})^2. \tag{31}$$

And with definition of equivariance error in Spherical CNNs [107], we could find that

$$\begin{aligned} \mathcal{L}_{\text{equ}} &:= \mathbb{E}\left[\left\|\rho^{(l)}(\mathfrak{g})\cdot\boldsymbol{f}^{(l)}(\mathcal{G}) - \boldsymbol{f}^{(l)}(\mathfrak{g}\cdot\mathcal{G})\right\|\right] \\ &= \mathbb{E}\left[\left\|\rho^{(l)}(\mathfrak{g})\cdot\boldsymbol{f}_{\text{else}}^{(l)}(\mathcal{G}) - \boldsymbol{f}_{\text{else}}^{(l)}(\mathfrak{g}\cdot\mathcal{G})\right\|\right] \\ &\le \mathbb{E}\left[\left\|\rho^{(l)}(\mathfrak{g})\cdot\boldsymbol{f}_{\text{else}}^{(l)}(\mathcal{G})\| + \|\boldsymbol{f}_{\text{else}}^{(l)}(\mathfrak{g}\cdot\mathcal{G})\right\|\right] \\ &\le 2\cdot\|\boldsymbol{f}_{\text{else}}^{(l)}(\mathcal{G})\|. \end{aligned} \tag{32}$$

Traditionally, practitioners have opted for data-driven methods, such as data augmentation, to achieve equivariance. This approach aims to constrain the norm $\|\boldsymbol{f}_{\text{else}}^{(l)}\|$ by adjusting the parameters $w_\beta^{(s,t)}$ throughout training, with the goal of minimizing their values. In contrast, models derived from geometric learning directly incorporate the equivariance constraint into the model architecture, effectively achieving $\|\boldsymbol{f}_{\text{else}}^{(l)}\| = 0$. A fundamental aspect of these models is the construction of the

basis set $\mathbb{B}^{(l)}$, allowing them to systematically remove irrelevant or non-existent components from the learning process.

Moreover, for a $\boldsymbol{f}^{(l)}(\mathcal{G})$, obtaining $\boldsymbol{f}_{\text{test}}^{(l)}(\mathcal{G})$ or calculating $\|\boldsymbol{f}_{\text{test}}^{(l)}(\mathcal{G})\|$ can be intractable. However, a plausible approach is to utilize another general model whose output is bounded by a constant $M$ to learn the residue. This leads us to the relationship $\|\boldsymbol{f}_{\text{test}}^{(l)}(\mathcal{G})\| \leq \|\boldsymbol{f}^{(l)}(\mathcal{G})\| \leq M$. We introduce this concept here, hoping it will inspire future research.

Additionally, it is important to emphasize that we only require the inference model to be equivariant. In practice, we allow the use of non-equivariant methods for training our model. For instance, noise injection in SaVeNet [108] and the Huber loss in MEAN [102] are examples of such methods. These approaches modify the coefficients $w_\alpha$, but they do not violate the equivariance constraint.

