# OpenReview forum: "Universally Invariant Learning in Equivariant GNNs"
_NeurIPS.cc/2025/Conference — NeurIPS 2025 poster_

### Official Review · Reviewer_q9Lr · 2025-06-28

**Clarity:** 2
**Significance:** 4
**Originality:** 3
**Rating:** 4
**Confidence:** 2

**Summary:**

The paper considers equivariant GNNs for geometric graphs (ones where the nodes / edges also have positional information). They extend existing work towards constructing complete GNNs (ones which can approximate any equivariant function) by reframing GNNs as sums over basis sets with weights that depend on the input graph. They provide two algorithms for constructing such GNNs, one of which is slower but works for all geometric graphs, the other of which requires that four non-coplanar reference nodes can be learned by an equivariant GNN. They evaluate their system against strong baselines across 7 datasets.

**Questions:**

Questions:
1. L114 -> what is C_H? It has not been defined.
2. L155 "assume uniform node and edge features" -> Is this a fair assumption? This will affect permutation invariance
3. L295-L297 "Strictly speaking, the virtual nodes forming the tetrahedron should also be included in Algo. 4, but we omit this term as it degrades performance in our experiments." -> Is the theory still valid without this? Do the authors have an intuition as to why this degraded performance?

Minor comments:
1. L211 "geometrically isomorphic if" -> "geometrically isomorphic if they fulfil both of the following isomorphisms"

**Ethical Concerns:**

["NO or VERY MINOR ethics concerns only"]

**Final Justification:**

I stand by my original score. The authors addressed all of my queries and the the work appears technically solid, but between (1) my lack of understanding of both this paper and the relevant background work, and (2) the apparent lack of concrete practical applicability, I cannot raise my score above a borderline accept or my confidence beyond a 2. I hope that the reviews from the other reviewers who are more familiar with this work will be weighted higher, and overall it seems to me that the paper should probably be accepted.

**Limitations:**

Limitations are briefly discussed in Appendix C.
I would suggest including them in the main section of the paper.

**Paper Formatting Concerns:**

Nothing major. Although I am unsure whether the inline figures are allowed in the style.

**Quality:**

3

**Strengths And Weaknesses:**

Strengths:
1. The paper builds well on existing work, and situates itself well within the literature. A great example of this is L249 "FastEGNN [41], it is assumed that non-coplanar virtual nodes can always be found, but in the discussion later, we will see that this is not the case"
2. The paper provides a rigorous theoretical grounding and motivation.
3. The theories are backed up by strong empirical performance over the baselines.
4. The appendices are very thorough.

Weaknesses:
1. The theory in the paper was hard to follow as someone not familiar with the related work. It should ideally be more self-contained. For example, L128-L129 "as lth-degree steerable features if they are transformable using D(l)" -> This cannot be understood within the confines of the paper. What does "transformable using D" mean in this context, and what is the Wigner-D matrix? Also, L146-L147 "the basis set that expands the function space of equivariant GNNs." -> it is also unclear what this means within the confines of the paper.
2. In general, the figures are not well used and do not do much to help convey the ideas in the paper. E.g. the image in Figure 2 is not helpful, the math in Figure 1 is not understandable at this point of the paper, and Figure 3 is far too small to draw anything useful from.
3. The datasets chosen for evaluation are quite limited: there is only one real-world dataset (MD17) and they are not tested on large graphs. The authors claim that the 100-body system dataset "tests its scalability in large systems"; however, it is not large in comparison to the other datasets used in "Improving Equivariant Graph Neural Networks on Large Geometric Graphs via Virtual Nodes Learning". It is just 100 nodes, compared to their datasets Proteins (800 nodes) and Water-3D (8000 nodes). Thus it is unclear from the experiments how the method scales to larger graphs and real-world scenarios.

---

> ### Author Rebuttal · Authors · 2025-07-30
>
> Thank you very much for your positive feedback and thoughtful, detailed comments. We greatly appreciate your careful reading and constructive suggestions, which will help us improve both the clarity and impact of our paper. We summarize and respond to your questions as follows:
>
> > **R1. Improve Presentation and Increase Background Knowledge**
>
> Thank you very much for your valuable comments. Your suggestions are both professional and elegant, and we greatly appreciate your insights. We will make the following revisions to the manuscript:
>
> - **(W1):** We will refer to review [A] for additional background information on irreducible representations, Wigner-D matrices, and related concepts, and have incorporated these materials to enhance clarity for the reader.
> - **(W2):** We will enhance the description of Fig. 1 for better understanding, and enlarge Fig. 3 to improve readability.
> - **(Q1):** We will clarify that $C_H$ represents the number of channels in the node feature matrix $\boldsymbol{H}$.
> - **(MC):** We will adopt your suggested rephrasing, as it is indeed more elegant. Thank you again.
> - **(L1):** We will move the discussion of limitations in Appendix C to the main text.
>
> > **R2 (Q2). Fairness of the Uniform** **Node and Edge Features**
>
> Thank you very much for your thoughtful question. The non-uniform case can be readily generalized and addressed within our framework. Our assumption of uniform node and edge features is primarily made to simplify the theoretical analysis, particularly in the context of Def. 3.3 on geometric isomorphism. We believe this assumption is reasonable for the purposes of our current work. Nevertheless, we agree that it is important to explicitly justify and clarify such assumptions:
>
> - In practice, our analysis can be extended to non-uniform node/edge features by: (1) first determining geometric isomorphism based on graph structure, and (2) then ensuring that the mapping between nodes/edges also respects feature matching.
> - We will revise the text to more explicitly state the scope and limitations of this assumption, and include supplementary comments in the appendix on extending results to the non-uniform case.
>
> We also clarify that the purpose of Fig. 2 is to demonstrate the role of geometric structure (independent of feature information) in distinguishing points and facilitating the construction of valid virtual nodes.
>
> > **R3 (W3). Additional Experiments** **on** **Large-scale Geometric Graphs**
>
> Thank you very much for your insightful comment and we decided to add an experiment on large-scale scenario. We acknowledge that our experiments do not cover ultra-large graph datasets (with 1K+ nodes). However, the main emphasis of our work is on establishing a theoretical framework for completeness, and the experimental results are intended primarily to provide supporting evidence for our theoretical contributions. It is worth emphasizing that, in the field of atomistic modeling, systems with more than one hundred nodes are widely recognized in literature as sufficiently challenging benchmarks. We believe that our results on such systems already provide a meaningful demonstration of our model's practical capabilities.
>
> To further evaluate the performance of our model on larger datasets, we have conducted additional experiments on the more representative Water-3D (average >8K nodes/system) dataset in [B] as you suggested. Due to time limitations, we restrict the evaluation to a 1/5 subset called Water-3D-mini with 3,000/300/300 samples for train/valid/test from the original dataset (15,000/1,500/1,500), with a maximum training epoch of 1,000.
>
> **Table S1. Results on Water-3D-mini ($\times 10^{-4}$).**
>
> ||1-layer|2-layer|3-layer|4-layer|
> |-|-|-|-|-|
> |EGNN| 4.904|4.323|3.649|3.338|
> |FastEGNN| 4.885|4.332|3.782|3.259|
> |HEGNN|4.885|4.138|3.519|3.287|
> |EGNN$_\text{cpl}$-global|4.368|3.711|3.294|3.248|
> |EGNN$_\text{cpl}$-local|**3.611**|**3.320**|**2.803**|**2.495**|
>
> Our approach (EGNN$_\text{cpl}$-global/global) consistently outperforms the baselines at every depth, and achieves superior results with fewer layers. We believe this further demonstrates the efficiency and scalability of our approach, highlighting its potential for large-scale and real-world applications.
>
> > **R4 (Q3). Discussion on Algo. 4 and Models**
>
> Thank you for raising this interesting and important question. The internal structure of the virtual node corresponds to $\vec{\boldsymbol{V}}^\top\vec{\boldsymbol{V}}$ in Algo. 4. According to our theoretical framework, this feature should indeed be included. However, we chose not to incorporate it mainly due to observed performance degradation in our empirical studies. We believe there are two possible explanations for this: (1) The virtual nodes in our model are learnable, and their parameters—i.e., the virtual node feature—involve in message passing, potentially providing sufficient informational capacity already; (2) In our experiments, we found that introducing this feature tended to reduce training stability. Notably, similar issues regarding training instability with inner product-like vector features have been reported in literature like [B].
>
> ***
>
> We sincerely appreciate your valuable suggestions, and we will include these clarifications and additional results in the revised manuscript.
>
> > **Reference**
>
> - [A] Han J, Cen J, Wu L, et al. A survey of geometric graph neural networks: Data structures, models and applications. Frontiers of Computer Science 2025.
> - [B] Zhang Y, Cen J, Han J, et al. Improving equivariant graph neural networks on large geometric graphs via virtual nodes learning. ICML 2024.
> - [C] Huang W, Han J, Rong Y, et al. Equivariant graph mechanics networks with constraints. ICLR 2022.

---

> > ### Comment · Reviewer_q9Lr · 2025-08-01
> > **Response**
> >
> > My thanks to the authors. Their response was thorough and addressed all of my queries / concerns.
> > I have no further questions or disagreements.

---

> > > ### Author Response · Authors · 2025-08-01
> > >
> > > Dear Reviewer,
> > >
> > > Thank you very much for your kind response. We’re truly glad that our rebuttal fully addressed your concerns, and we sincerely appreciate your constructive feedback which helped improve the paper.
> > >
> > > If you believe the revisions merit it, we would be grateful if you could consider updating your final score accordingly.
> > >
> > > Best regards,
> > > The Authors

---

> > > ### Author Response · Authors · 2025-08-08
> > >
> > > Dear Reviewer,
> > >
> > > Thank you once again for your thoughtful follow-up and for letting us know that our rebuttal has fully addressed your concerns. We are truly grateful for the time and effort you have dedicated to reviewing our work, as well as for the constructive suggestions that have helped us improve the paper.
> > >
> > > As the rebuttal period is nearing its close, we kindly wanted to check whether, in light of the clarifications and improvements provided, you might consider updating your final score if you feel it is warranted. We would greatly appreciate it, as your evaluation is invaluable to the final decision process.
> > >
> > > Best regards, The Authors

---

### Official Review · Reviewer_Qhpc · 2025-07-02

**Clarity:** 3
**Significance:** 4
**Originality:** 4
**Rating:** 5
**Confidence:** 3

**Summary:**

The paper describes an advanced, theoretical grounded method for obtaining equivariant predictions from point clouds. The derivation is such that the authors can guarantee any equivariant function can be parametrized through an canonicalization procedure (complete scalar function) and a corresponding complete steerable basis set. The paper supports such claims via 5 theorems and experiments in which geometric isomorphism tests, in which the model succeeds in all cases, where other models (not using the completeness procedure) fail.

**Questions:**

See numbered comments/questions above. Thank you.

**Ethical Concerns:**

["NO or VERY MINOR ethics concerns only"]

**Final Justification:**

My final justification follows my last comment:

I have no further comments/questions. I appreciate the extra work done, and I agree that the reference [A] provides all the context needed regarding the g-conv paradigm. I am sure the paper will be much improved with the mentioned updates, and considering the paper was already of high quality I continue to support for acceptance and as such maintain my score "accept".

**Limitations:**

Are adequately addressed. Checklist is included.

**Paper Formatting Concerns:**

no concerns

**Quality:**

3

**Strengths And Weaknesses:**

# Strengths

1.	Apart from some challenges to get into this rather technical paper (see comments), the exposition of the main method (the algorithm) and the derivation of a *complete* equivariant model is clear and the results of Theorem 3.2, 3.5-8 are an important contribution to the field of equivariant deep learning. The derivation seems sound, I only thoroughly went through the proof of 3.2, but seem intuitively correct and I trust the proofs given in the appendix based on my understanding of canonicalization methods and steerable/tensor field methods.
2.	The method is thoroughly validated with controlled experiments that confirm the theory.
3.	The code is provided as supplementary materials. It looks clean and the paper seems as such very reproducible (I didn’t not try running the code though).
4. The paper is quite extensive in technical details, but could still benefit from some more intuitive clarifications

# Weaknesses
1.	Although theoretically very strong, and with experiments supporting the theoretical claims, the practical impact is at this stage hard to assess as the tasks other than completeness-type tests still seem a bit artificial. The paper would be more impactful if demonstrates something on atomic point clouds, where equivariant models typically make most impact and where there is a strong need for efficient models.
2.	Although generally complete in discussion of literature on equivariance (and well written!), it misses in my opinion a clear connection between the scalarization methods and tensor-product based methods. They are unified through the group convolution viewpoint (cf [Vadgama et al 2024]) in which the scalar class can be converted into the other via point-wise SO(3)-Fourier transforms and working with CG products instead of otherwise constraind linear layers. As such, the distinction into two classes is a bit arbitrary and often comes down to an implementation preference. Related is the exposition of the method relative to canonicalization methods (here categorized as frame-based methods) can be made more explicit. It seems that the presented paper is a form of canonicalization which simultaneously expands the predictions in a particular basis. Could the authors clarify how the work should be considered relative to canonicalization methods (e.g. [31])?
3.	The paper has many forward references to equations and algorithms which makes the reading sometimes a bit difficult. A striking example is “Actually, we have presented an algorithm …” which confuses me and I starting to doubt myself, have I missed something, or did the authors forget to put in a citation, or a reference to a section in the paper.

# Minor comments
1.	In 3.2 the authors talk about an “equivariant function”. I understand how an *operator* (or a map, or layer, or network) can be equivariant, namely via eq (1). But how is a function equivariant. E.g. the text states that spherical harmonics provide a compete basis for SO(3) equivariant functions, but I am not sure if I follow. The SHs provide a complete basis for functions over S^2, but not for SO(3), that would require all Wigner-D functions. Or are we talking about a basis for linear operators (like tensor products)? I don’t think so as the next line talks about “SE(3) equivariant functions over the sphere”. But isn’t any function over the sphere (regardless of basis) equiriant via the regular reprsentation phi(n) -> phi(R.n)? Or is this function to be considered vector valued, for example a convolution kernel? (cf Lang and Weiler, 2020 “A Wigner-Eckart Theorem for group Equivariant Convolution Kernels”, https://arxiv.org/abs/2010.10952 ). I am probably misinterpreting the statements here, but perhaps this can be clarified. It would help to make precise in what sense a spherical function is called equivariant.
2.	The claim of “without loss of generality” on line 154 should be further justified, I would recommend doing this in an appendix, together with a response to the following comment:
3.	Could you please include a detailed derivation of Example 3.1? I’m worried that something is not correct, namely that TFN is of order \nu=2 and EGNN of order \nu=1. Aren’t both of order \nu=2? TFN can be written in group convolution form, and only involves pairwise interactions (albeit with l>0). Same for Schnet (albeit with l=0). EGNN is not of convolution form due to the position update, but it’s still pairwise interactions. I don’t understand how TFN is different to EGN in this regard. The tensor products used only involve two quantities. Apart from this concern, it would be nice to see where the equation comes form because it seems quite fundamental to the rest of the exposition.
4.	In definition 3.3 I don’t think doublestruck M is defined.
5.	When forward referencing to the algorithms, please mention where we can fined them (in the appendix)

# References

Vadgama et al, 2024 “Probing Equivariance and Symmetry Breaking in Convolutional Networks” https://arxiv.org/abs/2501.01999

---

> ### Author Rebuttal · Authors · 2025-07-30
>
> Thank you for your thorough and insightful review. We are grateful for your positive comments on both the theoretical contributions and the literature discussion, as well as for your helpful suggestions to further strengthen the manuscript.
>
> > **R1 (W1). Additional Experiment on Atomic Dataset**
>
> Thanks for your suggestion. Since there are few equivariant tasks on atomic point clouds, we have now included results for the large-scale Water-3D dataset [A], which requires predicting future coordinates for fluid particles (average >8K nodes/system). Due to time constraints, we only evaluate a substantial subset (Water-3D-mini: 3,000/300/300 for train/val/test) using up to 1,000 training epochs.
>
> **Table S1. Results on Water-3D-mini ($\times 10^{-4}$).**
>
> ||1-layer|2-layer|3-layer|4-layer|
> |-|-|-|-|-|
> |EGNN| 4.904|4.323|3.649|3.338|
> |FastEGNN| 4.885|4.332|3.782|3.259|
> |HEGNN|4.885|4.138|3.519|3.287|
> |EGNN$_\text{cpl}$-global|4.368|3.711|3.294|3.248|
> |EGNN$_\text{cpl}$-local|**3.611**|**3.320**|**2.803**|**2.495**|
>
> Our approach (EGNN$_\text{cpl}$-global/local) consistently outperforms all baselines at every depth, and achieves superior results with fewer layers. These findings further support both the scalability and efficiency of our method of handling large, real-world systems.
>
> > **R2 (W2 & MC1). Theoretical Discussion**
>
> Thank you very much for your thoughtful and insightful comments, which have greatly enriched our understanding of the theoretical landscape.
>
> **Connotation of Model Methodology:** As you astutely observe, there are indeed two main perspectives on unifying scalarization-based and tensor-product-based methodologies. The first approach, which we employ in our work, relies on message passing consisting of message computation and aggregation and it achieves equivariant constraints by leveraging known equivariant variables. The second approach, based on group convolution and decomposition of the convolution kernel (representation matrix), as you have perceptively pointed out, proceeds from the opposite direction—deducing equivariant variables directly from established equivariant constraints. Historically, the second perspective predates the first, and can be traced back to Eqs. (10), (11), and (16) in the 3D Steerable CNN paper [A]. The two additional works you referenced are, in fact, further developments based on [A]. This method offers profound insight into the underlying algebraic structures; however, it is often less accessible to researchers without a strong background in group representation theory. As a consequence, the second approach is most prevalent in research concerning equivariance in physical domains, while the first approach tends to be favored within the literature on equivariant graph neural networks. We will include a dedicated section in the appendix of our revised manuscript to introduce this alternative methodology and will update the main text to include appropriate citations.
>
> **Spherical Harmonics and Bases:** We apologize for the confusion. Our intention was to convey that "for functions on the sphere that are equivariant under $\mathrm{SO}(3)$, the spherical harmonics form a basis," rather than to imply that "the spherical harmonics are a basis for $\mathrm{SO}(3)$" itself. We will clarify this statement in the revised manuscript to prevent further misunderstanding.
>
> **From Sphere to Euclidean Space:** This can be addressed by incorporating an RBF function, as described in Lines 141–143 of the original manuscript. This design is also inspired by [A] (see after Eq. (16) in Section 4.2). We sincerely appreciate your thoughtful suggestions, which have helped clarify these important distinctions.
>
> **Model Interpretation from Canonicalization:** Very nice comment! As you have pointed out, the essence of Frame-Averaging [31] lies in eliminating the influence of group transformations on the input by constructing a suitable frame, thereby enabling compatibility with any downstream architecture. From this perspective, we agree that the canonical form utilized in our method is conceptually aligned with the principles underlying Frame-Averaging. However, we would like to note that our specific implementation differs from this general framework. While Frame-Averaging [31] proposes a versatile approach that can be applied to a broad range of groups, our method focuses on designing an efficient canonicalization procedure tailored specifically to geometric graphs. This enables us to achieve computational efficiency while maintaining the desired equivariant properties. We greatly appreciate your suggestion, and we will clarify this relationship more explicitly in the revised manuscript to better position our contribution within the broader context. Thank you again for your valuable feedback.
>
> > **R4. Improve Presentation and Correct Typos**
>
> Thank you very much for your valuable feedback. Your questions and suggestions are elegant and insightful. We will make the following changes to the manuscript:
>
> - **(W3 & MC5):** We will include cross-references in the revised manuscript to improve readability.
> - **(MC2):** We will provide the corresponding proof in a later section. Here is a brief explanation: we prioritize structural information to determine whether two graphs are geometrically isomorphic. For cases where node/edge features are not uniform, we can further determine this based on matching (please see our reply to MC4).
> - **(MC3):** Sorry, this is a typo. EGNN/HEGNN and TFN both model based on edges, so they are 2-body interactions (number of neighbors $\nu=1$), while MACE corresponds to $\nu\geq 1$.
> - **(MC4):** We will modify the statement of "Point Cloud Isomorphism" in Def. 3.3 as follows: "The two point clouds $\vec{\boldsymbol{X}}^{(\mathcal{G})}$ and $\vec{\boldsymbol{X}}^{(\mathcal{H})}$ are isomorphic, i.e. the matching set $\mathbb{M}(\mathcal{G}, \mathcal{H})=\\{(\sigma, \mathfrak{g})\in S\_N\times \mathrm{E}(3)\mid\forall i, \vec{\boldsymbol{x}}_i^{(\mathcal{G})}= \mathfrak{g}\cdot\vec{\boldsymbol{x}}\_{\sigma(i)}^{(\mathcal{H})}\\}$ is not empty.".
>
> ***
>
> Thank you again for your comprehensive review and high-level comments, which have significantly helped us to improve the clarity, rigor, and impact of our work. We will incorporate all clarifications, additional experiments, and references as discussed in the revised manuscript.
>
> > **Reference**
>
> - [A] Weiler M, Geiger M, Welling M, et al. 3d steerable cnns: Learning rotationally equivariant features in volumetric data. NIPS 2018.

---

> ### Author Response · Authors · 2025-08-04
>
> Dear Reviewer,
>
> We would like to express our sincere appreciation for your thorough and insightful review of our manuscript. Your positive feedback on both the theoretical contributions and the discussion of related literature, as well as your constructive suggestions, have been invaluable in shaping our revisions.
>
> We would be very grateful if you could let us know whether our responses have satisfactorily addressed your comments, or if there are any further questions or points we could clarify.
>
> Thank you again for your time, careful reading, and constructive feedback.
>
> Best regards, The Authors

---

### Official Review · Reviewer_Y7Ue · 2025-07-03

**Clarity:** 3
**Significance:** 4
**Originality:** 4
**Rating:** 5
**Confidence:** 3

**Summary:**

This paper introduces a novel dynamic method for constructing complete equivariant graph neural networks (GNNs) for geometric graphs. Unlike previous methods that use infinite basis to achieve completeness, the proposed method only needs a complete scalar function (of the geometric graph) and a full rank steerable basis, which are finite. This way, the architecture doesn't need to use higher-order representations or stacking multiple layers to achieve completeness. The authors introduce an efficient implementation of the method via geometric isomorphism for asymmetric graphs, and empirically demonstrate its expressiveness and performance through synthetic and real-world datasets.

**Questions:**

1. As mentioned in weakness, can you explain the definition of "steerable function" in line 146, and explain the Thm 3.2 in more detail, such as what is  $F^{(l)}(G)$ , how does it depends on G, how to construct it (and $V^{(l)}(G) )?

2. For simpler dataset like tetrahedron center prediction, 5-body, 100-body system, the proposed dymaic method significantly improves over the baselines, but for MD17, the improvement becomes much less, with some even underperformed the baselines. Can you explain why?

**Ethical Concerns:**

["NO or VERY MINOR ethics concerns only"]

**Final Justification:**

All the issues I mentioned are resolved. And as per our discussion, I'll maintain my score of "5 Accept".

**Limitations:**

yes

**Quality:**

3

**Strengths And Weaknesses:**

**Strengths**:
1. Novelty and Significance. The author proposes a new framework for constructing complete equivariant maps for geometric graphs, which is fundamentally different from previous approaches and potentially expands the perspective on this important question. Plus, the proposed framework is clear, rigorous, and straightforward to implement.
2. Experiment. The experiments and results are solid, covering different tasks, a strong set of baselines and insightful analysis. And the performance improvements brought by the newly proposed method are significant.

**Weakness**:
1. Clarity. The explanation of some concepts can be improved. For instance, in Thm 3.2, I'm not quite sure how does the $F^(l)(G)$ depends on G, I didn't see definition of it previously. Also, it's better to explain what a geometric graph and equivariance for geometric graph is in the introduction section. As I mainly work with topological graphs, I was initially confused by the description of completenss of equivariant GNNs. Another example is in line 145-146, I'm not quite sure what is "steerable function", is it just "equivariant function" w.r.t. O(3)?

---

> ### Author Rebuttal · Authors · 2025-07-30
>
> Thank you very much for your valuable and constructive feedback. We greatly appreciate your careful reading, positive remarks, and thoughtful questions. Please find our detailed responses below:
>
> > **R1 (W1 & Q1). Clarification of Concepts**
>
> Thank you for highlighting areas where further clarification and improved definitions are needed. We acknowledge that terms such as "geometric graph" and "steerable function" may not be universally familiar to all readers, especially those who primarily work with topological graphs. We will address these points and strengthen the introductory section to include more context and definitions.
>
> **Geometric Graph and Equivariance:** We agree that a clearer distinction between geometric and topological graphs would benefit readers. In our context, a "geometric graph" refers to a graph not only with nodes and edges, but with associated coordinates in Euclidean space. Equivariance for geometric graphs then refers to functions whose outputs are transformed in a specific way—consistent with group actions (e.g., $\mathrm{O}(3)$ rotations and reflections)—whenever the graph’s geometric data is transformed. You may refer to review [A] for a more detailed introduction.
>
> **Steerable Function:** Indeed, a "steerable function" is synonymous with an "equivariant function" under the action of a group, particularly $\mathrm{O}(3)$ in our case. The term "steerable" is often used in signal processing and geometric deep learning communities to denote equivariant mappings of higher degree ($l>1$).
>
> **Clarification of why $\mathbb{F}^{(l)}(\mathcal{G})$ depends on $\mathcal{G}$ in Thm. 3.2:**  Here, $\mathbb{F}^{(l)}(\mathcal{G}) \subset \mathbb{R}^{2l+1}$ represents the set of all $l$th-degree equivariant function values on the geometric graph $\mathcal{G}$. Each such function outputs a vector in $\mathbb{R}^{2l+1}$, corresponding to the degrees of freedom for the representation of degree $l$ under $\mathrm{O}(3)$ transformations. Importantly, space $\mathbb{F}^{(l)}(\mathcal{G})$ may not span the entire $\mathbb{R}^{2l+1}$, depending on the structure of the graph $\mathcal{G}$ and its geometric realization. As discussed in [B] and outlined in Appendix A.1.3, in some configurations (i.e. symmetric graphs), the equivariant functions are restricted to a subspace. Thus, the expression and universality results must take these distinctions into account.
>
> **Construction of $\tilde{\boldsymbol{V}}^{(l)}(\mathcal{G})$:** The construction of the basis $\tilde{\boldsymbol{V}}^{(l)}(\mathcal{G})$ is two fold: 1. We formally prove (Thm. 3.8 and Appendix A.5) that such a basis always exists for the space of $l$th-degree equivariant functions for any $\mathcal{G}$; 2. In Section 3.5, we provide a concrete example: the set of coordinate differences $\\{\vec{\boldsymbol{x}}_{ij}\\}$ used in EGNNs can be leveraged to construct $\tilde{\boldsymbol{V}}^{(l)}(\mathcal{G})$ in practice.
>
> We will update the main text to make the definitions, dependencies, and construction procedures more transparent, and will provide these explanations in the main manuscript rather than only in appendices.
>
> > **R2 (Q2). Discussion of** **Performance** **Improvement Difference** **over Baselines**
>
> Thank you for your insightful question regarding the performance on the MD17 dataset. Indeed, we notice that the improvements of our dynamic method are less pronounced on MD17, with some metrics even underperforming compared to the baselines. We believe this is due to the following reasons:
>
> - Baselines such as Equiformer and HEGNN directly utilize high-degree steerable features, which are closely related to spherical harmonics and can effectively represent complex patterns (such as electron cloud shapes) essential for molecular dynamics tasks. While their architectural advantages may contribute significantly to performance in this setting, it is important to note that computing high-degree steerable features also introduces substantial additional computational overhead.
> - In contrast, our approach does not rely on the high-degree steerable features. Nonetheless, our method achieves superior performance on 5 out of 8 tasks and demonstrates better generalizability.
>
> In addition, we supplement our work with a challenging experiment on the large-scale Water-3D dataset [C], which involves predicting future coordinates for fluid particles in systems averaging over 8,000 nodes each. Due to time constraints, our evaluation is conducted on a substantial subset (Water-3D-mini: 3,000/300/300 for train/validation/test) with training capped at 1,000 epochs.
>
> **Table S1. Results on Water-3D-mini ($\times 10^{-4}$).**
>
> ||1-layer|2-layer|3-layer|4-layer|
> |-|-|-|-|-|
> |EGNN| 4.904|4.323|3.649|3.338|
> |FastEGNN| 4.885|4.332|3.782|3.259|
> |HEGNN|4.885|4.138|3.519|3.287|
> |EGNN$_\text{cpl}$-global|4.368|3.711|3.294|3.248|
> |EGNN$_\text{cpl}$-local|**3.611**|**3.320**|**2.803**|**2.495**|
>
> Our approach (EGNN$_\text{cpl}$-global/local) consistently outperforms all baselines at every depth, and achieves superior results with fewer layers. These findings further support both the scalability and efficiency of our method of handling large, real-world systems.
>
> ***
>
> Thank you again for your generous recognition of our work; your encouragement is truly appreciated. We will address these considerations and limitations in the revised manuscript to provide a more balanced perspective on the strengths and applicable scenarios of our method.
>
> > **Reference**
>
> - [A] Han J, Cen J, Wu L, et al. A survey of geometric graph neural networks: Data structures, models and applications. Frontiers of Computer Science 2025.
> - [B] Cen J, Li A, Lin N, et al. Are high-degree representations really unnecessary in equivariant graph neural networks? NeurIPS 2024.
> - [C] Zhang Y, Cen J, Han J, et al. Improving equivariant graph neural networks on large geometric graphs via virtual nodes learning. ICML 2024.

---

> > ### Comment · Reviewer_Y7Ue · 2025-08-06
> >
> > Dear authors,
> >
> > Thank you for your detailed answers to my questions. I have no further questions/concerns about the paper. And thank you for mentioning the additional result, which is a good proof of the strength of the proposed method/framework. I will maintain my score of "5 Accept".

---

> > > ### Author Response · Authors · 2025-08-06
> > >
> > > Dear Reviewer,
> > >
> > > We are very pleased to hear that our responses have addressed your concerns. We sincerely appreciate your thoughtful suggestions on improving the presentation, and we will further revise the manuscript to enhance its readability and accessibility for a broader audience. Thank you once again for your support—it is truly encouraging to us.
> > >
> > > Best regards, The Authors

---

> ### Author Response · Authors · 2025-08-04
>
> Dear Reviewer,
>
> We would like to express our sincere gratitude for your careful review and positive evaluation of our paper. Your encouraging remarks and constructive suggestions have been truly valuable to us.
>
> We would be most grateful if you could kindly let us know whether our responses have fully resolved your concerns, or if there are any remaining questions or points we could further clarify.
>
> Thank you once again for your time and insightful feedback.
>
> Best regards, The Authors

---

### Official Review · Reviewer_yiPB · 2025-07-03

**Clarity:** 3
**Significance:** 3
**Originality:** 3
**Rating:** 4
**Confidence:** 4

**Summary:**

This paper presents a theoretically grounded and efficient framework for constructing complete equivariant graph neural networks (GNNs). While existing methods often rely on deep architectures, high-order body interactions, or high-dimensional steerable features to achieve universal approximation capability, they typically incur high computational costs. The authors show that completeness can be achieved through two key components: (1) a complete scalar function, referred to as the canonical form of the geometric graph, and (2) a full-rank steerable basis set. Based on this theoretical analysis, they propose an efficient algorithm for building complete equivariant GNNs using EGNN and TFN as base models. Experimental results demonstrate that the proposed approach achieves superior completeness and strong performance with only a few layers, significantly reducing computational overhead while maintaining practical effectiveness.

**Questions:**

See Weaknesses.

**Ethical Concerns:**

["NO or VERY MINOR ethics concerns only"]

**Final Justification:**

The theory presented in this paper does not have obvious flaws. In the rebuttal, the authors addressed some of my major concerns, particularly whether such invariant features can truly be effective in practical applications. I believe the paper's completeness meets the acceptance criteria. However, I cannot rate this paper higher than a 4, because the practical applications provided by the authors are quite limited, using non-mainstream datasets, and thus no comparison with state-of-the-art equivariant models is included. The real-world utility of the proposed approach still needs to be validated over time.

**Limitations:**

YES

**Quality:**

3

**Strengths And Weaknesses:**

**Strengths**

- 1. This paper systematically summarizes several tensor-product modules.

- 2. The paper is clearly written and well-organized.

- 3. The canonical form of scalar functions may help replace certain equivariant features, offering a new perspective on model design.

**Weaknesses**

- 1. The theorems in this paper mainly focus on proving completeness, but lack theoretical analysis regarding expressive power, similar to Theorem 2 in [1]. I would like to know whether this canonical form can only substitute for simple (low-order) equivariant functions in practical applications.

- 2. Constructing GNNs using high-quality invariant features is also a common practice. In my opinion, the authors should compare their method with existing invariant representations, such as FPFH in point cloud processing or the embedding layer of GotenNet [2].

- 3. The authors could try more complex real-world systems, such as QM9.

- 4. Theorem 3.5 is more like a definition.

[1] Nadav Dym and Haggai Maron. On the universality of rotation equivariant point cloud networks

[2] Sarp Aykent and Tian Xia. GotenNet: Rethinking Efficient 3D Equivariant Graph Neural Networks

---

> ### Author Rebuttal · Authors · 2025-07-30
>
> Thank you for your comments and questions, and we salute your hard work and time in the review process. We summarize and respond to your questions as follows:
>
> > **R1 (W1). Discussion of Expressive Power**
>
> Thank you for highlighting this important aspect. Expressive power refers to a model's ability to approximate functions. In general, the more functions a model can accurately approximate, the greater its expressive power. Completeness, actually, denotes the strongest level of expressive power: it is the capacity of a model to approximate any equivariant function to arbitrary precision.
>
> Both of Thm. 3.2 in our paper and Thm. 2 in D-spanning [1] ultimately establish completeness, although their derivation processes differ substantially. Specifically, whereas D-spanning [1] relies on constructing equivariant polynomials via sufficient layers of tensor-product operations, our approach requires a set of complete invariant scalar functions along with a full-rank steerable basis set.
>
> Furthermore, our canonical form can substitute for complex and high-order equivariant functions in practical applications (not just simple and low-order equivariant functions), as our porposed model is a complete model  of arbitrary degree $l$.
>
> > **R2 (W2). Comparison with Existing Invariant Representations**
>
> We are grateful for your insightful suggestion regarding comparisons with common invariant representations such as FPFH and GotenNet. In response, we have conducted additional experiments as follows:
>
> 1. Inv$_\text{FPFH}$: We construct additional node features using the FPFH descriptor and introduce them into the EGNN framework.
> 2. Inv$_\text{GotenNet}$: We similarly inject node features generated by the embedding layer of GotenNet.
>
> All methods are evaluated on the N-body dataset, with the results presented below (Table S1):
>
> **Table S1. Results on the N-body Dataset ($\times 10^{-2}$).**
>
> ||1-layer|2-layer|3-layer|4-layer|
> |-|-|-|-|-|
> |EGNN|4.214|0.780|0.710|0.712|
> |HEGNN|4.114|0.801|0.561|0.489|
> |EGNN+inv$_\text{FPFH}$|3.800|0.701|0.487|0.674|
> |EGNN+inv$_\text{GotenNet}$|2.307|1.922|2.083|1.030|
> |EGNN$_{\text{cpl}}^*$-global|0.985|0.554|0.533|0.498|
> |EGNN$_{\text{cpl}}^*$-local|**0.703**|**0.513**|**0.455**|**0.450**|
>
> We observe that, FPFH-derived features can enhance EGNN performance, and GotenNet embeddings only outperform EGNN at shallower depths. In contrast, our model consistently achieves leading or comparable results, especially as depth increases. We will discuss these insights more thoroughly in the revised manuscript.
>
> > **R3 (W3). Additional experiments on Complex Real-world Systems**
>
> Thank you for encouraging us to investigate more challenging datasets. As QM9 is primarily an invariant prediction task, but our work aims to advance the performance of equivariant functions, we instead select the large-scale Water-3D dataset from [A] for evaluation. This dataset involves an equivariant task to predict future coordinates for fluid particles (average >8K nodes/system). Due to time constraints, we only evaluate a substantial subset (Water-3D-mini: 3,000/300/300 for train/val/test) using up to 1,000 training epochs.
>
> **Table S2. Results on Water-3D-mini ($\times 10^{-4}$).**
>
> ||1-layer|2-layer|3-layer|4-layer|
> |-|-|-|-|-|
> |EGNN| 4.904|4.323|3.649|3.338|
> |FastEGNN| 4.885|4.332|3.782|3.259|
> |HEGNN|4.885|4.138|3.519|3.287|
> |EGNN$_\text{cpl}$-global|4.368|3.711|3.294|3.248|
> |EGNN$_\text{cpl}$-local|**3.611**|**3.320**|**2.803**|**2.495**|
>
> Our approach (EGNN$_\text{cpl}$-global/local) consistently outperforms the baselines at every depth, and achieves superior results with fewer layers. These findings further support both the scalability and efficiency of our method of handling large, real-world systems.
>
> > **R4 (W4). Improved the Presentation of Original Thm. 3.5**
>
> We apologize for our previous confusing presentation. Our intention was to present a theorem regarding the time complexity of Algo. 3. We will clearly state this theorem and provide a detailed proof to ensure the correctness and completeness of our claims in the revised manuscript.
>
> ***
>
> Thank you once again for your constructive feedback, which has directly contributed to improving the clarity, empirical evaluation, and theoretical discussion of our work. We will incorporate all clarifications and additional results into the revised version of the manuscript. We sincerely hope that our responses and revisions will address your concerns, and we would be grateful if you might consider these improvements in your evaluation.
>
> > **Reference**
>
> - [A] Zhang Y, Cen J, Han J, et al. Improving equivariant graph neural networks on large geometric graphs via virtual nodes learning. ICML 2024.

---

> ### Author Response · Authors · 2025-08-04
>
> Dear Reviewer,
>
> Thank you very much for your thoughtful and valuable feedback, and for the considerable time and effort you devoted to reviewing our work.
>
> If you have any further questions or concerns regarding our submission, please do not hesitate to let us know. We would greatly appreciate any additional opportunity for discussion or clarification that could further improve our work. We also sincerely hope that, in light of the revisions and clarifications we have provided, you might kindly reconsider your evaluation of our manuscript.
>
> Thank you once again for your valuable input and attention. We look forward to your reply.
>
> Best regards, The Authors

---

> ### Comment · Reviewer_yiPB · 2025-08-05
>
> Thank you for your rebuttal. Most concerns addressed and score raised.
>
> However, one point needs to be clarified: although QM9 predicts invariant features, it heavily relies on higher-degree equivariant features, such as energy-prediction methods. Invariant features are essentially a special case of equivariant features (with $l=0$). Many works based on spherical harmonic representations employ higher-degree equivariant representations to learn atomic interactions during the message-passing process, ultimately compressing these higher-degree equivariant features ($l>0$) into invariant ones. Therefore, I believe experiments on datasets like QM9, and even larger ones such as Molecule3D and OC20, are necessary, as they can accurately reflect whether such invariant representations are truly effective. I hope the authors can include related case studies in future work.

---

> > ### Author Response · Authors · 2025-08-05
> >
> > Dear Reviewer,
> >
> > Thank you very much for your recognition and constructive comments on our work. Your thoughtful feedback has greatly improved the quality and depth of our manuscript. With regard to further exploration on datasets such as QM9, we are committed to pursuing this direction in future work, including investigating how to leverage the strengths of existing invariant representations such as FPFH and GotenNet.
> >
> > Best regards, The Authors

---

### Author Response · Authors · 2025-08-09
**Rebuttal Acknowledgment**

Dear Reviewers and Area Chair,

We sincerely thank you for your time, effort, and thoughtful consideration in reviewing our paper. We greatly appreciate your recognition of the significance of the problems we tackle, the theoretical contributions we present, and the models we develop. Your insightful feedback has been a source of valuable inspiration and has meaningfully enhanced the quality of our work.

In this study, we introduce a unified theoretical framework and efficient algorithms for constructing complete equivariant GNNs, establishing that completeness can be achieved through the canonical form of the geometric graph and a full-rank steerable basis. Our proposed models realize completeness and demonstrate competitive empirical performance with fewer layers and reduced computational cost.

As the discussion phase concludes, we are pleased to have addressed the vast majority of the reviewers' concerns. We are grateful that Reviewer yiPB kindly raised the score from "borderline reject", Reviewers Y7Ue and Qhpc maintained their "clear accept", and Reviewer q9Lr acknowledged our response as thorough and responsive to all concerns. It is a genuine honor to receive uniformly positive feedback from all four reviewers, which serves as a great encouragement for our work. Our theoretical framework was described as "theoretically grounded" (Reviewers yiPB, Qhpc & q9Lr), "fundamentally different from previous approaches," and "clear, rigorous, and straightforward" (Reviewer Y7Ue). The proposed model was also praised for its "superior completeness and strong performance" (Reviewer yiPB); "solid, covering different tasks, a strong set of baselines, and insightful analysis" (Reviewer Y7Ue); "thoroughly validated," "very reproducible" (Reviewer Qhpc); and for its "strong empirical performance over the baselines" (Reviewer q9Lr). The reviewers' expert analyses and constructive feedback have been invaluable, and we will integrate all pertinent suggestions into our revised manuscript.

Finally, we wish to extend our special thanks to the Area Chair for the guidance and leadership in this process. We are truly grateful for the collective efforts which have helped refine and strengthen our work.

Thank you once again for your dedication and support.

Best regards,
The Authors

---

### Decision · Program_Chairs · 2025-09-17

**Decision:**

Accept (poster)

**Comment:**

This paper introduces a theoretically grounded and efficient framework for constructing complete equivariant GNNs (especially relevant for geometric graphs and point clouds). Completeness is achieved not through deep stacks or high-order features, but via two finite ingredients: a complete scalar function (graph canonical form) and a full-rank steerable basis. The authors provide two algorithms, one based on geometric isomorphism, and one faster assuming learnable non-coplanar reference nodes. These are instantiated on EGNN and TFN. The method achieves universality, good accuracy with fewer layers, and reduced computational overhead. Experiments complement the theoretical treatment. Generally the reviewers have liked the paper and have made suggestions regarding clarity and experiments. I mostly concur with the positive assessment of the reviewers. The authors are advised to include most of the points in their rebuttal into the main paper.